# Bridging State and History Representations: Understanding Self-Predictive RL

**Tianwei Ni♣, Benjamin Eysenbach♠, Erfan Seyedsalehi◇\*, Michel Ma♣, Clement Gehring♣, Aditya Mahajan◇, Pierre-Luc Bacon♣**

♣Mila, Université de Montréal, ♠Princeton University, ◇Mila, McGill University

{tianwei.ni, michel.ma, clement.gehring, pierre-luc.bacon}@mila.quebec,
eysenbach@princeton.edu, erfan.seyedsalehi@mail.mcgill.ca, aditya.mahajan@mcgill.ca

## Abstract

Representations are at the core of all *deep* reinforcement learning (RL) methods for both Markov decision processes (MDPs) and partially observable Markov decision processes (POMDPs). Many representation learning methods and theoretical frameworks have been developed to understand what constitutes an effective representation. However, the relationships between these methods and the shared properties among them remain unclear. In this paper, we show that many of these seemingly distinct methods and frameworks for state and history abstractions are, in fact, based on a common idea of *self-predictive* abstraction. Furthermore, we provide theoretical insights into the widely adopted objectives and optimization, such as the stop-gradient technique, in learning self-predictive representations. These findings together yield a minimalist algorithm to learn self-predictive representations for states and histories. We validate our theories by applying our algorithm to standard MDPs, MDPs with distractors, and POMDPs with sparse rewards. These findings culminate in a set of preliminary guidelines for RL practitioners.[1]

## 1 Introduction

Reinforcement learning holds great potential to automatically learn optimal policies, mapping observations to return-maximizing actions. However, the application of RL in the real world encounters challenges when observations are high-dimensional and/or noisy. These challenges become even more severe in partially observable environments, where the observation (history) dimension grows over time. In fact, current RL algorithms are often brittle and sample inefficient in these settings (Wang et al., 2019; Stone et al., 2021; Tomar et al., 2021; Morad et al., 2023).

To address the curse of dimensionality, a substantial body of work has focused on compressing observations into a latent state space, known as state abstraction in MDPs (Dayan, 1993; Dean & Givan, 1997; Li et al., 2006), history abstraction in POMDPs (Littman et al., 2001; Castro et al., 2009), and sufficient statistics or information states in stochastic control (Striebel, 1965; Kwakernaak, 1965; Bohlin, 1970; Kumar & Varaiya, 1986). Traditionally, this compression has been achieved through hand-crafted feature extractors (Sutton, 1995; Konidaris et al., 2011) or with the discovery of a set of core tests sufficient for predicting future observations (Littman et al., 2001; Singh et al., 2003). Modern approaches learn the latent state space using an encoder to automatically filter out irrelevant parts of observations (Lange & Riedmiller, 2010; Watter et al., 2015; Munk et al., 2016). Furthermore, *deep* RL enables end-to-end and online learning of compact state or history representations alongside policy training. As a result, numerous representation learning techniques for RL have surfaced (refer to Table 1), drawing inspiration from diverse fields within ML and RL. However, this abundance of methods may have inadvertently presented practitioners with a "paradox of choice", hindering their ability to identify the best approach for their specific RL problem.

This paper aims to offer systematic guidance regarding the essential characteristics that good representations should possess in RL (the "**what**") and effective strategies for learning such representations (the "**how**"). We begin our analysis from first principles by comparing and connecting various representations proposed in prior works for MDPs and POMDPs, resulting in a unified view. Remarkably, these representations are all connected by a **self-predictive** condition – the encoder can predict its next latent state (Subramanian et al., 2022). Next, we examine how to learn such self-predictive condition

---

\*Work done while ES was at McGill University.

[1]Please refer to https://arxiv.org/abs/2401.08898 for the 10-main-page version of this paper.

in RL, a difficult subtask due to the bootstrapping effect (Gelada et al., 2019; Schwarzer et al., 2020; Tang et al., 2022). We provide fresh insights on why the popular "stop-gradient" technique, in which the parameters of the encoder do not update when used as a target, has the promise of learning the desired condition without representational collapse in POMDPs. Building on our new theoretical findings, we introduce a minimalist RL algorithm that learns self-predictive representations end-to-end with a *single* auxiliary loss, *without* the need for reward model learning (thereby removing planning) (François-Lavet et al., 2019; Gelada et al., 2019; Tomar et al., 2021; Hansen et al., 2022; Ghugare et al., 2022; Ye et al., 2021; Subramanian et al., 2022), reward regularization (Eysenbach et al., 2021), multi-step predictions and projections (Schwarzer et al., 2020; Guo et al., 2020), and metric learning (Zhang et al., 2020; Castro et al., 2021). Furthermore, the simplicity of our approach allows us to investigate the role of representation learning in RL, in isolation from policy optimization.

The core contributions of this paper are as follows. We establish a unified view of state and history representations with novel connections (Sec. 3), revealing that many prior methods optimize a collection of closely interconnected properties, each representing a different facet of the same fundamental concept. Moreover, we enhance the understanding of self-predictive learning in RL regarding the choice of the objective and its impact on the optimization dynamics (Sec. 4). Our theory results in a simplified and novel RL algorithm designed to learn self-predictive representations fully end-to-end (Sec. 4.3). Through extensive experimentation across three benchmarks (Sec. 5), we provide empirical evidence substantiating all our theoretical predictions using our simple algorithm. Finally, we offer our recommendations for RL practitioners in Sec. 6. Taken together, we believe that our work potentially aids in addressing the longstanding challenge of learning representations in MDPs and POMDPs.

## 2 BACKGROUND

**MDPs and POMDPs.** In the context of a POMDP $\mathcal{M}_O = (\mathcal{O}, \mathcal{A}, P, R, \gamma, T)$[2], an agent receives an observation $o_t \in \mathcal{O}$ at time step $t$, selects an action $a_t \in \mathcal{A}$ based on the observed history $h_t := (h_{t-1}, a_{t-1}, o_t) \in \mathcal{H}_t$[3], and obtains a reward $r_t \sim R(h_t, a_t)$ along with the subsequent observation $o_{t+1} \sim P(\cdot \mid h_t, a_t)$. The initial observation $h_1 := o_1$ is sampled from the distribution $P(o_1)$. The total time horizon is denoted as $T \in \mathbb{N}^+ \cup \{+\infty\}$, and the discount factor is $\gamma \in [0, 1]$ (less than 1 for infinite horizon). To maintain brevity, we employ the "prime" symbol to represent the next time step, for example writing $h' = (h, a, o')$. Under the above assumptions, our agent acts according to a policy $\pi(a \mid h)$ with action-value $Q^\pi(h, a)$. Furthermore, it can be shown that there exists an optimal value function $Q^*(h, a)$ such that $Q^*(h, a) = \mathbb{E}[r \mid h, a] + \gamma \mathbb{E}_{o' \sim P(\mid h, a)}[\max_{a'} Q^*(h', a')]$, and a deterministic optimal policy $\pi^*(h) = \operatorname{argmax}_a Q^*(h, a)$. In an MDP $\mathcal{M}_S = (\mathcal{S}, \mathcal{A}, P, R, \gamma, T)$, the observation $o_t$ and history $h_t$ are replaced by the state $s_t \in \mathcal{S}$[4].

**State and history representations.** In a POMDP, an **encoder** is a function $\phi : \mathcal{H}_t \to \mathcal{Z}$ that produces a history **representation** $z = \phi(h) \in \mathcal{Z}$. Similarly, in an MDP, we replace $h$ with $s$, resulting in a state encoder $\phi : \mathcal{S} \to \mathcal{Z}$ and a state representation $z = \phi(s) \in \mathcal{Z}$. This representation is known as an "abstraction" (Li et al., 2006) or a "latent state" (Gelada et al., 2019). Such encoders are sometimes shared and simultaneously updated by downstream components (*e.g.* policy, value, world model) of an RL system (Hafner et al., 2020a; Hansen et al., 2022). In this paper, we are interested in such a shared encoder, or **the encoder learned for the value function** if the encoders are separately learned.

Below, we present the key abstractions that are central to this paper, along with their established connections. We will highlight the **conditions** met by each abstraction. We defer additional common abstractions and related concepts to Sec. A.2.

**1. $Q^*$-irrelevance abstraction.** An encoder $\phi_{Q^*}$ provides a $Q^*$-irrelevance abstraction (Li et al., 2006) if it contains the necessary information for predicting the return. Formally, if $\phi_{Q^*}(h_i) = \phi_{Q^*}(h_j)$, then $Q^*(h_i, a) = Q^*(h_j, a), \forall a$. A $Q^*$-irrelevance abstraction can be achieved as a by-product of learning an encoder $\phi$ through a value function $\mathcal{Q}(\phi(h), a)$ end-to-end using model-free RL. If the optimal values match, then $\mathcal{Q}^*(\phi_{Q^*}(h), a) = Q^*(h, a), \forall h, a$.

**2. Self-predictive (model-irrelevance) abstraction.** We view the model-irrelevance concept (Li et al., 2006) from a self-predictive standpoint. Specifically, a model-irrelevant encoder $\phi_L$ fulfills two

---

[2]While the classic definition of a POMDP (Cassandra et al., 1994) features a state space, we assume it to be unknown, thus our view of a POMDP is a black-box input-output system (Subramanian et al., 2022).

[3]In general, $h_t := (h_{t-1}, a_{t-1}, r_{t-1}, o_t)$ (Izadi & Precup, 2005). In this study, we assume rewards are inaccessible during policy inference.

[4]In finite-horizon MDPs, we assume $s$ includes the time step $t$.

conditions: **expected reward prediction (RP)** and **next latent state distribution prediction (ZP)**[5], ensuring that the encoder can be used to predict expected reward and the next latent state distribution.

$$\exists R_z : \mathcal{Z} \times \mathcal{A} \to \mathbb{R}, \quad s.t. \quad \mathbb{E}[r \mid h, a] = R_z(\phi_L(h), a), \quad \forall h, a, \tag{RP}$$

$$\exists P_z : \mathcal{Z} \times \mathcal{A} \to \Delta(\mathcal{Z}), \quad s.t. \quad P(z' \mid h, a) = P_z(z' \mid \phi_L(h), a), \quad \forall h, a, z', \tag{ZP}$$

$$\mathbb{E}[z' \mid h, a] = \mathbb{E}[z' \mid \phi_L(h), a], \quad \forall h, a. \tag{EZP}$$

A weak version of ZP is the **expected next latent state $z$ prediction (EZP)** condition. ZP can be interpreted as a sufficient statistics condition on $\phi_L$: the next latent state $z'$ is conditionally independent of the history $h$ when $\phi_L(h)$ and $a$ is known, symbolized as $z' \perp\!\!\!\perp h \mid \phi_L(h), a$. Satisfying ZP only is trivial and can be achieved by employing a constant representation $\phi(h) = c$, where $c$ is a fixed constant. Therefore, ZP must be used in conjunction with other conditions (*e.g.*, RP) to avoid such degeneration. The $\phi_L$ is known as a bisimulation generator (Givan et al., 2003) in MDPs and an information state generator (Subramanian et al., 2022) in POMDPs.

**3. Observation-predictive (belief) abstraction.** This abstraction is implicitly introduced by Subramanian et al. (2022), which we denote by $\phi_O$, and satisfies three conditions: expected reward prediction RP, **recurrent encoder (Rec)** and **next observation distribution prediction (OP)**[6].

$$\exists \psi_z : \mathcal{Z} \times \mathcal{A} \times \mathcal{O} \to \mathcal{Z}, \quad s.t. \quad \phi(h') = \psi_z(\phi_O(h), a, o'), \quad \forall h, a, o', \tag{Rec}$$

$$\exists P_o : \mathcal{Z} \times \mathcal{A} \to \Delta(\mathcal{O}), \quad s.t. \quad P(o' \mid h, a) = P_o(o' \mid \phi_O(h), a), \quad \forall h, a, o', \tag{OP}$$

$$\exists \psi_o : \mathcal{Z} \to \mathcal{O}, \quad s.t. \quad o = \psi_o(\phi_O(h)), \quad \forall h. \tag{OR}$$

Similarly, the OP condition is equivalent to $o' \perp\!\!\!\perp h \mid \phi_O(h), a$, and OP is closely related to **observation reconstruction (OR)**, widely used in practice (Yarats et al., 2021). The recurrent encoder (Rec) condition is satisfied for encoders parameterized with feedforward or recurrent neural networks (Elman, 1990; Hochreiter & Schmidhuber, 1997), but not Transformers (Vaswani et al., 2017). In this paper, we assume the Rec condition is always satisfied. In POMDPs, $\phi_O$ is well-known as a belief state generator (Kaelbling et al., 1998).

We extend the relations between these abstractions known in MDPs (Li et al., 2006) to POMDPs.

**Theorem 1** (**Relationships between common abstractions (informal)**). *An encoder satisfying $\phi_O$ also belongs to $\phi_L$; an encoder satisfying $\phi_L$ also belongs to $\phi_{Q^*}$; the reverse is not necessarily true.*

## 3 A UNIFIED VIEW ON STATE AND HISTORY REPRESENTATIONS

### 3.1 AN IMPLICATION GRAPH OF REPRESENTATIONS IN RL

Using the taxonomy of state and history abstractions, it becomes possible to establish theoretical links among different representations and their respective conditions discussed earlier. These connections are succinctly illustrated in a directed graph, as shown in Fig. 1. In this section, we highlight the most significantly novel finding, while postponing the other propositions and proofs to Sec. A.

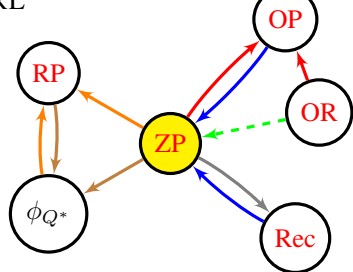

Figure 1: **An implication graph** showing the relations between the conditions on history representations. The source nodes of the edges with the same color *together* imply the target node. The dashed edge means it only applies to MDPs. All the connections are discovered in this work, except for (1) OP + Rec implying ZP, (2) ZP + RP implying $\phi_{Q^*}$.

The definition of self-predictive and observation-predictive abstractions suggests the classic *phased* training framework. In phased training, we alternatively train an encoder to predict expected rewards (RP) and predict next latent states (ZP) or next observations (OP), and also train an RL or planning agent on the latent space with the encoder "detached" from downstream components. On the other hand, we show in our Thm. 2 that if we learn an encoder *end-to-end* in a model-free fashion but using ZP (or OP) as an auxiliary task, then the ground-truth expected reward can be induced by the latent $Q$-value and latent transition. Thus, the encoder also satisfies RP and generates $\phi_L$ (or $\phi_O$) representation already.

**Theorem 2** (**ZP + $\phi_{Q^*}$ imply RP**). *If an encoder $\phi$ satisfies ZP, and $\mathcal{Q}(\phi(h), a) = Q^*(h, a), \forall h, a$, then we can construct a latent reward function $\mathcal{R}_z(z, a) := \mathcal{Q}(z, a) - \gamma \mathbb{E}_{z' \sim P_z(\cdot \mid z, a)}[\max_{a'} \mathcal{Q}(z', a')]$, such that $\mathcal{R}_z(\phi(h), a) = \mathbb{E}[r \mid h, a], \forall h, a.$*

---

[5]RP and ZP are labeled as (P1) and (P2), respectively, in Subramanian et al. (2022).

[6]OP and Rec are labeled as (P2a) and (P2b), respectively, in Subramanian et al. (2022).

Table 1: **Which optimal representation will be learned by the value function in prior works?** The "PO" column shows if the approach applies to POMDPs. The "Conditions" column shows the conditions that the encoder of the optimal value satisfies (see the appendix for the "metric" and "regularization" conditions). The ZP loss shows the loss function they use to learn ZP condition. The ZP target shows whether they use online or stop-gradient (including detached and EMA) encoder target. Due to the space limit, we omit the citations for recurrent model-free RL (Hausknecht & Stone, 2015; Kapturowski et al., 2018; Ni et al., 2022) and belief-based methods (Wayne et al., 2018; Hafner et al., 2019; Han et al., 2020; Lee et al., 2020).

| Work | PO? | Abstraction | Conditions | ZP loss | ZP target |
|---|---|---|---|---|---|
| Model-Free & Classic Model-Based RL | ✗ | $\phi_{Q^*}$ | $\phi_{Q^*}$ | N/A | N/A |
| MuZero (Schrittwieser et al., 2020) | ✗ | unknown | $\phi_{Q^*}$ + RP | N/A | N/A |
| MICo (Castro et al., 2021) | ✗ | unknown | $\phi_{Q^*}$ + metric | N/A | N/A |
| CRAR (François-Lavet et al., 2019) | ✗ | $\phi_L$ | $\phi_{Q^*}$ + RP + ZP + reg. | $\ell_2$ | online |
| DeepMDP (Gelada et al., 2019) | ✗ | $\phi_L$ | $\phi_{Q^*}$ + RP + ZP | W ($\ell_2$) | online |
| SPR (Schwarzer et al., 2020) | ✗ | $\phi_L$ | $\phi_{Q^*}$ + ZP | cos | EMA |
| DBC (Zhang et al., 2020) | ✗ | $\phi_L$ | $\phi_{Q^*}$ + RP + ZP + metric | FKL | detached |
| LSFM (Lehnert & Littman, 2020) | ✗ | $\phi_L$ | $\phi_{Q^*}$ + RP + EZP | SF | detached |
| Baseline in (Tomar et al., 2021) | ✗ | $\phi_L$ | $\phi_{Q^*}$ + RP + ZP | $\ell_2$ | detached |
| EfficientZero (Ye et al., 2021) | ✗ | $\phi_L$ | $\phi_{Q^*}$ + RP + ZP | cos | detached |
| TD-MPC (Hansen et al., 2022) | ✗ | $\phi_L$ | $\phi_{Q^*}$ + RP + ZP | $\ell_2$ | EMA |
| ALM (Ghugare et al., 2022) | ✗ | $\phi_L$ | $\phi_{Q^*}$ + ZP | RKL | EMA |
| TCRL (Zhao et al., 2023) | ✗ | $\phi_L$ | RP + ZP | cos | EMA |
| OFENet (Ota et al., 2020) | ✗ | $\phi_O$ | $\phi_{Q^*}$ + OP | N/A | N/A |
| Recurrent Model-Free RL | ✓ | $\phi_{Q^*}$ | $\phi_{Q^*}$ | N/A | N/A |
| PBL (Guo et al., 2020) | ✓ | $\phi_L$ | $\phi_{Q^*}$ + ZP | $\ell_2$ | detached |
| AIS (Subramanian et al., 2022) | ✓ | $\phi_L, \phi_O$ | RP + ZP or OP | $\ell_2$, FKL | detached |
| Belief-Based Methods | ✓ | $\phi_O$ | RP + ZP + OR | FKL | online |
| Causal States (Zhang et al., 2019) | ✓ | $\phi_O$ | RP + OP | N/A | N/A |
| Minimalist $\phi_L$ (this work) | ✓ | $\phi_L$ | $\phi_{Q^*}$ + ZP | $\ell_2$, KL | stop-grad |

This result holds significance, as it suggests that end-to-end approaches to learning $\phi_{Q^*}$ + ZP (OP) have similar theoretical justification as classic phased approaches.

## 3.2 WHICH REPRESENTATIONS DO PRIOR METHODS LEARN?

With the unified view of state and history representations, we can categorize prior works based on the conditions satisfied by the *optimal* encoders of their value functions. Table 1 shows representative examples. The unified view enables us to draw interesting connections between prior works, even though they may differ in RL or planning algorithms and the encoder objectives. Here we highlight some important connections and provide a more detailed discussion of all prior works in Sec. C.

To begin with, it is important to recognize that classic model-based RL actually learns $\phi_{Q^*}$ in value function. Model-based RL trains a policy and value by rolling out on the learned model. However, the policy and value do not share representations with the model (Sutton, 1990; Sutton et al., 2012; Chua et al., 2018; Kaiser et al., 2019; Janner et al., 2019), or learn their representations from maximizing returns (Tamar et al., 2016; Oh et al., 2017; Silver et al., 2017). Secondly, as shown in Table 1, there is a wealth of prior work on approximating $\phi_L$, stemming from different perspectives. These include bisimulation (Gelada et al., 2019), information states (Subramanian et al., 2022), variational inference (Eysenbach et al., 2021; Ghugare et al., 2022), successor features (Barreto et al., 2017; Lehnert & Littman, 2020), and self-supervised learning (Schwarzer et al., 2020; Guo et al., 2020). The primary differences between these approaches lie in their selection of (1) architecture (whether learning RP, $\phi_{Q^*}$, or both), (2) ZP objectives (such as $\ell_2$, cosine, forward or reverse KL, as discussed in Sec. 4.1), and (3) ZP targets for optimization (including online, detached, EMA, as detailed in Sec. 4.2). Finally, observation-predictive representations are typically studied in POMDPs, where they are known as belief states (Kaelbling et al., 1998) and predictive state representations (Littman et al., 2001).

## 4 ON LEARNING SELF-PREDICTIVE REPRESENTATIONS IN RL

The implication graph (Fig. 1) establishes the theoretical connections among various representations in RL, yet it does not address the core learning problems. This section aims to give some theoretical answers to **how** to learn self-predictive representations. While self-predictive representation holds promise, it poses significant learning challenges compared to grounded model-free and observation-predictive representations. The bootstrapping effect, where $\phi$ appears in both sides of ZP (since $z'$ also relies on $\phi(h')$), contributes to this difficulty. We present detailed analyses of the objectives in Sec. 4.1 and optimization in Sec. 4.2, with proofs deferred to Sec. B. Building on these analyses, we propose a simple representation learning algorithm for $\phi_L$ in RL in Sec. 4.3.

### 4.1 ARE PRACTICAL ZP OBJECTIVES BIASED?

Thm. 2 suggests that we can learn $\phi_L$ by simply training an auxiliary task of ZP on a model-free agent. Prior works have proposed several auxiliary losses, summarized in Table 1's ZP loss column. Formally, we parametrize an encoder with $f_\phi : \mathcal{H}_t \to \mathcal{Z}$ (deterministic case) or $f_\phi : \mathcal{H}_t \to \Delta(\mathcal{Z})$ (probabilistic case)[7]. The latent transition function $P_z(z' \mid z, a)$ is parameterized by $g_\theta : \mathcal{Z} \times \mathcal{A} \to \mathcal{Z}$ (deterministic case) or $g_\theta : \mathcal{Z} \times \mathcal{A} \to \Delta(\mathcal{Z})$ (probabilistic case). We use $\mathbb{P}_\phi(z \mid h)$ and $\mathbb{P}_\theta(z' \mid z, a)$ to represent the encoder and latent transition, respectively. The self-predictive metric for ZP is *ideally*:

$$\mathcal{L}_{ZP,\mathbb{D}}(\phi, \theta; h, a) := \mathbb{E}_{z \sim \mathbb{P}_\phi(|h)}[\mathbb{D}(\mathbb{P}_\theta(z' \mid z, a) \,||\, \mathbb{P}_\phi(z' \mid h, a))], \tag{1}$$

where $\mathbb{P}_\phi(z' \mid h, a) = \mathbb{E}_{o' \sim P(\cdot|h,a)}[\mathbb{P}_\phi(z' \mid h')]$. $\mathbb{D}(\cdot \,||\, \cdot) \in \mathbb{R}_{\geq 0}$ compares two distributions. When Eq. 1 reaches minimum, then for any $z \sim \mathbb{P}_\phi(|\ h)$, the ZP condition is satisfied.

When designing *practical* ZP loss, prior works are mainly divided into deterministic $\ell_2$ approach (Gelada et al., 2019; Schwarzer et al., 2020; Tomar et al., 2021; Hansen et al., 2022; Ye et al., 2021)[8] or probabilistic f-divergence approach (Zhang et al., 2020; Ghugare et al., 2022; Hafner et al., 2019) that includes forward and reverse KL divergences (in short, FKL and RKL):

$$J_\ell(\phi, \theta, \widetilde{\phi}; h, a) := \mathbb{E}_{o' \sim P(|h,a)}\Big[\|g_\theta(f_\phi(h), a) - f_{\widetilde{\phi}}(h')\|_2^2\Big], \tag{2}$$

$$J_{D_f}(\phi, \theta, \widetilde{\phi}; h, a) := \mathbb{E}_{z \sim \mathbb{P}_\phi(|h), o' \sim P(|h,a)}\Big[D_f\Big(\mathbb{P}_{\widetilde{\phi}}(z' \mid h') \,||\, \mathbb{P}_\theta(z' \mid z, a)\Big)\Big], \tag{3}$$

where $\widetilde{\phi}$, called ZP **target**, can be **online** (exact $\phi$ that allows gradient backpropagation), or the **stop-gradient** version $\overline{\phi}$ (detached from the computation graph and using a copy or exponential moving average (EMA) of $\phi$). The update rule is $\overline{\phi} \leftarrow \tau\overline{\phi} + (1 - \tau)\phi$, with $\tau = 0$ for **detached** and $\tau \in (0, 1)$ for generic **EMA**. We summarize the choices of ZP targets in one column of Table 1.

We first investigate the relationship between the ideal objective Eq. 1 and practical objectives Eq. 2 and Eq. 3 to better understand their implications.

**Proposition 1** (The practical $\ell_2$ objective Eq. 2 is an **upper bound** of the ideal objective Eq. 1 $\mathcal{L}_{ZP,\ell}(\phi, \theta; h, a)$ that targets EZP condition. The equality holds in deterministic environments.)**.**

**Proposition 2** (The practical f-divergence objective Eq. 3 is an **upper bound** of the ideal objective Eq. 1 $\mathcal{L}_{ZP,D_f}(\phi, \theta; h, a)$ that targets ZP condition. The equality holds in deterministic environments.)**.**

These propositions show that environment stochasticity ($P(o' \mid h, a)$) affects both the practical $\ell_2$ and f-divergence objectives. While unbiased in *deterministic* tasks to learn the ZP condition[9], they are problematic in *stochastic* tasks (*e.g.* with data augmentation or noisy distractors) due to double sampling issue (Baird, 1995), as the ideal objective Eq. 1 cannot be used (see Sec. B for discussion).

### 4.2 WHY DO STOP-GRADIENTS WORK FOR ZP OPTIMIZATION?

In this subsection, we further discuss optimizing the practical ZP objective Eq. 2. Specifically, we aim to justify that stop-gradient (detached or EMA) ZP targets, widely used in practice (Schwarzer et al., 2020; Zhang et al., 2020; Ghugare et al., 2022), play an important role in optimization. We find that they may lead to EZP condition in stochastic environments (Prop. 3) and can avoid representational collapse under some linear assumptions (Thm. 3). Meanwhile, online ZP targets lack these properties.

Before introducing the results, we want to clarify the discrepancy between learning ZP and the well-known TD learning (Sutton, 1988). At first glance, Eq. 2 (stop-gradient version) is reminiscent of mean-squared TD error, both having the bootstrapping structures. However, Eq. 2 has an extra challenge due to the missing reward, leading to a trivial solution of constant representation, known as complete representational collapse (Jing et al., 2021). Moreover, ZP requires distribution matching, while the Bellman equations that TD learning aims to optimize only require matching expectations.

**Proposition 3** (The $\ell_2$ objective Eq. 2 with *stop gradients* ($J_\ell(\phi, \theta, \overline{\phi}; h, a)$) ensures stationary points that satisfy EZP, but the $\ell_2$ objective with *online* targets lacks this guarantee.)**.**

---

[7]Despite being a special case of probabilistic encoders, deterministic encoders deserve distinct discussion because they can be optimal in POMDPs and have been frequently used in prior works. In addition, it should be noted that a probabilistic one may help as it smooths the objective.

[8]Cosine distance is an $\ell_2$ distance on the normalized vector space $\mathcal{Z} = \{ z \in \mathbb{R}^d \mid \|z\|_2 = 1\}$.

[9]The EZP condition is equivalent to the ZP condition in deterministic tasks.

---

**Algorithm 1 Minimalist** $\phi_L$: learning self-predictive representations in RL

---

**Require:** Encoder $f_\phi : \mathcal{H}_t \to \mathcal{Z}$, Actor $\pi_\nu : \mathcal{Z} \to \mathcal{A}$, Critic $Q_\omega : \mathcal{Z} \times \mathcal{A} \to \mathbb{R}$, Latent Transition
    Model $g_\theta : \mathcal{Z} \times \mathcal{A} \to \mathcal{Z}$. Learning Rate $\alpha > 0$ and Loss Coefficient $\lambda > 0$.
1: **procedure** UPDATE($h, a, o', r$)
2:     Compute any model-free RL loss $\mathcal{L}_{\text{RL}}$ (based on DDPG (Lillicrap et al., 2016) here) let
    $Q^{\text{tar}}(h', r) := r + \gamma Q_{\overline{\omega}}(f_{\overline{\phi}}(h'), \pi_{\overline{\nu}}(f_{\overline{\phi}}(h')))$,

$$\mathcal{L}_{\text{RL}}(\phi, \omega, \nu; h', r) = (Q_\omega(f_\phi(h), a) - Q^{\text{tar}}(h', r))^2 - Q_{\overline{\omega}}(f_{\overline{\phi}}(h), \pi_\nu(f_\phi(h))). \quad (4)$$

3:     Compute the auxiliary ZP loss $\mathcal{L}_{\text{aux}}(\phi, \theta; h') = \|g_\theta(f_\phi(h), a) - f_{\overline{\phi}}(h')\|_2^2$.
4:     Optimize all parameters using the sum of losses:

$$[\phi, \theta, \nu, \omega] \leftarrow [\phi, \theta, \nu, \omega] - \alpha \nabla(\mathcal{L}_{\text{RL}}(\phi, \omega, \nu; h', r) + \lambda \mathcal{L}_{\text{aux}}(\phi, \theta; h')). \quad (5)$$

---

Prop. 3 suggests the adoption of stop-gradient targets in Eq. 2 to preserve the stationary points of EZP in both deterministic and stochastic tasks.

**Theorem 3 (Stop-gradient provably avoids representational collapse in linear models).** *Assume a linear encoder $f_\phi(h) := \phi^\top h_{-k:} \in \mathbb{R}^d$ with parameters $\phi \in \mathbb{R}^{k(|\mathcal{O}|+|\mathcal{A}|) \times d}$, which always operates on $h_{-k:}$, a recent-k truncation of history $h$. Assume a linear deterministic latent transition $g_\theta(z, a) := \theta_z^\top z + \theta_a^\top a \in \mathbb{R}^d$ with parameters $\theta_z \in \mathbb{R}^{d \times d}$ and $\theta_a \in \mathbb{R}^{|\mathcal{A}| \times d}$. If we train $\phi, \theta$ using the stop-gradient $\ell_2$ objective $\mathbb{E}_{h,a}\left[ J_\ell(\phi, \theta, \overline{\phi}; h, a) \right]$ without RL loss, and $\theta$ relies on $\phi$ by reaching the stationary point with $\nabla_\theta \mathbb{E}_{h,a}\left[ J_\ell(\phi, \theta, \overline{\phi}; h, a) \right] = 0$, then the matrix multiplication $\phi^\top \phi$ will retain its initial value over continuous-time training dynamics.*

Thm. 3 extends the results of (Tang et al., 2022, Theorem 1) to action-dependent latent transition, POMDP, and EMA settings. This theorem also implies that $\phi$ will keep full-rank during training if the initialized $\phi$ is full-rank[10].

Similar to Tang et al. (2022), we illustrate our theoretical contribution by examining the behavior of the learned encoder over time when starting from a random orthogonal initialization. We extend these results by considering both the MDP and the POMDP setting and consider two classical domains, mountain car (Moore, 1990) (MDP) and load-unload (Meuleau et al., 2013) (POMDP), where we fit an encoder $\phi$ with a

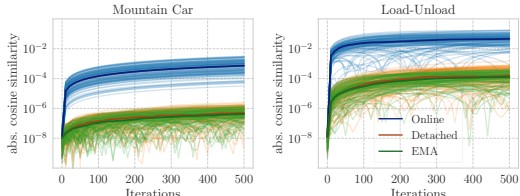

Figure 2: The absolute normalized inner product of the two column vectors in the learned encoder when using online, detached, or EMA ZP target in an MDP *(left)* and a POMDP *(right)*. We plot the results for 100 different seeds, which controls the rollouts used to sample transition and the initialization of the representation. The bold lines represent the median of the seeds.

latent state dimension of 2. Fig. 2 shows the orthogonality-preserving effect of the stop-gradient by comparing the cosine similarity between columns of the learned $\phi$. As expected by Thm. 3, we see this similarity stay several orders of magnitude smaller when using stop-gradient (detached or EMA) compared to the online case. Note that although our theory discusses the continuous-time dynamics, we can approximate them with gradient steps with a small learning rate, as was done for these results.

### 4.3 A MINIMALIST RL ALGORITHM FOR LEARNING SELF-PREDICTIVE REPRESENTATIONS

Our theory leads to a straightforward RL algorithm that can target $\phi_L$. Essentially, it integrates a single auxiliary task into any model-free RL algorithm (*e.g.*, DDPG (Lillicrap et al., 2016) and R2D2 (Kapturowski et al., 2018)), as indicated by Thm. 2. Algo. 1 provides the pseudocode for the update rule of all parameters in our algorithm given a tuple of transition data, with PyTorch code included in Appendix. The $\ell_2$ ZP loss can be replaced with KL objective with probabilistic encoder and latent model, especially in stochastic environments, as suggested by Prop. 2. The $f_{\overline{\phi}}(h')$ in ZP loss stops the gradient from the encoder, following Sec. 4.2. The actor loss $-Q_{\overline{\omega}}(f_{\overline{\phi}}(h), \pi_\nu(f_\phi(h)))$ freezes the parameters of the critic and encoder, suggested by recent work on memory-based RL (Ni et al., 2023).

Our algorithm greatly simplifies prior self-predictive methods and enables a fair comparison spanning from model-free to observation-predictive representation learning. It is characterized as **minimalist** by removing reward learning, planning, multi-step predictions, projections, and metric learning. It is also **novel** by being the first to learn self-predictive representations end-to-end in POMDP literature.

Our algorithm also bridges model-free and observation-predictive representation learning. We derive learning $\phi_{Q^*}$ by setting the coefficient $\lambda$ to 0, and learning $\phi_O$ by replacing ZP loss with OP loss like OFENet (Ota et al., 2020). As a by-product, by comparing $\{\phi_{Q^*}, \phi_L, \phi_O\}$ derived from

---

[10]This is due to the fact that $\text{rank}(A^\top A) = \text{rank}(A)$ for any real-valued matrix $A$.

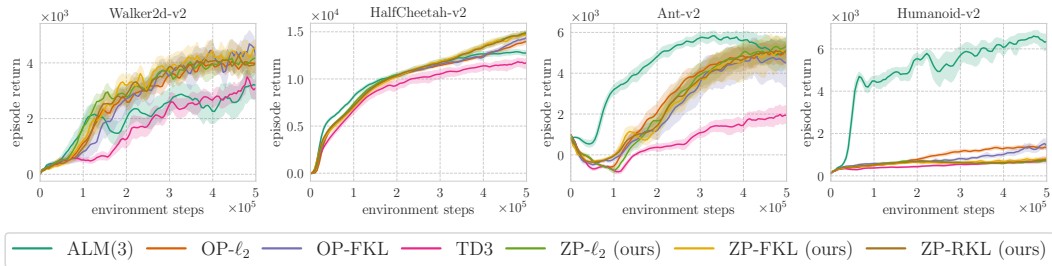

Figure 3: **Decoupling representation learning from policy optimization using our algorithm based on ALM(3)** (Ghugare et al., 2022). Comparison between $\phi_{Q*}$ (TD3), $\phi_L$ (our algorithm (ZP-$\ell_2$, ZP-FKL, ZP-RKL) and ALM(3)), $\phi_O$ (OP-$\ell_2$, OP-FKL), in the standard MuJoCo benchmark for 500k steps, averaged over 12 seeds. The observation dimension increases from left figure to right figure (17, 17, 111, 376).

our algorithm, we can **disentangle** representation learning from policy optimization because all representations are learned by the same RL algorithm. This is rarely seen in prior works, as learning $\phi_O$ or $\phi_L$ typically involves planning, while model-free RL does not. Such disentanglement allows us to examine the sample efficiency benefits derived purely from representation learning.

## 5 EXPERIMENTS

We conduct experiments to compare RL agents learning the three representations $\{\phi_{Q*}, \phi_L, \phi_O\}$, respectively. To decouple representation learning from policy optimization, we follow our minimalist algorithm (Algo. 1) to learn $\phi_L$, and instantiate $\phi_{Q*}$ and $\phi_O$ by setting $\lambda = 0$ and replacing ZP loss with OP loss, as we discuss in Sec. 4.3. We evaluate the algorithms in standard MDPs, distracting MDPs[11], and sparse-reward POMDPs. The experimental details are shown in Sec. E. Through the subsequent experiments, we aim to validate five hypotheses based on our theoretical insights in Sec. 3 and Sec. 4, with their motivation shown in Sec. D.1.

- **Sample efficiency hypothesis**: do the extra OP and ZP signals help $\phi_O$ and $\phi_L$ have better sample-efficiency than $\phi_{Q*}$ in standard MDPs (Sec. 5.1) and *especially* in sparse-reward tasks (Sec. 5.3)?
- **Distraction hypothesis** (Sec. 5.2): since learning $\phi_O$ may struggle with predicting distracting observations, is it less sample-efficient than learning $\phi_L$?
- **End-to-end hypothesis** (Sec. 5.3): as predicted by Thm. 2, is training an encoder end-to-end with an auxiliary task of ZP (OP) comparable to the phased training with RP and ZP (OP)?
- **ZP objective hypothesis** (Sec. 5.1, Sec. 5.2): as predicted by Prop. 1 and Prop. 2, is using $\ell_2$ loss as ZP objective similar to KL loss in deterministic tasks, but not necessarily stochastic tasks?
- **ZP stop-gradient hypothesis** (Sec. 5.1, Sec. 5.3): as predicted by Thm. 3, does stop-gradient on ZP targets mitigate representational collapse compared to online ZP targets?

### 5.1 STATE REPRESENTATION LEARNING IN STANDARD MDPS

We first evaluate our algorithm in the relatively low-dimensional MuJoCo benchmark (Todorov et al., 2012). We implement it by simplifying ALM(3) (Ghugare et al., 2022), a state-of-the-art algorithm on MuJoCo. ALM(3) aims to learn $\phi_L$ end-to-end with EMA ZP target and reverse KL, shown by Thm. 2. ALM(3) requires a reward model in the encoder objective and optimizes the actor via SVG (Heess et al., 2015) with planning for 3 steps. Following our Algo. 1, we remove the reward model and multi-step predictions;

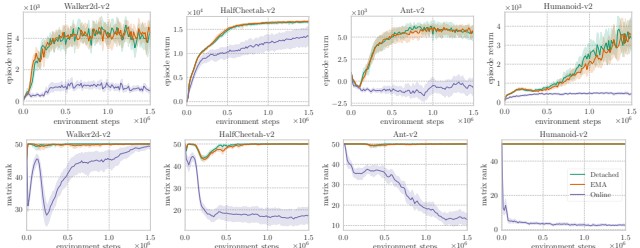

Figure 4: **Representation collapse with online targets.** On four benchmark tasks, we observe that using the online ZP target in $\ell_2$ objectives results in lower returns *(top)* and low-rank representations *(bottom)*. In line with our theory, using a detached or EMA ZP target mitigates the representational collapse and yields higher returns.

instead, we train the encoder using our loss (Eq. 2 or Eq. 3) and the actor-critic is conditioned on representations and trained using model-free TD3 (Fujimoto et al., 2018), while keeping the other hyperparameters the same. These simplifications result in a 50% faster speed than ALM(3). Please see Sec. E.2 for a detailed comparison between our algorithm and ALM(3).

**Validation of sample efficiency and ZP objective hypotheses.** Fig. 3 shows that our minimalist $\phi_L$ using EMA ZP targets can attain similar (in Ant) or *even better* (in HalfCheetah and Walker2d)

---

[11]Distracting MDPs refers to MDPs with distracting observations irrelevant to optimal control in this work.

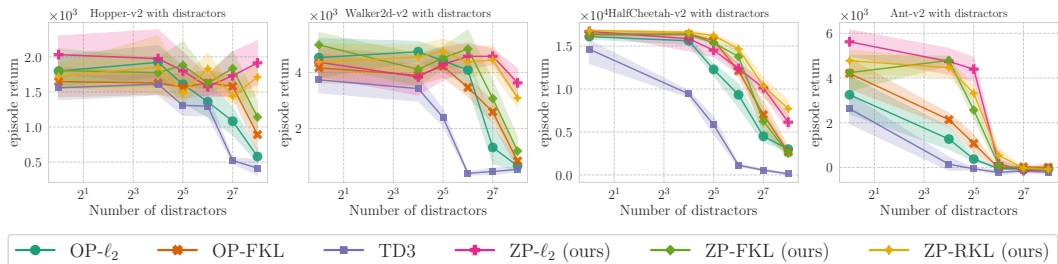

Figure 5: **Self-predictive representations are more robust.** Comparison between $\phi_{Q^*}$ (TD3), $\phi_L$ (ZP-$\ell_2$, ZP-FKL, ZP-RKL) using our algorithm, $\phi_O$ (OP-$\ell_2$, OP-FKL) in the **distracting** MuJoCo benchmark, varying the distractor dimension from $2^4$ to $2^8$, averaged over 12 seeds. The y-axis is final performance at 1.5M steps.

sample efficiency at 500k steps compared to ALM(3), and greatly outperforms $\phi_{Q^*}$, across the entire benchmark except for Humanoid task. This suggests that the primary advantage ALM(3) brings to model-free RL in these MuJoCo tasks, lies in state representation rather than policy optimization. This benefit could be further enhanced by streamlining the algorithmic design. In Humanoid task, ALM(3)'s superior performance is likely due to SVG policy optimization's use of first-order gradient information from latent dynamics, which is particularly beneficial in high-dimensional tasks like this. In line with Prop. 1 and Prop. 2, different ZP objectives ($\ell_2$, FKL, RKL) perform similarly on these tasks, which are nearly deterministic. Thus, these results support our sample efficiency and ZP objective hypotheses.

**Validation of ZP stop-gradient hypothesis.** We observe a significant performance degradation when switching from the stop-gradient ZP targets to *online* ones in all MuJoCo tasks for all ZP objectives ($\ell_2$, FKL, RKL). Fig. 4 (top) shows the results for the $\ell_2$ objective Eq. 2. To estimate the rank of the associated linearized operator for the MLP encoder, we compute the matrix rank of latent states given a batch of inputs[12], where the batch size is 512 and the latent state dimension is 50. The MLP encoders are orthogonally initialized (Saxe et al., 2013) with a full rank of 50. Fig. 4 (bottom) shows that the estimated rank of $\ell_2$ objective with online targets collapses from full rank to low rank, a phenomenon known as *dimensional collapse* in self-supervised learning (Jing et al., 2021). Interestingly, the high dimensionality of a task worsens the rank collapse by comparing from left to right figures. In contrast, stop-gradient targets suffer less from rank collapse, in support to our ZP stop-gradient hypothesis. For KL objectives, switching to online targets also decreases the rank, though less severely (see Fig. 11).

## 5.2 STATE REPRESENTATION LEARNING IN DISTRACTING MDPS

We then evaluate the robustness of representation learning by augmenting states with distractor dimensions in the MuJoCo benchmark. The distractors are i.i.d. standard isotropic Gaussians, varying the number of dimensions from $2^4$ to $2^8$, following the practice in Nikishin et al. (2022). The distracting task has the same optimal return as the original one and can be challenging to RL algorithms, even if model-based RL could perfectly model Gaussians. We use the same code in standard MDPs. Fig. 5 shows the final averaged returns of each algorithm (variant) in the distracting MuJoCo benchmark.

**Validation of distraction hypothesis.** By comparing ZP with OP objectives, with higher-dimensional distractors, learning $\phi_O$ degrades much faster than learning $\phi_L$, verifying our distraction hypothesis. Surprisingly, model-free RL ($\phi_{Q^*}$) performs worse than $\phi_O$, as $\phi_{Q^*}$ does not need to predict the distractors. However, we still observe a severe degradation in Ant for all methods when there are 128 distractors, which we will study in the future work.

**Extending ZP objective hypothesis to stochastic tasks.** In stochastic tasks, Prop. 1 and Prop. 2 tell us about the (strict) upper bounds for learning EZP and ZP conditions with the practical $\ell_2$ and KL objectives, respectively. Yet, they do not indicate which objective is better for stochastic tasks based on these bounds. In fact, we observe that both $\ell_2$ and reverse KL perform better than forward KL in the distracting tasks, possibly because the entropy term in reverse KL smooths the training objective, and $\ell_2$ objective simplifies the learning.

## 5.3 HISTORY REPRESENTATION LEARNING IN SPARSE-REWARD POMDPS

Finally, we perform an extensive empirical study on 20 MiniGrid tasks (Chevalier-Boisvert et al., 2018). These tasks, featuring partial observability and sparse rewards, serve as a rigorous test-bed for *history* representation learning. The rewards are only non-zero upon successful task completion. Episode returns are between 0 and 1, with higher returns indicating faster completion. Each task has an observation space of $7 \times 7 \times 3 = 147$ dimensions and a discrete action space of 7 options. Our

---

[12]The rank of an $m \times n$ matrix $A$ is the dimension of the image of mapping $f : \mathbb{R}^n \to \mathbb{R}^m$, where $f(x) = Ax$.

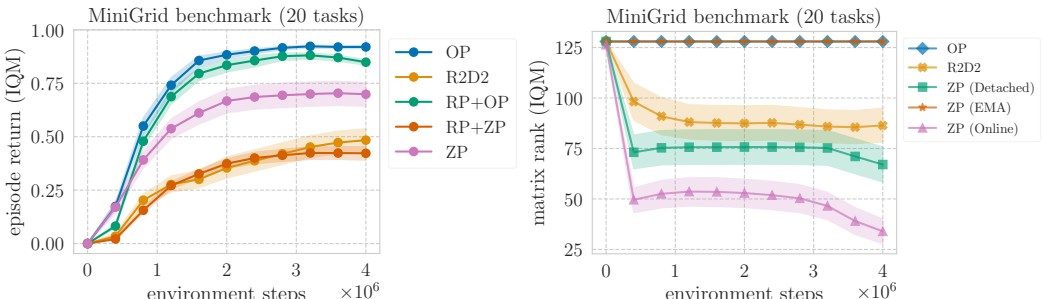

Figure 6: **End-to-end learned self-predictive and observation-predictive representations stand out in sparse-reward tasks.** We show the interquartile mean (IQM) (Agarwal et al., 2021) across **20 MiniGrid tasks**, computed over 9 seeds per task with 95% stratified bootstrap confidence intervals. The individual task plots are shown in Appendix Fig. 14 and Fig. 15. **Left**: comparison of episode returns between $\phi_{Q*}$ (R2D2), $\phi_L$ (ZP, RP + ZP), $\phi_O$ (OP, RP + OP). RP + ZP and RP + OP methods are phased, while ZP and OP methods are end-to-end. **Right**: comparison of estimated matrix rank between ZP targets (online, detached, EMA), R2D2, and OP. The maximal achievable rank is 128.

model-free baseline is R2D2 (Kapturowski et al., 2018), composed of double Q-learning (Van Hasselt et al., 2016) with LSTMs (Hochreiter & Schmidhuber, 1997) as history encoders. We implement it based on recent work (Seyedsalehi et al., 2023). R2D2 uses stored LSTM hidden states for initialization, a 50-step burn-in period, and a 10-step history rollout during training. This result in a high dimensionality of $(50 + 10) \times (147 + 7) = 9240$ on histories.

Based on R2D2, we implement our algorithm by adding an auxiliary task of ZP (or OP) under $\ell_2$ objectives, which is sufficient for solving these deterministic tasks (see Prop. 1). To compare the phased training (Seyedsalehi et al., 2023), we further implement methods (RP + OP, RP + ZP) that also predict rewards and freeze the encoders during Q-learning. Due to the space limit, we show the aggregated plots in Fig. 6 and defer the individual task plots to Sec. G.

**Validation of sample efficiency hypothesis.** By examining the aggregated learning curves of end-to-end methods in Fig. 6 left, we find that minimalist $\phi_L$ (ZP) significantly outperforms $\phi_{Q*}$ (R2D2) on average. It, however, fall shorts of $\phi_O$ (OP), which aligns with our expectations given that MiniGrid tasks are deterministic with medium-dimensional clean observations. The enhanced performance of $\phi_L$ and $\phi_O$ over $\phi_{Q*}$ provides empirical validation of our hypothesis in sparse-reward tasks.

**Validation of end-to-end hypothesis.** By comparing the aggregated learning curves between end-to-end methods and phased methods (*i.e.*, OP vs RP + OP, and ZP vs RP + ZP) in Fig. 6 left, we observe that end-to-end training (OP or ZP) yields equal or superior sample efficiency relative to phased training (RP + OP or RP + ZP) when learning observation-predictive or self-predictive representations. This supports our hypothesis, and is particularly noticeable when end-to-end learning $\phi_L$ (ZP) markedly excels over its phased learning counterpart (RP + ZP).

**Validation of ZP stop-gradient hypothesis.** We extend our rank analysis from MuJoCo to MiniGrid using the same estimation metric. As depicted in Fig. 6 right, our finding averaged across the benchmark indicates that EMA ZP targets are able to preserve their rank, while both detached and online ZP targets degrade the rank during training. Notably, while our theory does not distinguish detached and EMA targets, the observed lower rank of online targets relative to both detached and EMA targets is in line with our hypothesis. Finally, without any auxiliary task such as ZP and OP, R2D2 degrades the rank, which is predictable in the sparse-reward setting (Lyle et al., 2021).

## 6    DISCUSSION

**Recommendations.** Based on our theoretical and empirical results, we suggest the following preliminary guidance to RL practitioners:

1. **Analyze your task first.** For example, in noisy or distracting tasks, consider using self-predictive representations. In sparse-reward tasks, consider using observation-predictive representations. In deterministic tasks, choose the deterministic $\ell_2$ objectives for representation learning.

2. **Use our minimalist algorithm as your baseline.** Our algorithm allows for an independent evaluation of representation learning and policy optimization effects. Start with end-to-end learning and model-free RL for policy optimization.

3. **Implementation tips.** For our minimalist algorithm, we recommend adopting the $\ell_2$ objective with EMA ZP targets first. When tackling POMDPs, start with recurrent networks as the encoder.

Please refer to Sec. D.2 for our discussion on limitations and conclusion.

ACKNOWLEDGEMENTS AND DISCLOSURE OF FUNDING

We thank Pierluca D'Oro and Zhixuan Lin for their technical help. We thank Amit Sinha, David Kanaa, David Yu-Tung Hui, Dinghuai Zhang, Doina Precup, Léo Gagnon, Pablo Samuel Castro, Raj Ghugare, Shreyas Chaudhari, and Ziyan Luo for the constructive discussion. This work was enabled by the computational resources provided by the Calcul Québec (www.calculquebec.ca) and the Digital Research Alliance of Canada (https://alliancecan.ca/), with material support from NVIDIA Corporation. This work was funded by IBM Research and Google DeepMind.

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
