### A.1   NOTATION

Table 2 shows the glossary used in this paper.

### A.2   ADDITIONAL BACKGROUND

**Remark on the latent state distribution.**   In this paper, we assume the latent space $\mathcal{Z}$ as a pre-specified Banach space, which is a complete normed vector space. We further assume any latent state distribution defined on $\mathcal{Z}$ has a finite expectation. To avoid a measure-theoretic treatment, we assume that Z is discrete-valued in our proof analysis. The proof arguments are easily generalized to the case when $\mathcal{Z}$ lies in a Banach space using standard arguments.

**Remark on the existence of optimal value and policy in POMDPs.**   In an MDP, it is well-known that there exists a unique optimal value function following the Bellman equation, which induces an optimal deterministic policy (Puterman, 1994). In POMDPs, the result is complicated. For a *finite-horizon* POMDP, one can construct a finite-dimensional state space by stacking all previous observations and actions to convert a POMDP into an MDP, thus the MDP result can be directly applied. For an *infinite-horizon* POMDP, Subramanian et al. (2022, Theorem 25) shows that the unique optimal value function exists when the POMDP has a time-invariant finite-dimensional information

Table 2: **Glossary of notations** used in this paper.

| Notation | Text description | Math description |
|---|---|---|
| $\gamma$ | Discount factor | $\gamma \in [0, 1]$ |
| $T$ | Horizon | $T \in \mathbb{N} \cup \{+\infty\}$ |
| $s_t$ | State at step $t$ | $s \in \mathcal{S}$ |
| $o_t$ | Observation at step $t$ | $o \in \mathcal{O}$ |
| $a_t$ | Action at step $t$ | $a \in \mathcal{A}$ |
| $r_t$ | Reward at step $t$ | $r \in \mathbb{R}$ |
| $h_t$ | History at step $t$ | $h_t = (h_{t-1}, a_{t-1}, o_t) \in \mathcal{H}_t, h_1 = o_1$ |
| $P(o_{t+1} \mid h_t, a_t)$ | Environment transition | |
| $R(h_t, a_t)$ | Environment reward function | $R : \mathcal{H}_t \times \mathcal{A} \to \Delta(\mathbb{R})$ |
| $\pi(a_t \mid h_t)$ | Policy (actor) | |
| $\pi^*(h_t)$ | Optimal policy (actor) | |
| $Q^\pi(h_t, a_t)$ | Value (critic) | |
| $Q^*(h_t, a_t)$ | Optimal value (critic) | |
| $\phi$ | Encoder of history | $\phi : \mathcal{H}_t \to \mathcal{Z}$ |
| $z_t$ | Latent state at step $t$ | $z_t = \phi(h_t) \in \mathcal{Z}$ |
| $P_z(z_{t+1} \mid z_t, a_t)$ | Latent transition | |
| $R_z(z_t, a_t)$ | Latent reward function | |
| $\pi_z(a_t \mid z_t)$ | Latent policy (actor) | |
| $\pi_z^*(z_t)$ | Optimal latent policy (actor) | |
| $Q_z^{\pi_z}(z_t, a_t)$ | Latent value (critic) | |
| $Q_z^*(z_t, a_t)$ | Optimal latent value (critic) | |
| RP | Expected **R**eward **P**rediction | $\mathbb{E}[r_t \mid h_t, a_t] = R_z(\phi(h_t), a_t)$ |
| OR | **O**bservation **R**econstruction | $o_t = \psi_o(\phi(h_t))$ |
| OP | Next **O**bservation **P**rediction | $P(o_{t+1} \mid h_t, a_t) = P_o(o_{t+1} \mid \phi(h_t), a_t)$ |
| ZP | Next Latent State $z$ **P**rediction | $P(z_{t+1} \mid h_t, a_t) = P_z(z_{t+1} \mid \phi(h_t), a_t)$ |
| EZP | **E**xpected Next Latent State $z$ **P**rediction | $\mathbb{E}[z_{t+1} \mid h_t, a_t] = \mathbb{E}[z_{t+1} \mid \phi(h_t), a_t]$ |
| Rec | **Rec**urrent Encoder | $\phi(h_{t+1}) = \psi_z(\phi(h_t), a_t, o_{t+1})$ |
| ZM | **M**arkovian Latent Transition | $z_{t+1} \perp\!\!\!\perp z_{1:t-1}, a_{1:t-1} \mid \phi(h_t), a_t$ |
| $\phi_{\pi^*}$ | $\pi^*$-irrelevance abstraction | $\phi(h_1) = \phi(h_2) \implies \pi^*(h_1) = \pi^*(h_2)$ |
| $\phi_{Q^*}$ | $Q^*$-irrelevance abstraction | $\phi(h_1) = \phi(h_2) \implies Q^*(h_1, a) = Q^*(h_2, a)$ |
| $\phi_M$ | Markovian abstraction | RP + ZM |
| $\phi_L$ | Self-predictive abstraction | RP + ZP $\iff$ $\phi_{Q^*}$ + ZP |
| $\phi_O$ | Observation-predictive abstraction | RP + OP + Rec $\iff$ $\phi_{Q^*}$ + OP + Rec |

state, which is the case when the unobserved state space is finite. The POMDP experiments shown in Sec. 5.3 satisfy this assumption because they have a finite state space. For POMDPs with infinite-dimensional information states, the result remains unclear.

### A.2.1 ADDITIONAL ABSTRACTIONS AND FORMALIZING THE RELATIONSHIP

First, we present two additional abstractions not shown in the main paper, which are also used in prior work. Then we formalize Thm. 1 with Thm. 4 using the concept of granularity in relation.

$\pi^*$**-irrelevance abstraction.** An encoder $\phi_{\pi^*}$ yields a $\pi^*$-irrelevance abstraction (Li et al., 2006) if it contains the necessary information (a "sufficient statistics") for selecting return-maximizing actions. Formally, if $\phi_{\pi^*}(h_i) = \phi_{\pi^*}(h_j)$ for some $h_i, h_j \in \mathcal{H}_t$, then $\pi^*(h_i) = \pi^*(h_j)$. One way of obtaining a $\pi^*$-irrelevance abstraction is to learn an encoder $\phi$ end-to-end with a policy $\pi_z(a \mid \phi(h))$ by model-free RL (Sutton et al., 1999) such that $\pi_z^*(\phi_{\pi^*}(h)) = \pi^*(h), \forall h$.

**Markovian abstraction**. An encoder $\phi_M$ provides Markovian abstraction if it satisfies the expected reward condition RP and **Markovian latent transition (ZM)** condition: for any $z_k = \phi_M(h_k)$,

$$P(z_{t+1} \mid z_{1:t}, a_{1:t}) = P(z_{t+1} \mid z_t, a_t), \quad \forall z_{1:t+1}, a_{1:t}. \tag{ZM}$$

This extends Markovian abstraction (Allen et al., 2021) in MDPs to POMDPs.

**Granularity in relation**. In MDPs, it is well-known that state representations form a hierarchical structure (Li et al., 2006, Theorem 2), but this idea had not been extended to the POMDP case. We do so here by defining an equivalent concept of "granularity". We say that an encoder $\phi_A$ is finer than or equal to another encoder $\phi_B$, denoted as $\phi_A \succeq \phi_B$, if and only if for any histories $h_i, h_j \in \mathcal{H}_t$, $\phi_A(h_i) = \phi_A(h_j)$ implies $\phi_B(h_i) = \phi_B(h_j)$. The relation $\succeq$ is a partial ordering. Using this notion, we can show Thm. 4:

**Theorem 4** (**Granularity of state and history abstractions (the formal version of Thm. 1)**)**.**
$\phi_O \succeq \phi_L \succeq \phi_{Q^*} \succeq \phi_{\pi^*}$.

**Abstract MDP.** Given an encoder $\phi$, we can construct an abstract MDP (Li et al., 2006) $\mathcal{M}_\phi = (\mathcal{Z}, \mathcal{A}, P_z, R_z, \gamma, T)$ for a POMDP $\mathcal{M}_O$. The latent reward $R_z$ and latent transition $P_z$ are then given by: $R_z(z, a) = \int P(h \mid z)\mathbb{E}[r \mid h, a]dh$, $P_z(z' \mid z, a) = \int P(h \mid z)P(o' \mid h, a)\delta(z' = \phi(h'))dhdo'$, where $P(h \mid z) = 0$ for any $\phi(h) \neq z$ and is normalized to a distribution. The optimal latent (Markovian) value function $Q_z^*(z, a)$ statisfies $Q_z^*(z, a) = R_z(z, a) + \gamma\mathbb{E}_{z' \sim P_z(\mid z, a)}[\max_{a'} Q_z^*(z', a')]$, and the optimal latent policy $\pi_z^*(z) = \mathrm{argmax}_a Q_z^*(z, a)$. It is important to note that this definition focuses solely on the process by which the encoder induces a corresponding abstract MDP, without addressing the quality of the encoder itself.

### A.2.2 Alternative Definitions

In the main paper (Sec. 2), we present the concepts of self-predictive abstraction $\phi_L$ and observation-predictive abstraction $\phi_O$. In most prior works, these concepts were defined in an alternative way – using a pair of states (histories). In comparison, our definition is based on a pair of a state (history) and a latent state, which we believe is more comprehensible and help derive the auxiliary objectives.

For completeness, here we restate their definition, extended to POMDPs, and then show the equivalence between their and our definitions.

**Model-irrelevance abstraction (Li et al., 2006) (bisimulation relation (Givan et al., 2003))** $\Phi_L$**.** If for any two histories $h_i, h_j \in \mathcal{H}$ such that $\Phi_L(h_i) = \Phi_L(h_j)$, then

$$\mathbb{E}[r \mid h_i, a] = \mathbb{E}[r \mid h_j, a], \quad \forall a \in \mathcal{A}, \tag{6}$$

$$P(z' \mid h_i, a) = P(z' \mid h_j, a), \quad \forall a \in \mathcal{A}, z' \in \mathcal{Z}, \tag{7}$$

where $P(z' \mid h, a) = \int P(o' \mid h, a)\delta(z' = \Phi_L(h'))do'$. Here we extend the concept from MDPs (Li et al., 2006; Givan et al., 2003) into POMDPs. It is worth noting that while original concepts assume deterministic rewards or require reward distribution matching for stochastic rewards (Castro et al., 2009) in Eq. 6, the requirement can indeed be relaxed. As shown by Subramanian et al. (2022), it is sufficient to ensure expected reward matching to maintain optimal value functions. As such, we adopt this relaxed requirement of expectation matching in our concept.

**Proposition 4** ($\Phi_L$ **is equivalent to** $\phi_L$)**.**

*Proof.* It is easy to see that $\phi_L$ implies $\Phi_L$. If $\phi_L(h_i) = \phi_L(h_j)$, then by RP,

$$\mathbb{E}[r \mid h_i, a] = R_z(\phi_L(h_i), a) = R_z(\phi_L(h_j), a) = \mathbb{E}[r \mid h_j, a], \tag{8}$$

and by ZP,

$$P(z' \mid h_i, a) = P_z(z' \mid \phi_L(h_i), a) = P_z(z' \mid \phi_L(h_j), a) = P(z' \mid h_j, a). \tag{9}$$

Therefore, $\phi_L$ implies $\Phi_L$.

Now we want to show $\Phi_L$ implies $\phi_L$. We will use the following fact: for any two random variables $X, Y$ and a function $f$ that maps $Y$ into a random variable $Z$, we have $X \perp\!\!\!\perp f(Y) \mid Y$. This is equivalent to say:

$$P(X = x \mid Y = y) = P(X = x \mid Y = y, Z = f(y)), \quad \forall x, y. \tag{10}$$

A corollary is on the conditional expectation:

$$\mathbb{E}[X \mid Y = y] = \mathbb{E}[X \mid Y = y, Z = f(y)]. \tag{11}$$

First, to see RP condition: using the fact (Eq. 11),

$$\mathbb{E}[r \mid \mathcal{H} = h_i, \mathcal{A} = a] = \mathbb{E}[r \mid \mathcal{H} = h_i, \mathcal{A} = a, \mathcal{Z} = \phi(h_i)]. \tag{12}$$

By Eq. 6, we have for any $h_i, h_j$ such that $\phi(h_i) = \phi(h_j) \coloneqq z$,

$$\mathbb{E}[r \mid \mathcal{H} = h_i, \mathcal{A} = a, \mathcal{Z} = z] = \mathbb{E}[r \mid \mathcal{H} = h_j, \mathcal{A} = a, \mathcal{Z} = z]. \tag{13}$$

This exactly indicates RP condition: $\mathbb{E}[r \mid \mathcal{H} = h_i, \mathcal{A} = a]$ is a function of $\phi(h_i), a$.

Similar to the proof of showing RP, we can show ZP: using the fact (Eq. 10),

$$P(z' \mid \mathcal{H} = h_i, \mathcal{A} = a) = P(z' \mid \mathcal{H} = h_i, \mathcal{A} = a, \mathcal{Z} = \phi(h_i)). \tag{14}$$

By Eq. 7, we have for any $h_i, h_j$ such that $\phi(h_i) = \phi(h_j) \coloneqq z$,

$$P(z' \mid \mathcal{H} = h_i, \mathcal{A} = a, \mathcal{Z} = z) = P(z' \mid \mathcal{H} = h_j, \mathcal{A} = a, \mathcal{Z} = z). \tag{15}$$

This exactly indicates ZP condition: $P(z' \mid \mathcal{H} = h_i, \mathcal{A} = a)$ is a distribution conditioned on $\phi(h_i), a$.

$\square$

**Belief abstraction $\Phi_O$ (weak belief bisimulation relation (Castro et al., 2009)).** It satisfies Rec, and if for any two histories $h_i, h_j \in \mathcal{H}$ such that $\Phi_O(h_i) = \Phi_O(h_j)$, then

$$\mathbb{E}[r \mid h_i, a] = \mathbb{E}[r \mid h_j, a], \quad \forall a \in \mathcal{A}, \tag{16}$$

$$P(o' \mid h_i, a) = P(o' \mid h_j, a), \quad \forall a \in \mathcal{A}, o' \in \mathcal{O}. \tag{17}$$

This concept is known as a naive abstraction in MDPs (Jiang, 2018) and weak belief bisimulation relation in POMDPs (Castro et al., 2009). Similarly, prior concepts assume deterministic reward or distribution matching for stochastic rewards, while we relax it to expected reward matching.

**Proposition 5 ($\Phi_O$ is equivalent to $\phi_O$).**

*Proof.* The proof is almost the same as the proof of Prop. 4 by replacing $z'$ with $o'$. $\square$

### A.3 PROPOSITIONS AND PROOFS

With the additional background in Sec. A.2, we show the complete implication graph in Fig. 7 built on Fig. 1.

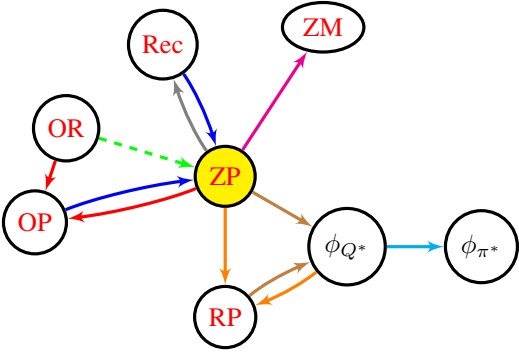

Figure 7: **The complete implication graph** showing the relations between the conditions on history representations. The source nodes of the edges with the same color together imply the target node. The dashed edge means it only applies to MDPs. As a quick reminder, RP: expected reward prediction, OP: next observation prediction, OR: observation reconstruction, ZP: next latent state prediction, Rec: recurrent encoder, ZM: Markovian latent transition. All the connections are discovered in this work, except for (1) OP + Rec implying ZP, (2) ZP + RP implying $\phi_{Q^*}$, (3) $\phi_{Q^*}$ implying $\phi_{\pi^*}$.

#### A.3.1 RESULTS RELATED TO ZP

**Lemma 1** (Functions of independent random variables are also independent). *If $X \perp\!\!\!\perp Y$, then for any functions $f, g$, we have $f(X) \perp\!\!\!\perp g(Y)$.*

*Proof.* This is a well-known result. Here is an elementary proof. Let $A, B$ be any two sets,

$$P(f(X) \in A, g(Y) \in B) = P(X \in f^{-1}(A), Y \in g^{-1}(B)) \tag{18}$$

$$\stackrel{X \perp\!\!\!\perp Y}{=} P(X \in f^{-1}(A))P(Y \in g^{-1}(B)) = P(f(X) \in A)P(g(Y) \in B). \tag{19}$$

$\square$

**Lemma 2.** *If $X \perp\!\!\!\perp Y \mid Z$, then for any function $f$, we have $X \perp\!\!\!\perp Y, f(Z) \mid Z$.*

*Proof.*

$$P(Y, f(Z) \mid X, Z) = P(f(Z) \mid X, Z)P(Y \mid X, Z, f(Z)) \tag{20}$$

$$= P(f(Z) \mid Z)P(Y \mid Z) = P(f(Z) \mid Z)P(Y \mid Z, f(Z)) = P(Y, f(Z) \mid Z). \tag{21}$$

$\square$

**Proposition 6 (ZP implies both ZM and Rec.).**

*Remark* 1. These are **new** results. ZP implying ZM means $\phi_L \succeq \phi_M$.

*Proof of Prop. 6 (ZP implies ZM).* Since ZP that $z_{t+1} \perp\!\!\!\perp h_t \mid z_t, a_t$, this implies $z_{t+1} \perp\!\!\!\perp f(h_t) \mid z_t, a_t$ for any transformation $f$ by Lemma 1. One special case of $f$ is that $f(h_t) = (z_{1:t}, a_{1:t-1})$, where $z_k = \phi(h_k)$, which is ZM. $\square$

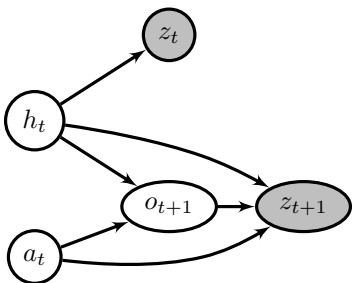

Figure 8: **The graphical model of the interaction between history encoder and the environment.**

*Proof of Prop. 6 (ZP implies Rec).* Let $\phi$ satisfy ZP, *i.e.* $z' \perp\!\!\!\perp h \mid \phi(h), a$. Then we can show that $z' \perp\!\!\!\perp h \mid \phi(h), a, o'$. This is because the graphical model ($(h, a) \to o'$ and $(h, a, o') \to z'$; see Fig. 8) does not have v-structure such that $(h, z') \to o'$, thus adding the variable $o'$ to conditionals preserves conditional independence, by the principle of d-separation (Pearl, 1988).

As $(a, o')$ also appears in the condition, we have $z' \perp\!\!\!\perp h' \mid \phi(h), a, o'$ by Lemma 2, which is the probabilistic form of Rec. $\square$

**Proposition 7 (OP and Rec imply ZP (Subramanian et al., 2022).).**

*Proof of Prop. 7 and Thm. 4 ($\phi_O \succeq \phi_L$).* We directly follow the proof in (Subramanian et al., 2022, Proposition 4). Let $\phi$ satisfy OP and Rec, then we will have ZP:

$$P(z' \mid h, a) = \int P(z', o' \mid h, a)do' = \int \delta(z' = \phi(h'))P(o' \mid h, a)do' \tag{22}$$

$$\stackrel{(Rec, OP)}{=} \int \delta(z' = \psi_z(\phi(h), a, o'))P_o(o' \mid \phi(h), a)do' \tag{23}$$

$$= \int P(z', o' \mid \phi(h), a)do' = P_z(z' \mid \phi(h), a). \tag{24}$$

$\square$

*Proof of Thm. 4* ($\phi_{Q^*} \succeq \phi_{\pi^*}$). If $\phi_{Q^*}(h_i) = \phi_{Q^*}(h_j)$, then $Q^*(h_i, a) = Q^*(h_j, a), \forall a$, and then taking argmax we get the optimal policy, $\pi^*(h_1) = \operatorname{argmax}_a Q^*(h_1, a) = \operatorname{argmax}_a Q^*(h_2, a) = \pi^*(h_2)$. $\qquad\square$

**Proposition 8** (**OR** and **ZP** imply **OP**).

*Proof.* Consider given $h, a$, for any $o'$,

$$P(o', \phi(h') \mid h, \phi(h), a) = P(\phi(h') \mid h, a)P(o' \mid \phi(h'), h, \phi(h), a) \tag{25}$$

$$\overset{(ZP, OR)}{=} P_z(\phi(h') \mid \phi(h), a)\delta(o' = \psi_o(\phi(h'))) \tag{26}$$

$$= P_z(\phi(h') \mid \phi(h), a)P(o' \mid \phi(h'), \phi(h), a) \tag{27}$$

$$= P(o', \phi(h') \mid \phi(h), a), \tag{28}$$

where Ln. 27 follows that the OR condition $o' \perp\!\!\!\perp h' \mid \phi(h')$ implies $o' \perp\!\!\!\perp \phi(h), a \mid \phi(h')$ by Lemma 1. Therefore, $o', \phi(h') \perp\!\!\!\perp h \mid \phi(h), a$. By Lemma 1, we have OP $o' \perp\!\!\!\perp h \mid \phi(h), a$.

$\qquad\square$

**Proposition 9** (**In MDPs, OR implies ZP and OP**).

*Proof.* Assume $\phi$ satifies OR in MDPs, *i.e.* there exists $\psi_s : \mathcal{Z} \to \mathcal{S}$ such that $\psi_s(\phi(s)) = s$. We want to show that $z', s \perp\!\!\!\perp s \mid \phi(s), a$ which implies ZP by Lemma 1. In fact,

$$P(z', s \mid s, \phi(s), a) = P(z', s \mid s, a) = P(z' \mid s, a)\delta(s = s) \tag{29}$$

$$P(z', s \mid \phi(s), a) = P(s \mid \phi(s), a)P(z' \mid s, \phi(s), a) \tag{30}$$

$$\overset{OR}{=} \delta(s = s)P(z' \mid s, a). \tag{31}$$

Similar proof to OP by replacing $z'$ with $s'$. $\qquad\square$

### A.3.2 Results Related to Multi-Step Conditions

Below are results on multi-step RP, ZP, and OP, and due to space limit, we do not show these connections in Fig. 7.

**Proposition 10** (**ZP is equivalent to multi-step ZP**). *For $k \in \mathbb{N}^+$, define $k$-step ZP as*

$$P(z_{t+k} \mid h_t, a_{t:t+k-1}) = P(z_{t+k} \mid \phi(h_t), a_{t:t+k-1}), \quad \forall h, a, z. \tag{32}$$

*Proof.* As ZP is 1-step ZP, thus multi-step ZP implies ZP. Now we show that ZP implies multi-step ZP.

$$P(z_{t+k} \mid h_t, a_{t:t+k-1}) = \int P(z_{t+1:t+k}, o_{t+1:t+k} \mid h_t, a_{t:t+k-1})do_{t+1:t+k}dz_{t+1:t+k-1} \tag{33}$$

$$= \int \prod_{i=1}^{k} \delta(z_{t+i} = \phi(h_{t+i}))P(o_{t+i} \mid h_{t+i-1}, a_{t+i-1})do_{t+1:t+k}dz_{t+1:t+k-1} \tag{34}$$

$$= \int \left( \int \delta(z_{t+k} = \phi(h_{t+k}))P(o_{t+k} \mid h_{t+k-1}, a_{t+k-1})do_{t+k} \right) \tag{35}$$

$$\prod_{i=1}^{k-1} \delta(z_{t+i} = \phi(h_{t+i}))P(o_{t+i} \mid h_{t+i-1}, a_{t+i-1})do_{t+1:t+k-1}dz_{t+1:t+k-1} \tag{36}$$

$$= \int P(z_{t+k} \mid h_{t+k-1}, a_{t+k-1}) \prod_{i=1}^{k-1} \delta(z_{t+i} = \phi(h_{t+i}))P(o_{t+i} \mid h_{t+i-1}, a_{t+i-1})do_{t+1:t+k-1}dz_{t+1:t+k-1} \tag{37}$$

$$\overset{ZP}{=} \int P(z_{t+k} \mid \phi(h_{t+k-1}), a_{t+k-1}) \prod_{i=1}^{k-1} \delta(z_{t+i} = \phi(h_{t+i}))P(o_{t+i} \mid h_{t+i-1}, a_{t+i-1})do_{t+1:t+k-1}dz_{t+1:t+k-1} \tag{38}$$

$$= \int P(z_{t+k} \mid z_{t+k-1}, a_{t+k-1}) \prod_{i=1}^{k-1} \delta(z_{t+i} = \phi(h_{t+i})) P(o_{t+i} \mid h_{t+i-1}, a_{t+i-1}) do_{t+1:t+k-1} dz_{t+1:t+k-1} \tag{39}$$

$$= \ldots \tag{40}$$

$$= \int \prod_{i=2}^{k} P(z_{t+i} \mid z_{t+i-1}, a_{t+i-1}) P(z_{t+1} \mid h_t, a_t) dz_{t+1:t+k-1} \tag{41}$$

$$\overset{ZP}{=} \int \prod_{i=2}^{k} P(z_{t+i} \mid z_{t+i-1}, a_{t+i-1}) P(z_{t+1} \mid \phi(h_t), a_t) dz_{t+1:t+k-1} \tag{42}$$

$$\overset{ZM}{=} \int P(z_{t+1:t+k} \mid \phi(h_t), a_{t:t+i-1}) dz_{t+1:t+k-1} \tag{43}$$

$$= P(z_{t+k} \mid \phi(h_t), a_{t:t+i-1}). \tag{44}$$

$\square$

**Proposition 11 (ZP and RP imply multi-step RP).** *For $k \in \mathbb{N}^+$, define $k$-step RP as*

$$\mathbb{E}[r_{t+k} \mid h_t, a_{t:t+k}] = \mathbb{E}[r_{t+k} \mid \phi(h_t), a_{t:t+k}], \quad \forall h, a \tag{45}$$

*Proof.*

$$\mathbb{E}[r_{t+k} \mid h_t, a_{t:t+k}] = \int P(o_{t+1:t+k} \mid h_t, a_{t:t+k-1}) \mathbb{E}[r_{t+k} \mid h_{t+k}, a_{t+k}] do_{t+1:t+k} \tag{46}$$

$$\overset{RP}{=} \int P(o_{t+1:t+k} \mid h_t, a_{t:t+k-1}) R_z(\phi(h_{t+k}), a_{t+k}) do_{t+1:t+k} \tag{47}$$

$$= \int P(o_{t+1:t+k} \mid h_t, a_{t:t+k-1}) \delta(z_{t+k} = \phi(h_{t+k})) R_z(z_{t+k}, a_{t+k}) do_{t+1:t+k} dz_{t+k} \tag{48}$$

$$= \int \left( \int P(o_{t+1:t+k} \mid h_t, a_{t:t+k-1}) \delta(z_{t+k} = \phi(h_{t+k})) do_{t+1:t+k} \right) R_z(z_{t+k}, a_{t+k}) dz_{t+k} \tag{49}$$

$$= \int P(z_{t+k} \mid h_t, a_{t:t+k-1}) R_z(z_{t+k}, a_{t+k}) dz_{t+k} \tag{50}$$

$$\overset{k\text{-step } ZP}{=} \int P(z_{t+k} \mid \phi(h_t), a_{t:t+k-1}) R_z(z_{t+k}, a_{t+k}) dz_{t+k} \tag{51}$$

$$= \mathbb{E}[r_{t+k} \mid \phi(h_t), a_{t:t+k}], \tag{52}$$

where $k$-step ZP is implied by ZP by Prop. 10. $\square$

**Proposition 12 (OP implies multi-step OP in MDPs, but not POMDPs).** *For $k \in \mathbb{N}^+$, define $k$-step OP as*

$$P(o_{t+k} \mid h_t, a_{t:t+k-1}) = P(o_{t+k} \mid \phi(h_t), a_{t:t+k-1}), \quad \forall h, a, o. \tag{53}$$

*Proof.* We first show the result in MDPs. Assume a state encoder $\phi$ satisfies OP,

$$P(s_{t+k} \mid s_t, a_{t:t+k-1}) = \int P(s_{t+1:t+k} \mid s_t, a_{t:t+k-1}) ds_{t+1:t+k-1} \tag{54}$$

$$\overset{\text{MDPs}}{=} \int P(s_{t+1} \mid s_t, a_t) \prod_{i=2}^{k} P(s_{t+i} \mid s_{t+i-1}, a_{t+i-1}) ds_{t+1:t+k-1} \tag{55}$$

$$\overset{OP}{=} \int P(s_{t+1} \mid \phi(s_t), a_t) \prod_{i=2}^{k} P(s_{t+i} \mid s_{t+i-1}, a_{t+i-1}) ds_{t+1:t+k-1} \tag{56}$$

$$\overset{\text{MDPs}}{=} \int P(s_{t+1:t+k} \mid \phi(s_t), a_{t:t+k-1}) ds_{t+1:t+k-1} \tag{57}$$

$$= P(s_{t+k} \mid \phi(s_t), a_{t:t+k-1}). \tag{58}$$

However, in POMDPs, OP does not imply multi-step OP. This can be shown by a counterexample in Castro et al. (2009, Theorem 4.10), where the weak belief bisimulation relation corresponds to single-step OP and RP, while trajectory equivalence corresponds to multi-step OP and RP. The idea is to show that for two histories $h_t^1$ and $h_t^2$, if $P(o_{t+1} \mid h_t^1, a_t) = P(o_{t+1} \mid h_t^2, a_t), \forall o_{t+1}, a_t$, it does not imply that $P(o_{t+2} \mid h_t^1, a_t, o_{t+1}, a_{t+1}) = P(o_{t+2} \mid h_t^2, a_t, o_{t+1}, a_{t+1}), \forall o_{t+1:t+2}, a_{t:t+1}$. □

### A.3.3 RESULTS RELATED TO $\phi_{Q^*}$

*Proof sketch of **ZP + RP** ($\phi_L$) imply $\phi_{Q^*}$.* To show $Q^*(h, a) = Q_z^*(\phi_L(h), a), \forall h, a$, please see (Subramanian et al., 2022, Theorem 5 and Theorem 25) for finite-horizon and infinite-horizon POMDPs, respectively. For the approximate version, please see (Subramanian et al., 2022, Theorem 9 and Theorem 27). By definition, $Q^*(h, a) = Q_z^*(\phi_L(h), a), \forall h, a$ implies that $\phi_L$ is a kind of $\phi_{Q^*}$. □

*Proof of **Thm. 2** (**ZP** + $\phi_{Q^*}$ imply **RP**).* Suppose $\phi$ satisfies ZP and we train model-free RL with value parameterized by $\mathcal{Q}(\phi(h), a)$ to satisfy the Bellman optimality equation:

$$\mathcal{Q}(\phi(h_t), a_t) = \begin{cases} \mathbb{E}[r_t \mid h_t, a_t] & t = T, \\ \mathbb{E}[r_t \mid h_t, a_t] + \gamma \mathbb{E}_{o_{t+1} \sim P(\mid h_t, a_t)} \left[ \max_{a_{t+1}} \mathcal{Q}(\phi(h_{t+1}), a_{t+1}) \right] & \text{else,} \end{cases} \tag{59}$$

where the case $t = T$ only applies to finite-horizon problems (the same below). This is equivalent to say that $\mathcal{Q}(\phi(h_t), a_t) = Q^*(h_t, a_t), \forall h_t, a_t$, where $Q^*$ satisfies the Bellman optimality equation, too:

$$Q^*(h_t, a_t) = \begin{cases} \mathbb{E}[r_t \mid h_t, a_t] & t = T, \\ \mathbb{E}[r_t \mid h_t, a_t] + \gamma \mathbb{E}_{o_{t+1} \sim P(\mid h_t, a_t)} \left[ \max_{a_{t+1}} Q^*(h_{t+1}, a_{t+1}) \right] & \text{else.} \end{cases} \tag{60}$$

Now we can construct an abstract MDP with $\phi$. The latent transition matches due to ZP. The latent reward function is purely defined by latent value and latent transition[13]:

$$\mathcal{R}_z(z_t, a_t) := \begin{cases} \mathcal{Q}(z_t, a_t) & t = T, \\ \mathcal{Q}(z_t, a_t) - \gamma \mathbb{E}_{z_{t+1} \sim P(\mid z_t, a_t)} \left[ \max_{a_{t+1}} \mathcal{Q}(z_{t+1}, a_{t+1}) \right] & \text{else.} \end{cases} \tag{61}$$

We want to show RP condition: $\mathcal{R}_z(\phi(h_t), a_t) = \mathbb{E}[r_t \mid h_t, a_t], \forall h_t, a_t$.

Here is our proof. Recall that the grounded reward function can also be derived reversely by $Q^*$:

$$\mathbb{E}[r_t \mid h_t, a_t] := \begin{cases} Q^*(h_t, a_t) & t = T, \\ Q^*(h_t, a_t) - \gamma \mathbb{E}_{o_{t+1} \sim P(\mid h_t, a_t)} \left[ \max_{a_{t+1}} Q^*(h_{t+1}, a_{t+1}) \right] & \text{else.} \end{cases} \tag{62}$$

If the problem is finite-horizon with horizon $T$ and when $t = T$, RP holds due to $\mathcal{Q}(\phi(h_T), a_T) = Q^*(h_T, a_T)$.

Now consider general case when $t < T$ in finite-horizon ($\gamma = 1$) and any $t$ in infinite-horizon ($\gamma < 1$). Due to $Q$-value match ($\mathcal{Q}(\phi(h_t), a_t) = Q^*(h_t, a_t)$), it is equivalent to show that

$$\mathbb{E}_{o_{t+1} \sim P(\mid h_t, a_t)} \left[ \max_{a_{t+1}} Q^*(h_{t+1}, a_{t+1}) \right] = \mathbb{E}_{z_{t+1} \sim P(\mid \phi(h_t), a_t)} \left[ \max_{a_{t+1}} \mathcal{Q}(z_{t+1}, a_{t+1}) \right], \quad \forall h_t, a_t, t. \tag{63}$$

Proof for this:

$$\text{LHS} \stackrel{\phi_{Q^*}}{=} \mathbb{E}_{o_{t+1} \sim P(\mid h_t, a_t)} \left[ \max_{a_{t+1}} \mathcal{Q}(\phi(h_{t+1}), a_{t+1}) \right] \tag{64}$$

---

[13] In the main paper, we omit the finite-horizon case due to space limit.

$$= \int \left( \int P(o_{t+1} \mid h_t, a_t) \delta(z_{t+1} = \phi(h_{t+1})) do_{t+1} \right) \max_{a_{t+1}} \mathcal{Q}(z_{t+1}, a_{t+1}) dz_{t+1} \tag{65}$$

$$= \int P(z_{t+1} \mid h_t, a_t) \max_{a_{t+1}} \mathcal{Q}(z_{t+1}, a_{t+1}) dz_{t+1} \tag{66}$$

$$\stackrel{ZP}{=} \int P(z_{t+1} \mid \phi(h_t), a_t) \max_{a_{t+1}} \mathcal{Q}(z_{t+1}, a_{t+1}) dz_{t+1} = \text{RHS}. \tag{67}$$

$$\square$$

**Lemma 3** (**Integral probability metric** (Subramanian et al., 2022))**.** *Given by a function class $\mathcal{F}$, integral probability metric (IPM) between two distributions $\mathbb{P}, \mathbb{Q} \in \Delta(\mathcal{Z})$ is*

$$\mathcal{D}_{\mathcal{F}}(\mathbb{P}, \mathbb{Q}) = \sup_{f \in \mathcal{F}} |\mathbb{E}_{x \sim \mathbb{P}}[f(x)] - \mathbb{E}_{y \sim \mathbb{Q}}[f(y)]|. \tag{68}$$

*For any real-valued function $g$, the following inequality is derived by definition:*

$$|\mathbb{E}_{x \sim \mathbb{P}}[g(x)] - \mathbb{E}_{y \sim \mathbb{Q}}[g(y)]| \leq \rho_{\mathcal{F}}(g) \mathcal{D}_{\mathcal{F}}(\mathbb{P}, \mathbb{Q}), \tag{69}$$

*where $\rho_{\mathcal{F}}(g) := \inf\{\rho \in \mathbb{R}_+ \mid \rho^{-1} g \in \mathcal{F}\}$ is a Minkowski functional.*

*Remark* 2. Some examples include:

- Total Variance (TV) distance is an IPM defined by $\mathcal{F}_{\text{TV}} = \{f : \|f\|_\infty \leq 1\}$.

- Wasserstein (W) distance is an IPM defined by $\mathcal{F}_{\text{W}} = \{f : \|f\|_L \leq 1\}$.

- KL divergence is not an IPM, but is an upper bound of TV distance by Pinsker's inequality:

$$\mathcal{D}_{\mathcal{F}_{\text{TV}}}(\mathbb{P}, \mathbb{Q}) \leq \sqrt{2 D_{\text{KL}}(\mathbb{P} \mid\mid \mathbb{Q})}. \tag{70}$$

**Theorem 5** (**Approximate version of Thm. 2 (approximate ZP and approximate $\phi_{Q^*}$ imply approximate $\phi_L$**)**.** *Suppose the encoder $\phi$ satisfies **approximate ZP** (AZP) and we train model-free RL with value parameterized by $\mathcal{Q}(\phi(h), a)$ to **approximate** $Q^*(h, a)$, namely: $\forall t, h_t, a_t$,*

$$\exists P_z : \mathcal{Z} \times \mathcal{A} \to \Delta(\mathcal{Z}), \quad \text{s.t.} \quad \mathcal{D}_{\mathcal{F}}(P(z_{t+1} \mid h_t, a_t), P_z(z_{t+1} \mid \phi(h_t), a_t)) \leq \delta_t, \tag{AZP}$$

$$|Q^*(h_t, a_t) - \mathcal{Q}(\phi(h_t), a_t)| \leq \alpha_t. \tag{Approx. $\phi_{Q^*}$}$$

*where $\mathcal{D}_{\mathcal{F}}$ is an IPM. Under these conditions, we can construct a latent reward function:*

$$\mathcal{R}_z(z_t, a_t) := \begin{cases} \mathcal{Q}(z_t, a_t) & t = T, \\ \mathcal{Q}(z_t, a_t) - \gamma \mathbb{E}_{z_{t+1} \sim P_z(\mid z_t, a_t)} \left[ \max_{a_{t+1}} \mathcal{Q}(z_{t+1}, a_{t+1}) \right] & \text{else,} \end{cases} \tag{71}$$

*such that*

$$|\mathbb{E}[r_t \mid h_t, a_t] - \mathcal{R}_z(\phi(h_t), a_t)| \leq \epsilon_t, \quad \forall t, h_t, a_t, \tag{ARP}$$

$$\text{where} \quad \epsilon_t = \begin{cases} \alpha_T & t = T, \\ \alpha_t + \gamma(\alpha_{t+1} + \rho_{\mathcal{F}}(\mathcal{V}_{t+1})\delta_t) & \text{else,} \end{cases} \tag{72}$$

$$\mathcal{V}(z_t) = \max_{a_t} \mathcal{Q}(z_t, a_t), \tag{73}$$

*where $\mathcal{V}_{t+1}$ is the latent state-value function $\mathcal{V}$ at step $t+1$.*

*Proof.* For the case of $t = T$ in finite-horizon, ARP holds by the assumption of approx. $\phi_{Q^*}$. Now we discuss generic case of $t$. Recall the reward and latent reward can be rewritten as:

$$\mathbb{E}[r_t \mid h_t, a_t] = Q^*(h_t, a_t) - \gamma \mathbb{E}_{o_{t+1} \sim P(\mid h_t, a_t)} \left[ \max_{a_{t+1}} Q^*(h_{t+1}, a_{t+1}) \right], \tag{74}$$

$$\mathcal{R}_z(z_t, a_t) = \mathcal{Q}(z_t, a_t) - \gamma \mathbb{E}_{z_{t+1} \sim P(\mid z_t, a_t)} \left[ \max_{a_{t+1}} \mathcal{Q}(z_{t+1}, a_{t+1}) \right]. \tag{75}$$

Therefore, the reward gap is upper bounded:

$$|\mathbb{E}[r_t \mid h_t, a_t] - \mathcal{R}_z(\phi(h_t), a_t)| \tag{76}$$

$$\leq |Q^*(h_t, a_t) - \mathcal{Q}(\phi(h_t), a_t)| \tag{77}$$

$$+ \gamma \left| \mathbb{E}_{o_{t+1} \sim P(\cdot|h_t, a_t)} \left[ \max_{a_{t+1}} Q^*(h_{t+1}, a_{t+1}) \right] - \mathbb{E}_{z_{t+1} \sim P(\cdot|\phi(h_t), a_t)} \left[ \max_{a_{t+1}} \mathcal{Q}(z_{t+1}, a_{t+1}) \right] \right| \tag{78}$$

$$\leq \alpha_t + \gamma \left| \mathbb{E}_{o_{t+1} \sim P(\cdot|h_t, a_t)} \left[ \max_{a_{t+1}} Q^*(h_{t+1}, a_{t+1}) - \max_{a_{t+1}} \mathcal{Q}(\phi(h_{t+1}), a_{t+1}) \right] \right| \tag{79}$$

$$+ \gamma \left| \mathbb{E}_{o_{t+1} \sim P(\cdot|h_t, a_t)} \left[ \max_{a_{t+1}} \mathcal{Q}(\phi(h_{t+1}), a_{t+1}) \right] - \mathbb{E}_{z_{t+1} \sim P(\cdot|\phi(h_t), a_t)} \left[ \max_{a_{t+1}} \mathcal{Q}(z_{t+1}, a_{t+1}) \right] \right| \tag{80}$$

$$\leq \alpha_t + \gamma \mathbb{E}_{o_{t+1} \sim P(\cdot|h_t, a_t)} \left[ \max_{a_{t+1}} |Q^*(h_{t+1}, a_{t+1}) - \mathcal{Q}(\phi(h_{t+1}), a_{t+1})| \right] \tag{81}$$

$$+ \gamma \left| \mathbb{E}_{z_{t+1} \sim P(\cdot|h_t, a_t)} \left[ \max_{a_{t+1}} \mathcal{Q}(z_{t+1}, a_{t+1}) \right] - \mathbb{E}_{z_{t+1} \sim P(\cdot|\phi(h_t), a_t)} \left[ \max_{a_{t+1}} \mathcal{Q}(z_{t+1}, a_{t+1}) \right] \right| \tag{82}$$

$$\leq \alpha_t + \gamma \alpha_{t+1} + \gamma \left| \mathbb{E}_{z_{t+1} \sim P(\cdot|h_t, a_t)}[\mathcal{V}(z_{t+1})] - \mathbb{E}_{z_{t+1} \sim P(\cdot|\phi(h_t), a_t)}[\mathcal{V}(z_{t+1})] \right| \tag{83}$$

$$\leq \alpha_t + \gamma(\alpha_{t+1} + \rho_{\mathcal{F}}(\mathcal{V}_{t+1})\delta_t), \tag{84}$$

where Eq. 77 is by triangle inequality, Eq. 79 is by triangle inequality and approx. $\phi_{Q^*}$, Eq. 81 is by the maximum-absolute-difference inequality $|\max f(x) - \max g(x)| \leq \max|f(x) - g(x)|$, Eq. 83 is by approx. $\phi_{Q^*}$, and Ln. 84 is by the property of IPM (Eq. 69 in Lemma 3) and AZP. □

*Remark* 3. In the infinite-horizon problem, assume $\delta_t = \delta$ and $\alpha_t = \alpha$ for any $t$, and $\mathcal{D}_{\mathcal{F}}$ is Wasserstein distance. Furthermore, assume the latent reward $\mathcal{R}_z(z, a)$ is $L_r$-Lipschitz and the latent transition $P_z(z' \mid z, a)$ is $L_p$-Lipschitz, then by (Subramanian et al., 2022, Lemma 44), if $\gamma L_p < 1$,

$$\rho_{\mathcal{F}}(\mathcal{V}_{t+1}) = \|\mathcal{V}\|_L \leq \frac{L_r}{1 - \gamma L_p}, \quad \forall t. \tag{85}$$

Thus, the reward difference bound can be rewritten as

$$\epsilon \leq (1 + \gamma)\alpha + \frac{\gamma L_r \delta}{1 - \gamma L_p}. \tag{86}$$

## B    OBJECTIVES AND OPTIMIZATION IN SELF-PREDICTIVE RL

*Proof of Prop. 1.* First, we show $\mathcal{L}_{ZP,\ell}(\phi, \theta; h, a) \leq J_\ell(\phi, \theta, \phi; h, a), \forall h, a$.

$$J_\ell(\phi, \theta, \phi; h, a) - \mathcal{L}_{ZP,\ell}(\phi, \theta; h, a) \tag{87}$$

$$= \mathbb{E}_{o' \sim P(\cdot|h,a)} \left[ \|g_\theta(f_\phi(h), a) - f_\phi(h')\|_2^2 \right] - \|g_\theta(f_\phi(h), a) - \mathbb{E}_{o' \sim P(\cdot|h,a)}[f_\phi(h')]\|_2^2 \tag{88}$$

$$= \|g_\theta(f_\phi(h), a)\|_2^2 - 2\mathbb{E}_{o'}[\langle g_\theta(f_\phi(h), a), f_\phi(h') \rangle] + \mathbb{E}_{o'} \left[ \|f_\phi(h')\|_2^2 \right] \tag{89}$$

$$- \|g_\theta(f_\phi(h), a)\|_2^2 + 2\langle g_\theta(f_\phi(h), a), \mathbb{E}_{o'}[f_\phi(h')] \rangle - \|\mathbb{E}_{o'}[f_\phi(h')]\|_2^2 \tag{90}$$

$$= \mathbb{E}_{o'} \left[ \|f_\phi(h') - \mathbb{E}_{o'}[f_\phi(h')]\|_2^2 \right] \geq 0. \tag{91}$$

The inner product terms are cancelled due to the fact that inner product is bilinear. The equality holds when $P(o' \mid h, a)$ is deterministic.

Second, the ideal objective using $\ell_2$ distance $\mathcal{L}_{ZP,\ell}(\phi, \theta; h, a) = \|g_\theta(f_\phi(h), a) - \mathbb{E}_{o' \sim P(\cdot|h,a)}[f_\phi(h')]\|_2^2$ can only lead to EZP condition when reaching optimum of zero. This is because when $g_\theta(f_\phi(h), a) = \mathbb{E}_{o' \sim P(\cdot|h,a)}[f_\phi(h')], \forall h, a$, it precisely satisfies the EZP condition. □

*Proof of Prop. 2.* The goal is to show $\mathcal{L}_{ZP,D_{\mathtt{f}}}(\phi, \theta; h, a) \leq J_{D_{\mathtt{f}}}(\phi, \theta, \phi; h, a), \forall h, a$.

Recall the definition of $\mathtt{f}$-divergence that subsumes forward and reverse KL divergences: $D_{\mathtt{f}}(Q \mid\mid P) = \int P(x) f\left(\frac{Q(x)}{P(x)}\right) dx$, where $f : [0, \infty) \to \mathbb{R}$ is a convex function.

$$\mathcal{L}_{ZP,D_{\mathtt{f}}}(\phi, \theta; h, a) = \int \mathbb{P}_\phi(z \mid h) D_{\mathtt{f}}(\mathbb{P}_\phi(z' \mid h, a) \mid\mid \mathbb{P}_\theta(z' \mid z, a)) dz \tag{92}$$

$$= \iint \mathbb{P}_\phi(z \mid h)\mathbb{P}_\theta(z' \mid z,a)f\left(\frac{\mathbb{E}_{o'}[\mathbb{P}_\phi(z' \mid h')]}{\mathbb{P}_\theta(z' \mid z,a)}\right)dzdz' \tag{93}$$

$$= \iint \mathbb{P}_\phi(z \mid h)\mathbb{P}_\theta(z' \mid z,a)f\left(\mathbb{E}_{o'}\left[\frac{\mathbb{P}_\phi(z' \mid h')}{\mathbb{P}_\theta(z' \mid z,a)}\right]\right)dzdz' \tag{94}$$

$$\leq \iint \mathbb{P}_\phi(z \mid h)\mathbb{P}_\theta(z' \mid z,a)\mathbb{E}_{o'}\left[f\left(\frac{\mathbb{P}_\phi(z' \mid h')}{\mathbb{P}_\theta(z' \mid z,a)}\right)\right]dzdz' \tag{95}$$

$$= \mathbb{E}_{z\sim P_\phi(\mid h),o'\sim P(\mid h,a)}\left[\int \mathbb{P}_\theta(z' \mid z,a)f\left(\frac{\mathbb{P}_\phi(z' \mid h')}{\mathbb{P}_\theta(z' \mid z,a)}\right)dz'\right] \tag{96}$$

$$= J_{D_f}(\phi,\theta,\phi;h,a), \tag{97}$$

where we use Jensen's inequality by the convexity of $f$. The equality holds when $P(o' \mid h,a)$ is deterministic according to Jensen's inequality. $\square$

**Discussion on the double sampling issue.** The ideal objective Eq. 1 is hard to have an *unbiased* estimate in stochastic environments. This is due to double sampling issue (Baird, 1995) that we do not allow agent to i.i.d. sample twice from transition, *i.e.* $o'_1, o'_2 \sim P(o' \mid h,a)$. To see it more clearly, for example, when $\mathbb{D}$ is forward KL divergence, The ideal objective becomes:

$$\mathcal{L}_{ZP,\text{FKL}}(\phi,\theta;h,a) = \mathbb{E}_{z\sim\mathbb{P}_\phi(\mid h)}[D_{\text{KL}}(\mathbb{P}_\phi(z' \mid h,a) \mid\mid \mathbb{P}_\theta(z' \mid z,a))] \tag{98}$$

$$= \mathbb{E}_{z\sim\mathbb{P}_\phi(\mid h)}[D_{\text{KL}}(\mathbb{E}_{o'}[\mathbb{P}_\phi(z' \mid h')] \mid\mid \mathbb{P}_\theta(z' \mid z,a))] \tag{99}$$

$$= \mathbb{E}_{z\sim\mathbb{P}_\phi(\mid h)}\left[\int \mathbb{E}_{o'}[\mathbb{P}_\phi(z' \mid h')]\log\frac{\mathbb{E}_{o'}[\mathbb{P}_\phi(z' \mid h')]}{\mathbb{P}_\theta(z' \mid z,a)}dz'\right] \tag{100}$$

$$= \mathbb{E}_{z\sim\mathbb{P}_\phi(\mid h),o'\sim P(\mid h,a),z'\sim\mathbb{P}_\phi(\mid h')}\left[\log\frac{\mathbb{E}_{o'_+}[\mathbb{P}_\phi(z' \mid h'_+)]}{\mathbb{P}_\theta(z' \mid z,a)}\right], \tag{101}$$

where $o', o'_+ \sim P(\mid h,a)$ are two i.i.d. samples, and $h' = (h,a,o'), h'_+ = (h,a,o'_+)$.

*Proof of Prop. 3.* Recall the stop-gradient objective Eq. 2:

$$J := J_\ell(\phi,\theta,\overline{\phi};h,a) = \mathbb{E}_{o'\sim P(\mid h,a)}\left[\|g_\theta(f_\phi(h),a) - f_{\overline{\phi}}(h')\|_2^2\right]. \tag{102}$$

The gradients are:

$$\nabla_{\phi,\theta}\mathbb{E}_{h,a}[J] = \mathbb{E}_{h,a}\left[(g_\theta(f_\phi(h),a) - \mathbb{E}_{o'}[f_{\overline{\phi}}(h')])^\top \nabla_{\phi,\theta}g_\theta(f_\phi(h),a)\right]. \tag{103}$$

When $\theta,\phi$ reaches a stationary point, we have $\nabla_{\phi,\theta}\mathbb{E}_{h,a}[J] = 0$ and thus $\overline{\phi} = \phi$. Therefore, we have *a* stationary point $(\theta^*,\phi^*)$ such that: $g_\theta(f_\phi(h),a) = \mathbb{E}_{o'\sim P(\mid h,a)}[f_\phi(h')]$, for any $h,a$, which is the **expected** ZP (EZP) condition. In a deterministic environment, EZP is equivalent to ZP. In a stochastic environment, EZP is to match the expectation (instead of distribution).

However, in online objective, the gradient w.r.t. $\phi$ contains an extra term:

$$\mathbb{E}_{h,a,o'}\left[(g_\theta(f_\phi(h),a) - f_\phi(h'))^\top \nabla_\phi f_\phi(h')\right]. \tag{104}$$

Thus, when EZP holds, the gradient is zero in deterministic environments, but can be non-zero in stochastic environments.

$\square$

*Proof of Thm. 3.* **The setup.** Let $h_{t:-k}$ a vectorization of the recent truncation of history $h_t$ with window size of $k \in \mathbb{N}$, *i.e.* $h_{t:-k} = \text{vec}(a_{t-k},o_{t-k+1},\dots,a_{t-1},o_t) \in \mathbb{R}^x$,[14] where $x = k(|\mathcal{O}| + |\mathcal{A}|)$. We assume a linear encoder that maps history $h_t \in \mathcal{H}_t$ into $z_t$:

$$z_t = f_\phi(h_t) := \phi^\top h_{t:-k} \in \mathbb{R}^d, \tag{105}$$

---

[14]We zero pad $a_i$ and $o_i$ if $i \leq 0$.

where $k \in \mathbb{N}$ is a constant, and the parameters $\phi \in \mathbb{R}^{x \times d}$. In other words, the linear encoder only operates on recent histories of a fixed window size. We assume a linear deterministic latent transition

$$z_{t+1} = g_\theta(z_t, a_t) := \theta_z^\top z_t + \theta_a^\top a_t \in \mathbb{R}^d, \tag{106}$$

where the parameters $\theta_z \in \mathbb{R}^{d \times d}$ and $\theta_a \in \mathbb{R}^{a \times d}$. In fact, the result can be generalized to a non-linear dependence of actions.

**The proof.** The continuous-time training dynamics of $\phi$:

$$\dot{\phi} = -\mathbb{E}_{h_t, a_t}\left[\nabla_\phi J_\ell(\phi, \theta, \overline{\phi}; h_t, a_t)\right] \tag{107}$$

$$= -\mathbb{E}_{h_t, a_t, o_{t+1}}\left[\nabla_\phi \|\theta_z^\top \phi^\top h_{t:-k} + \theta_a^\top a_t - \overline{\phi}^\top h_{t+1:-k}\|_2^2\right] \tag{108}$$

$$= -\mathbb{E}_{h_t, a_t}\left[(\theta_z^\top \phi^\top h_{t:-k} + \theta_a^\top a_t - \mathbb{E}_{o_{t+1}}\left[\overline{\phi}^\top h_{t+1:-k}\right])^\top \nabla_\phi \theta_z^\top \phi^\top h_{t:-k}\right] \tag{109}$$

$$= -\mathbb{E}_{h_t, a_t}\left[h_{t:-k}(\theta_z^\top \phi^\top h_{t:-k} + \theta_a^\top a_t - \mathbb{E}_{o_{t+1}}\left[\overline{\phi}^\top h_{t+1:-k}\right])^\top\right]\theta_z^\top. \tag{110}$$

The gradient of the loss w.r.t. $\theta_z$:

$$\nabla_{\theta_z}\mathbb{E}_{h_t, a_t}\left[J_\ell(\phi, \theta, \overline{\phi}; h_t, a_t)\right] \tag{111}$$

$$= \mathbb{E}_{h_t, a_t}\left[(\theta_z^\top \phi^\top h_{t:-k} + \theta_a^\top a_t - \mathbb{E}_{o_{t+1}}\left[\overline{\phi}^\top h_{t+1:-k}\right])^\top \nabla_{\theta_z}(\theta_z^\top \phi^\top h_{t:-k} + \theta_a^\top a_t)\right] \tag{112}$$

$$= \phi^\top \mathbb{E}_{h_t, a_t}\left[h_{t:-k}(\theta_z^\top \phi^\top h_{t:-k} + \theta_a^\top a_t - \mathbb{E}_{o_{t+1}}\left[\overline{\phi}^\top h_{t+1:-k}\right])^\top\right] \in \mathbb{R}^{d \times d}. \tag{113}$$

Therefore, we have

$$\phi^\top \dot{\phi} = -\nabla_{\theta_z}\mathbb{E}_{h_t, a_t}\left[J_\ell(\phi, \theta, \overline{\phi}; h_t, a_t)\right]\theta_z^\top. \tag{114}$$

Following the practice in (Tang et al., 2022), we assume $\nabla_{\theta_z}\mathbb{E}_{h_t, a_t}\left[J_\ell(\phi, \theta, \overline{\phi}; h_t, a_t)\right] = 0$, *i.e.* $\theta_z$ reaches the stationary point of the inner optimization that depends on $\phi$, then $\phi^\top \dot{\phi} = 0$. Thus, the training dynamics of $\phi^\top \phi$ is

$$\frac{d(\phi^\top \phi)}{dt} = \dot{\phi}^\top \phi + \phi^\top \dot{\phi} = \dot{\phi}^\top \phi + (\dot{\phi}^\top \phi)^\top = 0. \tag{115}$$

This means that $\phi^\top \phi$ keeps same value during training. $\qquad\square$

## C    ANALYZING PRIOR WORKS ON STATE AND HISTORY REPRESENTATION LEARNING

In this section, we provide a concise but analytical overview of previous works that learn or approximate self-predictive or observation-predictive representations on states or histories. Please see Lesort et al. (2018) for an early and detailed survey on *state* representation learning.

We focus on the objectives of state or history encoders in their value functions. For each work discussed, we present a summary of the conditions that their encoders aim to satisfy or approximate at the beginning of each paragraph. In cases where multiple encoder objectives are proposed, we select the one employed in their primary experiments for our discussion. In particular, we list the exact objectives they aim to optimize, which might be redundant for *exact* conditions. For example, multi-step RP can be implied by RP + ZP by Prop. 11 (or $\phi_{Q^*}$ + ZP by Thm. 2), and multi-step ZP can be implied by ZP by Prop. 10.

### C.1    SELF-PREDICTIVE REPRESENTATIONS

**CRAR (François-Lavet et al., 2019):** $\phi_{Q^*}$ **+ RP + ZP with online $\ell_2$ + regularization.** Combined reinforcement via abstract representations (CRAR) is designed to learn self-predictive representations in MDPs. It incorporates RP and ZP auxiliary losses into the end-to-end RL objective. They assume the deterministic case (for the encoder, transition and latent transition), thus using $\ell_2$ objective is sufficient (Prop. 1). They use online ZP target and observe the representation collapse when the reward signals are scarce. To prevent this issue, they introduce regularization terms into the encoder objective. These terms minimize $\mathbb{E}_{s_1, s_2}[\exp(-\|\phi(s_1) - \phi(s_2)\|_2)] + \mathbb{E}_s\left[\max(\|\phi(s)\|_\infty^2 - 1, 0)\right]$, where $s_1, s_2, s$ are samples from the state space. These terms, similar to entropy maximization, encourage diversity within the latent space.

**DeepMDP (Gelada et al., 2019):** $\phi_{Q^*}$ **+ RP + ZP with online** $\ell_2$**.** DeepMDP aims to learn state representations that match RP and ZP. In their experiments, they assume deterministic case, resulting in dirac distributions $\mathbb{P}_\phi(z' \mid s')$ and $\mathbb{P}_\theta(z' \mid z, a)$. Although they use the Wasserstein distance, it reduces to $\ell_2$ distance for two dirac distributions. They use an online target in ZP loss. In their toy DonutWorld task, they try phased training with RP + ZP, but the agent tends to be trapped in a local minimum of zero ZP. Then they try $\phi_{Q^*}$ + RP + ZP in Atari by training RP + ZP as an auxiliary task of a distributional RL baseline, outperforming the baseline in their main result. They also find that $\phi_{Q^*}$ + RP + ZP is comparable to $\phi_{Q^*}$ + ZP, aligned with our theoretical prediction based on Thm. 2. They also try phased training in Atari and find that RP + ZP performs poorly, while RP + ZP + OR yields good results.

**SPR (Schwarzer et al., 2020):** $\phi_{Q^*}$ **+ multi-step ZP with EMA** cos**.** Self-Predictive Representations (SPR) improves the ZP objective in DeepMDP. They use a special kind of $\ell_2$ loss (*i.e.* cos distance) to bound the loss scale, and use an EMA target. They use multi-step prediction loss to learn the condition:

$$P(z_{t+1:t+k} \mid s_t, a_{t:t+k-1}) = P(z_{t+1:t+k} \mid \phi(s_t), a_{t:t+k-1}), \tag{116}$$

where $k = 5$ in their experiments. In addition, to reduce the large latent space generated by CNNs, they use a linear projection of the latent states to satisfy ZP.

**DBC (Zhang et al., 2020):** $\phi_{Q^*}$ **+ RP + stronger ZP with detached FKL.** Deep Bisimulation for Control (DBC) trains the state encoder $\phi$ with several auxiliary losses, including RP and ZP. The ZP loss uses a forward KL objective with a detached target. Their main contribution is the introduction of the bisimulation metric (Ferns et al., 2004) into state representation learning: for any $s_i, s_j \in \mathcal{S}$ and $a_i, a_j \in \mathcal{A}$,

$$\|\phi(s_i) - \phi(s_j)\|_1 = |R(s_i, a_i) - R(s_j, a_j)| + \gamma W(\mathbb{P}_\theta(z' \mid \phi(s_i), a_i), \mathbb{P}_\theta(z' \mid \phi(s_j), a_j)), \tag{metric}$$

where $W$ is Wasserstein distance and $\mathbb{P}_\theta$ is modeled as a Gaussian. The metric condition enforces the latent space to be structured with a $\ell_1$ metric. They train $\phi$ satisfying the metric condition by minimizing the mean square error on it as another auxiliary loss. This leads to a stronger ZP condition.

**PBL (Guo et al., 2020):** $\phi_{Q^*}$ **+ indirect multi-step ZP.** Predictions of Bootstrapped Latents (PBL) designs two auxiliary losses, reverse prediction and forward prediction, for their history encoder $\phi$, transition model $\theta$, observation encoder $f$, and projector $g$:

$$\min_{f,g} \mathbb{E}_h \left[ \|g(f(o)) - \phi(h)\|_2^2 \right], \tag{Reverse}$$

$$\min_{\phi,\theta} \mathbb{E}_{h,a,o'} \left[ \|\theta(\phi(h), a) - f(o')\|_2^2 \right]. \tag{Forward}$$

To understand their connection with ZP, assume the two losses reach zero with $\phi(h) = g(f(o))$ and $\theta(\phi(h), a) = \mathbb{E}_{o' \sim P(\cdot|h,a)}[f(o')]$ for any $h, a$, although in theory this may be unrealizable. Furthermore, assume deterministic transition, then

$$g(\theta(\phi(h), a)) = g(f(o')) = \phi(h'). \tag{117}$$

Therefore, in deterministic environments, reverse and forward prediction together is equivalent to ZP if they reach the optimum. They also adopt multi-step version of their loss with a horizon of 20. While forward and reverse prediction both appear critical in this work, the follow-up work BYOL-explore (Guo et al., 2022) removes reverse prediction.

**Successor Representations and Features (Barreto et al., 2017; Lehnert & Littman, 2020):** $\phi_{Q^*}$ **+ RP + weak ZP.** Here, we introduce successor features (SF) with our notation. Suppose the expected reward function can be computed as

$$\mathbb{E}[r \mid s, a] = g(\phi(s), a)^\top w, \quad \forall s, a, \tag{118}$$

where $\phi : \mathcal{S} \to \mathcal{Z}$ is a state encoder and $g : \mathcal{Z} \times \mathcal{A} \to \mathbb{R}^d$ is called state-action feature extractor, and $w \in \mathbb{R}^d$ are weights[15]. In our notation, Eq. 118 is RP condition for $\phi$.

---

[15] Although it is linear w.r.t. $w$, it can recover any reward function, *e.g.* when $\phi(s) = s$ and $g(s, a)_i = \mathbb{E}[r \mid s, a]$ for some $i$.

As a special case, in tabular MDPs with finite state and action spaces with state-dependent reward $R(s)$, let $\phi(s) \in \{0,1\}^{|S|}$ be one-hot state representation, and let $g(\phi(s),a) = \phi(s)$ and weight $w_s = \mathbb{E}[r \mid s]$, this satisfies Eq. 118. This special case is known as **successor representation** (SR) setting (Dayan, 1993). In deep SR (Kulkarni et al., 2016; Lehnert & Littman, 2020), they allow learning $\phi$ with assuming $g(\phi(s),a) = \phi(s)$.

The $Q$-value function of a policy $\pi$ can be rewritten as

$$Q^\pi(s,a) = \mathbb{E}_\pi\left[\sum_{t=0}^\infty \gamma^t r_t \mid S_0 = s, A_0 = a\right] \tag{119}$$

$$= \mathbb{E}_\pi\left[\sum_{t=0}^\infty \gamma^t g(\phi(s_t),a_t)^\top w \mid S_0 = s, A_0 = a\right] \tag{120}$$

$$= \mathbb{E}_\pi\left[\sum_{t=0}^\infty \gamma^t g(\phi(s_t),a_t) \mid S_0 = s, A_0 = a\right]^\top w \tag{121}$$

$$:= \psi^\pi(s,a)^\top w, \tag{122}$$

where $\psi^\pi(s,a)$ is called successor features (Barreto et al., 2017), a geometric sum of future $g(\phi(s),a)$. Although $\psi^\pi$ can belong to any function class, following deep SR (Kulkarni et al., 2016; Lehnert & Littman, 2020), we assume it is parametrized by the state encoder as $\psi^\pi(s,a) = f^\pi(\phi(s),a)$ where $f^\pi : \mathcal{Z} \times \mathcal{A} \to \mathbb{R}^d$. Then, by plugging Eq. 122 in Bellman equation $Q^\pi(s,a) = \mathbb{E}_{s',a' \sim \pi}[R(s,a) + \gamma Q^\pi(s',a')]$, we have

$$f^\pi(\phi(s),a) = g(\phi(s),a) + \gamma \mathbb{E}_{s',a' \sim \pi}[f^\pi(\phi(s'),a')]. \tag{123}$$

Therefore, Eq. 123 can be viewed as a **weak** version of ZP, because given any current latent state and action pair $(\phi(s),a)$, Eq. 123 can predict the expectation of some function of next latent state $\phi(s')$. ZP can imply Eq. 123 because it can predict exactly the distribution of next latent state.

With a combination of RP (Eq. 118), $\phi_{Q^*}$ (implied by Eq. 123 when $\pi$ is optimal), and a weak version of ZP, we show that the state encoder that successor features learn, belongs to a weak version of $\phi_L$.

As a special case, in Linear Successor Feature Model (LSFM) (Lehnert & Littman, 2020, Theorem 2), they show that SF is **exactly** the bisimulation ($\phi_L$) under several assumptions: finite action and latent space, the successor features $f^\pi(z,a) = F_a z$ is a linear function, and the policy $\pi : \mathcal{Z} \to \Delta(\mathcal{A})$ conditions on latent space. However, here we point it out that with the assumptions above implies EZP (not necessarily ZP), thus, still a **weak** version of bisimulation.

Following Lehnert & Littman (2020), assume the finite latent space is composed of one-hot vectors: $\mathcal{Z} = \{e_1, e_2, \ldots, e_n\}$, we can construct a matrix $F^\pi \in \mathbb{R}^{d \times n}$ with each column $F^\pi(i) = \mathbb{E}_{a \sim \pi(\cdot | e_i)}[F_a e_i]$.

$$\frac{1}{\gamma}(f^\pi(\phi(s),a) - g(\phi(s),a)) = \mathbb{E}_{s',a' \sim \pi}[f^\pi(\phi(s'),a')] \tag{124}$$

$$= \mathbb{E}_{s' \sim P(\cdot|s,a), a' \sim \pi(\cdot|\phi(s'))}[F_{a'}\phi(s')] \tag{125}$$

$$= \mathbb{E}_{s' \sim P(\cdot|s,a)}[F^\pi \phi(s')] = F^\pi \mathbb{E}_{s' \sim P(\cdot|s,a)}[\phi(s')]. \tag{126}$$

By (Lehnert & Littman, 2020, Lemma 4), $F^\pi$ is invertible, thus there exists a function $J : \mathcal{Z} \times \mathcal{A} \to \mathcal{Z}$ such that $J(\phi(s),a) = \mathbb{E}_{s' \sim P(\cdot|s,a)}[\phi(s')]$, *i.e.*, EZP holds.

**EfficientZero (Ye et al., 2021):** $\phi_{Q^*}$ + **RP** + multi-step **ZP with detached** cos. EfficientZero improves MuZero (Schrittwieser et al., 2020) by introducing ZP loss as one of their main contributions. We consider it especially crucial to planning algorithms because ZP enforces the latent model to be accurate. Similar to SPR (Schwarzer et al., 2020), they use 5-step cos objective with a projection on latent states, and add image data augmentation for visual RL tasks.

**RPC (Eysenbach et al., 2021):** $\phi_{\pi^*}$ + **ZP with online forward KL.** From the perspective of information compression, robust predictive control (RPC) aims to jointly learn the encoder of policy $\mathbb{P}_\phi(z \mid s)$ and the latent policy $\pi_z(a \mid z)$ in MDPs. The policy $\pi(a \mid s)$ is not only maximizing return,

but also imposed a constraint on $\mathbb{E}_\pi[I(s_{1:\infty}; z_{1:\infty})] \leq C$ where $C > 0$ is a predefined constant. By applying variational information bottleneck, this constraint induces the algorithm RPC to maximize the following objective w.r.t. $\phi$ and $\theta$ (see their Eq. 6):

$$\mathcal{L}(\phi, \theta; s_t, a_t) = \mathbb{E}_{z_t \sim \mathbb{P}_\phi(\cdot|s_t), s_{t+1} \sim P(\cdot|s_t, a_t), z_{t+1} \sim \mathbb{P}_\phi(\cdot|s_{t+1})} \left[ \log \frac{\mathbb{P}_\theta(z_{t+1} \mid z_t, a_t)}{\mathbb{P}_\phi(z_{t+1} \mid s_{t+1})} \right] \tag{127}$$

$$= -\mathbb{E}_{z_t \sim \mathbb{P}_\phi(\cdot|s_t), s_{t+1} \sim P(\cdot|s_t, a_t)}[D_{\mathrm{KL}}(\mathbb{P}_\phi(z_{t+1} \mid s_{t+1}) \,||\, \mathbb{P}_\theta(z_{t+1} \mid z_t, a_t))] \tag{128}$$

which is exactly the practical forward KL objective Eq. 3. In practice, the authors formulate it as constrained optimization and use gradient descent-ascent to update the encoder and Lagrange multiplier. In addition, they also use this objective as an intrinsic reward to regularize the latent policy's reward-maximizing objective. It is worth noting that while RPC aims to learn the ZP condition along with reward maximization, it does not explicitly learn representations to fulfill the RP or $\phi_{Q^*}$ conditions. As a result, we can consider it as an approach that approximates self-predictive representations.

**ALM (Ghugare et al., 2022): $\phi_{Q^*}$ + multi-step ZP with EMA reverse KL.** Aligned Latent Models (ALM) is based on variational inference, and aims to learn the latent model $\mathbb{P}_\theta(z' \mid z, a)$, the state encoder $\mathbb{P}_\phi(z \mid s)$ and the latent policy $\pi_z(z)$ to jointly maximize the lower bound of the expected return. The objective of their encoder includes maximizing the return and ZP loss, instantiated as 3-step reverse KL with an EMA target. Specifically, given a tuple of $(s, a, s')$, the 1-step objective for their encoder is computed as

$$\min_\phi \mathbb{E}_{z \sim \mathbb{P}_\phi(\cdot|s)} \left[ -R_z(z, a) + D_{\mathrm{KL}}(\mathbb{P}_\theta(z' \mid z, a) \,||\, \mathbb{P}_\phi(z' \mid s')) - \mathbb{E}_{z' \sim \mathbb{P}_\theta(\cdot|z,a)} \left[ Q^\pi(z', \pi_z(\overline{z'})) \right] \right], \tag{129}$$

where $R_z(z, a)$ is the latent reward, learned by the RP condition (with $\phi$ detached), and $\overline{z'}$ indicates stop-gradient. With the latent reward and also their intrinsic rewards, they perform SVG algorithm (Heess et al., 2015) for policy optimization with a planning horizon of 3 steps. We provide a detailed description of ALM and its variants in Sec. E.2.

**AIS (Subramanian et al., 2022): RP + ZP with detached $\ell_2$ or forward KL in their approach, while RP + OP with detached $\ell_2$ in their experiments.** Approximate Information States (AIS) adopts a phased training framework where the history encoder $\phi$ learns from RP instead of maximizing returns. In their approach section (Subramanian et al., 2022, Sec. 6.1.2), they propose using MMD with $\ell_2$ distance-based kernel $k_d$ to learn ZP, and detach the target. The distance-based kernel (Sejdinovic et al., 2013) takes a pair of latent states $z_1, z_2 \in \mathcal{Z}$ as inputs, and is defined as $k_d(z_1, z_2) = \frac{1}{2}(d(z_0, z_1) + d(z_0, z_2) - d(z_1, z_2))$ where $z_0 \in \mathcal{Z}$ is arbitrary. In this case, $d(z_1, z_2) = \|z_1 - z_2\|_2^2$ is $\ell_2$ distance.

Let $f_\phi(h)$ be the deterministic encoder, $\mathbb{P}_\theta(z' \mid f_\phi(h), a) := \mathbb{P}_{\phi,\theta}$ be the predicted next latent distribution, and $\mathbb{Q}_\phi(z' \mid h, a)$ be real next latent distribution. The MMD with $k_d$ can be reduced to $\ell_2$ distance between the expectations of two distributions:

$$\mathrm{MMD}_{k_d}^2(\mathbb{P}_{\phi,\theta}, \mathbb{Q}_\phi; h, a) \tag{130}$$

$$= -\mathbb{E}_{z_1', z_2' \sim \mathbb{P}_{\phi,\theta}}[d(z_1', z_2')] + 2\mathbb{E}_{z_1' \sim \mathbb{P}_{\phi,\theta}, z_2' \sim \mathbb{Q}_\phi}[d(z_1', z_2')] - \mathbb{E}_{z_1', z_2' \sim \mathbb{Q}_\phi}[d(z_1', z_2')] \tag{131}$$

$$= -\mathbb{E}_{z_1', z_2' \sim \mathbb{P}_{\phi,\theta}}\left[\|z_1' - z_2'\|_2^2\right] + 2\mathbb{E}_{z_1' \sim \mathbb{P}_{\phi,\theta}, z_2' \sim \mathbb{Q}_\phi}\left[\|z_1' - z_2'\|_2^2\right] - \mathbb{E}_{z_1', z_2' \sim \mathbb{Q}_\phi}\left[\|z_1' - z_2'\|_2^2\right] \tag{132}$$

$$= 2\|\mathbb{E}_{z' \sim \mathbb{P}_{\phi,\theta}}[z' \mid h, a] - \mathbb{E}_{z' \sim \mathbb{Q}_\phi}[z' \mid h, a]\|_2^2. \tag{133}$$

Therefore, the MMD objective can be viewed as ZP with $\ell_2$ distance (*i.e.*, EZP). They also propose forward KL to instantiate ZP loss. Nevertheless, AIS (Subramanian et al., 2022) do not show experiment results on learning ZP. Instead, they and the follow-up works (Patil et al., 2022; Seyedsalehi et al., 2023) implement AIS by learning OP loss with MMD objectives, resulting in learning *observation-predictive* representations. Another follow-up work, Discrete AIS (Yang et al., 2022), learns ZP loss with $\ell_2$ objective in a discrete latent space, so that they can apply value iteration.

**TD-MPC (Hansen et al., 2022): $\phi_{Q^*}$ + RP + multi-step ZP with EMA $\ell_2$.** Temporal Difference learning for Model Predictive Control (TD-MPC) uses a planning horizon of 5 for the encoder objective and the latent value objective with TD learning. TD-MPC also uses MPC for action selection during inference. They find that learning ZP works better than learning OR or not learning ZP in the DM Control suite.

**TCRL (Zhao et al., 2023): RP + multi-step ZP with EMA** cos. Temporal consistency reinforcement learning (TCRL) simplifies TD-MPC (Hansen et al., 2022) by removing the planning component, replacing $\ell_2$ loss with cos loss, and detaching the encoder parameters during value function learning. They validate their approach on the state-based DM Control suite. Although the paper refers to TCRL as *minimalist* for learning representations, it is worth noting that TCRL is more complicated than our approach, as it still requires reward prediction and multi-step prediction.

### C.2 Observation-Predictive Representations

**PSR (Littman et al., 2001) and belief trajectory equivalence (Castro et al., 2009): Rec + multi-step OP and RP.** Predictive State Representation (PSR) aims to learn a history encoder $\phi$ and transition model $P_O$ such that

$$P(o_{t+1:t+k} \mid h_t, a_{t:t+k-1}) = P_O(o_{t+1:t+k} \mid \phi(h_t), a_{t:t+k-1}), \quad \forall h, a, o, \tag{134}$$

which implies multi-step OP (defined in Prop. 12) in POMDPs. The original PSR uses linear transition models. Follow-up work on PSRs (James et al., 2004) and belief trajectory equivalence introduce multi-step RP to PSR. In Castro et al. (2009), they show that single-step OP and RP do not necessarily imply multi-step OP and RP in POMDPs, summarized in Prop. 12. In this sense, PSR is a stronger notion of belief abstraction.

**Causal state representations (Zhang et al., 2019): Rec + OP + RP.** This work connects observation-predictive representations in POMDPs with causal state models in computational mechanics (Shalizi & Crutchfield, 2001). Specifically, they show that belief trajectory equivalence (Rec + multi-step OP and RP) (Castro et al., 2009) implies a causal state of a stochastic process, where RP means reward *distribution* prediction. The resulting abstract MDP is a causal state model or an $\epsilon$-machine, generating minimal sufficient representations for predicting future observations. In the implementation, they train a deterministic RNN encoder and a deterministic transition model to satisfy OP and RP conditions, and also train a latent Q-value function using Q-learning by freezing encoder parameters. Optionally, they also train a discretizer on the latent space in finite POMDPs.

**Belief-Based Methods (Hafner et al., 2019; 2020b; Han et al., 2020; Lee et al., 2020): RP + OR + ZP with online forward KL.** As a major approach to solving POMDPs, belief-based methods extends belief MDPs (Kaelbling et al., 1998) to deep RL through variational inference, deriving the encoder objective as ELBO. Let the latent variables are $z_{1:T}$, the world model $p(o_{1:T}, r_{1:T} \mid a_{1:T})$, and the posterior are $q(z_{1:T} \mid o_{1:T}, a_{1:T})$ with the factorization:

$$p(z_{1:T+1}, o_{1:T+1}, r_{1:T} \mid a_{1:T}) = p(z_1)p(o_1 \mid z_1) \prod_{t=1}^{T} p(r_t \mid z_t, a_t)p(z_{t+1} \mid z_t, a_t)p(o_{t+1} \mid z_{t+1}), \tag{135}$$

$$q(z_{1:T+1} \mid h_{T+1}) = \prod_{t=0}^{T} q(z_{t+1} \mid h_{t+1}) = \prod_{t=0}^{T} q(z_{t+1} \mid z_t, a_t, o_{t+1}), \tag{136}$$

where $h_{t+1} = (h_t, a_t, o_{t+1})$ in our notation. The log-likelihood has a lower bound:

$$\mathbb{E}_{h_{T+1}, r_{1:T}}[\log p_\theta(o_{1:T+1}, r_{1:T} \mid a_{1:T})] \tag{137}$$

$$= \mathbb{E}_{h_{T+1}, r_{1:T}}\left[\log \mathbb{E}_{q(z_{1:T+1}|h_{T+1})}\left[\frac{p(z_{1:T+1}, o_{1:T+1}, r_{1:T} \mid a_{1:T})}{q(z_{1:T+1} \mid h_{T+1})}\right]\right] \tag{138}$$

$$\geq \mathbb{E}_{h_{T+1}, r_{1:T}, z_{1:T} \sim q(\cdot|h_{T+1})}\left[\log \frac{p(z_{1:T}, o_{1:T+1}, r_{1:T} \mid a_{1:T})}{q(z_{1:T+1} \mid h_{T+1})}\right] \tag{139}$$

$$= \mathbb{E}_{h_{T+1}, r_{1:T}, z_{1:T+1} \sim q(h_{T+1})}\left[\sum_{t=0}^{T} \underbrace{\log p(o_{t+1} \mid z_{t+1})}_{(1)} + \underbrace{\log p(r_t \mid z_t, a_t)}_{(2)} - \underbrace{\log \frac{q(z_{t+1} \mid h_{t+1})}{p(z_{t+1} \mid z_t, a_t)}}_{(3)}\right]. \tag{140}$$

When $p, q$ are trained to optimal, the first term becomes OR condition and the second term becomes reward distribution matching that implies RP. The third term with expectation can be written as $\mathbb{E}_{z_t,h_{t+1}}[D_{\mathrm{KL}}(q(z_{t+1} \mid h_{t+1}) \parallel p(z_{t+1} \mid z_t, a_t))]$, which is exactly our practical forward KL objective Eq. 3 to learn ZP. From our relation graph (Fig. 1; Prop. 8), ZP + OR imply OP, thus belief-based methods aim to approximate observation-predictive representation (RP + OP). Normally, they use an online target in forward KL, because they have OR signals that can help prevent representational collapse. They also train encoders without maximizing returns.

We can also build the connections between OR and RP objectives and maximizing mutual information. Let $P(o, z)$ be the marginal joint distribution of observation and latent state at the same time-step, where $P(o', z') = \int P(o', z', h, a)dhda = \int P(h, a)P(o' \mid h, a)P(z' \mid h')dhda$. Consider,

$$\mathbb{I}(o'; z') = \mathbb{E}_{o',z' \sim P(o',z')}\left[\log \frac{P(o', z')}{P(o')P(z')}\right] \tag{141}$$

$$= \mathbb{E}_{o',z' \sim P(o',z')}\left[\log \frac{P(o' \mid z')}{P(o')}\right] \tag{142}$$

$$= \mathbb{E}_{o',z' \sim P(o',z')}[\log P(o' \mid z')] + \mathbb{H}(P(o')) \tag{143}$$

$$= \mathbb{E}_{h,a \sim P(h,a),o' \sim P(\mid h,a),z' \sim P(\mid h')}[\log P(o' \mid z')] + \mathbb{H}(P(o')). \tag{144}$$

Since the entropy term is independent of latent states, the OR objective in belief-based methods is **exactly** maximizing the $\mathbb{I}(o; z)$. Similarly, the RP objective in belief-based methods is exactly maximizing $\mathbb{I}(r; z)$.

**OFENet (Ota et al., 2020): $\phi_{Q^*}$ + OP.** Online Feature Extractor Network (OFENet) trains the state encoder using an auxiliary task of OP loss with $\ell_2$ distance. This is perhaps the most related algorithm to our Algo. 1 for learning $\Phi_O$. They show strong performance of their approach over model-free baseline in standard MuJoCo benchmark. Follow-up work (Lange et al., 2023) empirically find that $\phi_{Q^*}$ + RP slightly improves up model-free RL, but much worse than $\phi_{Q^*}$ + OP in MuJoCo benchmark.

**SAC-AE (Yarats et al., 2021): $\phi_{Q^*}$ + OR.** Soft Actor-Critic with AutoEncoder (SAC-AE) trains the state encoder with an auxliary task of OR loss with forward KL and also $\ell_2$-regularization. They detach the state encoder in policy objective. As in MDPs, OR implies OP (Prop. 9), SAC-AE also approximates observation-predictive representation.

### C.3    OTHER RELATED REPRESENTATIONS

**UNREAL (Jaderberg et al., 2016), Loss is its own Reward (Shelhamer et al., 2016).** These works make early attempts at auxiliary task design for RL. UNREAL trains recurrent A3C agent with several auxiliary tasks, including reward prediction (RP), pixel control and value function replay. Loss is its own Reward trains A3C agent with several auxiliary tasks, including reward prediction (RP), observation reconstruction (OR), inverse dynamics, and a proxy of forward dynamics (OP) that finds the corrupted observation from a time series. Among them, inverse dynamics condition in MDPs is that

$$\exists P_{\mathrm{inv}} : \mathcal{Z} \times \mathcal{Z} \to \Delta(\mathcal{A}), \quad s.t. \quad P_{\mathrm{inv}}(a \mid \phi(s), \phi(s')) = P(a \mid s, s'), \quad \forall s, a, s', \tag{145}$$

but this condition does not direct relation with forward dynamics (OP).

**VPN (Oh et al., 2017), MuZero (Schrittwieser et al., 2020): $\phi_{Q^*}$ + RP.** From Thm. 2, we know that $\phi_{Q^*}$ + RP is implied by $\phi_{Q^*}$ + ZP, thus this representation lies between $\phi_{Q^*}$ and $\phi_L$. Both VPN and MuZero learn the shared state encoder and latent model from maximizing the return and predicting rewards. Their policies are learned by the MCTS algorithm.

**E2C (Watter et al., 2015) and World Model (Ha & Schmidhuber, 2018): ZP + OR.** They are similar to belief-based methods, but remove the reward prediction loss from the encoder objective. Instead, reward signals are only accessible to latent policies or values.

**Contrastive representation learning in RL (CURL (Laskin et al., 2020), DRIML (Mazoure et al., 2020), ContraBAR (Choshen & Tamar, 2023)): $\phi_{Q^*}$ (RP) + weak OP (OR).** CURL ($\phi_{Q^*}$ + weak OR) introduces contrastive learning using the infoNCE objective (Oord et al., 2018) as an auxiliary task in MDPs. InfoNCE between positive and negative examples is shown to be a lower bound of mutual information between input and latent state variables (Poole et al., 2019). In MDPs, it is a lower bound of $\mathbb{I}(s; z)$, which corresponds to OR objectives Eq. 141. Therefore, CURL can be interpreted as maximizing a lower bound of OR.

DRIML ($\phi_{Q^*}$ + weak OP) proposes an auxiliary task named InfoMax in MDPs. In its single-step prediction variant, InfoMax maximizes the lower bound of $\mathbb{I}(z'; z, a)$ via the infoNCE objective. Similar to the analysis (Rakelly et al., 2021), by data processing inequality:

$$\mathbb{I}(z'; z, a) \leq \mathbb{I}(z'; s, a) \leq \mathbb{I}(s'; s, a), \tag{146}$$

$$\mathbb{I}(z'; z, a) \leq \mathbb{I}(s'; z, a) \leq \mathbb{I}(s'; s, a). \tag{147}$$

When all equalities hold (*e.g.* $\phi$ satisfies OR), these imply $z' \perp\!\!\!\perp s, a \mid z, a$ (ZP) and $s' \perp\!\!\!\perp s, a \mid z, a$ (OP).

ContraBAR (weak RP and weak OP) introduces infoNCE objectives to meta-RL, which requires incorporating reward signals into observations when viewed as POMDPs (Ni et al., 2022). Similar to DRIML, in its single-step prediction variant, the objective is to maximize the lower bound of mutual information of $\mathbb{I}(z'; z, a)$ where $z$ is a joint representation of state $s$ and reward $r$. As shown in the ContraBAR paper (Choshen & Tamar, 2023, Theorem 4.3), under certain optimality condition, the objective can lead to learning RP and OP conditions.

**Learning Markov State Abstraction (Allen et al., 2021): $\phi_{Q^*}$ + ZM.** From Prop. 6, we know that ZM is implied by ZP, thus representation lies between $\phi_{Q^*}$ and $\phi_L$. They show that ZM can be implied by inverse dynamics and density ratio matching in MDPs. Thus, they train on these two objectives as auxiliary losses.

**MICo (Castro et al., 2021): $\phi_{Q^*}$ + metric.** With a state encoder $\phi$, matching under Independent Coupling (MICo) defines a distance metric $U_\phi$ in the state space. For any pair of states $x, y \in \mathcal{S}$,

$$U_\phi(x, y) = |r_x^\pi - r_y^\pi| + \gamma \mathbb{E}_{x' \sim P_x^\pi, y' \sim P_y^\pi}[U_\phi(x', y')], \tag{metric}$$

where $r_x^\pi = \mathbb{E}_{a \sim \pi(\cdot|x)}[R(x, a)]$ and $P_x^\pi(x' \mid x) = \mathbb{E}_{a \sim \pi(\cdot|x)}[P(x' \mid x, a)]$. The metric $U_\phi$ is parameterized with

$$U_\phi(x, y) = \frac{1}{2}(\|\phi(x)\|_2^2 + \|\phi(y)\|_2^2) + \beta \arctan(\sqrt{1 - \cos(\phi(x), \phi(y))^2}, \cos(\phi(x), \phi(y))). \tag{148}$$

They learn the MICo metric by an auxiliary loss using mean squared error.

**Denoised MDPs (Wang et al., 2022): OR + RP + ZP in a factorized latent space (implying $\phi_{\pi^*}$, not $\phi_{Q^*}$).** This work aims to learn a state abstraction that ignores components that are either reward-irrelevant or uncontrollable. Such an abstraction can retain the optimal policy while not necessarily preserving optimal value functions. Consequently, denoised MDPs can be conceptualized as approximating $\phi_{\pi^*}$. Technically, the authors postulate that the latent state of an MDP is composed of elements $(x, y, z)$ where the transition in $y$ is independent of actions. Additionally, the reward function $r(s)$, independent of $z$, is decomposed into $r_x(x)$ and $r_y(y)$. Thus, the optimal policy (though not its value) can only depend on the latent state component $x$.

In practice, they introduce variational objectives to learn the encoder $p(x, y, z \mid s)$ with observation reconstruction (OR), reward prediction (RP), and next latent state prediction (ZP) using online forward KL divergences. The structure of the latent state space helps the partial encoder $p(x \mid s)$ to gravitate towards $\phi_{\pi^*}$ abstraction, despite the absence of a theoretical guarantee. Finally, they use model-free RL to optimize a policy on the latent $x$ space.

**TD7 (Fujimoto et al., 2023): ZP with detached $\ell_2$.** TD7 algorithm is introduced for addressing MDPs and evaluated on the MuJoCo benchmark. TD7 learns a state encoder using ZP loss with detaching the next latent states, which performs better than EMA version. They use $\ell_2$ loss and normalize latent states by average $\ell_1$ norm, which performs better than cos loss and other normalization

methods. They find that training with ZP loss only is slightly better than training with ZP + RP, and much better than end-to-end training (ZP + $\phi_Q^*$). Lastly, it is noteworthy that in TD7, the critic not only takes a state $s$ and an action $a$ as inputs, but also the latent state $f_\phi(s)$ and the predicted next latent state $g_\theta(f_\phi(s), a)$, which is named as *state-action embedding* in the paper.

# D   ADDITIONAL DISCUSSION

## D.1   MOTIVATING OUR HYPOTHESES

Here we provide our motivation for our hypotheses shown in Sec. 5.

- **Motivating the sample efficiency hypothesis.** The performance of deep RL algorithms is notably influenced by task structure, and no single algorithm consistently outperforms others across all tasks (Wang et al., 2019; Li et al., 2022; Ni et al., 2022). Common wisdom suggests that certain algorithms excel in specific types of tasks (Mohan et al., 2023). For instance, self-predictive representations are often effective in distracting tasks (Zhang et al., 2020; Zhao et al., 2023), while observation-predictive representations typically perform well in sparse-reward scenarios (Zintgraf et al., 2021; Zhang et al., 2021). However, these methods often incorporate additional complexities like intrinsic rewards and metric learning or are primarily evaluated in pixel-based tasks.

  Given these considerations, we propose the use of our minimalist algorithm as a tool to focus solely on the impact of representation learning in vector-based tasks. This approach aims to provide a clearer understanding of how different representation learning strategies affect sample efficiency in various task structures (including popular standard benchmarks), without the confounding factors present in more complex algorithms or environments.

- **Motivating the distraction hypothesis.** The belief that algorithms predicting observations tend to underperform in distracting tasks is supported by several studies (Zhang et al., 2020; Okada & Taniguchi, 2021; Fu et al., 2021; Deng et al., 2022). The challenge arises from the need for these models to predict every detail of observations, including irrelevant features, which can be extremely difficult due to randomness and high dimensionality. Similar to the motivation in our sample efficiency hypothesis, prior works primarily focus on complex algorithms evaluated in pixel-based tasks, often with real-world video backgrounds as distractors.

  Considering these, we propose a shift towards studying the impact of distractions using a minimalist algorithm in simpler, configurable environments. This approach aims to isolate and understand the specific effects of distracting elements in tasks, providing a more straightforward and controlled setting for analysis.

- **Motivating the end-to-end hypothesis.** According to our Thm. 2, learning an encoder end-to-end with the auxiliary task of ZP can implicitly learn the reward prediction conditioned on its optimality, potentially making it comparable to the phased learning. However, this is a theoretical prediction and may not necessarily translate to practical scenarios, particularly considering that RL agents rarely achieve global optima. On the other hand, prior works (Schwarzer et al., 2020; Ghugare et al., 2022) have shown the success of the end-to-end learning, but these algorithms incorporate other moving components (multi-step prediction, intrinsic rewards) and are not directly applicable to POMDPs.

  These limitations underscore the importance of empirically testing whether the benefits of the end-to-end learning extend to POMDPs when employing a minimalist approach in representation learning.

- **Motivating the ZP objective hypothesis.** Our Prop. 1 and Prop. 2 suggest that it suffices to use $\ell_2$ objective in deterministic tasks while KL divergences might be more effective in stochastic ones. However, these are theoretical assumptions that do not fully account for the complexities of the learning process. Additionally, most existing research tends to focus on a single objective type in deterministic settings (as summarized in Table 1), leaving the performance of alternative objectives, particularly in stochastic tasks, largely unexplored. A notable exception is AIS (Subramanian et al., 2022) which discusses various ZP objectives but lacks practical evaluation on them.

  These gaps in the literature motivate us to undertake a thorough comparison of these ZP objectives in practical settings.

- **Motivating the ZP stop-gradient hypothesis.** Our Thm. 3 suggests that applying stop-gradient to ZP targets could help mitigate representational collapse. However, this prediction is based on linear

models without incorporating RL loss, which is a significant departure from deep RL scenarios. While most prior studies focus on one type of ZP target without delving into collapse issues (as summarized in Table 1), SPR (Schwarzer et al., 2020) is an exception, comparing online and EMA encoders in Atari tasks. Nonetheless, SPR's analysis focuses on return performance and lacks direct evidence of representational collapse.

Addressing these research gaps, we aim to conduct an extensive comparison of ZP targets in both MDPs and POMDPs. Our analysis includes providing direct evidence through the estimation of representational rank.

### D.2 LIMITATIONS AND CONCLUSION

**Limitations.** The limitations of our work can be divided into theoretical and empirical aspects. On the theoretical side, although we show a continuous-time analysis of auxiliary learning dynamics with linear models in Thm. 3, we do not provide a convergence analysis for the joint optimization of RL and auxiliary losses and the results may not hold beyond linear assumption. On the empirical side, our experiment scope does not cover more complicated domains that require pixel-based observations.

**Conclusion.** This work has offered a principled analysis of state and history representation learning in reinforcement learning, bridging the gap between various approaches. Our unified view and analysis of self-predictive learning also inspire a minimalist RL algorithm for learning self-predictive representations. Extensive empirical studies in benchmarks across standard MDPs, distracting MDPs, and sparse-reward POMDPs, validate most of our hypotheses suggested by our theory.

## E EXPERIMENTAL DETAILS

### E.1 SMALL SCALE EXPERIMENTS TO ILLUSTRATE THM. 3

In this section, we discuss the details of the experiments used to explore the empirical effects of using stop-gradient to detach the ZP target in the self-predictive loss. First, we discuss the details shared between both domains and then discuss domain-specific details.

We learn on data obtained by rolling out 10 trajectories under a fixed, near-optimal policy starting from a random state. Trajectories are followed until termination or until 200 transition have been observed, whichever happens first. The encoder, $\phi \in \mathbb{R}^{k \times 2}$ where $k$ is the number of observed features, is updated using full gradient descent with a small learning rate, $\alpha = 0.01$, for 500 steps. At every 10 steps, the absolute cosine similarity between the 2 columns of $\phi$ is computed, i.e., $f(x, y) = |x^\top y|/(||x||_2||y||_2)$ and the results are plotted in Fig. 2. The optimal transition model $\theta^* = \begin{bmatrix} \theta_z^{*\top} & \theta_a^{*\top} \end{bmatrix}^\top$ is solved using singular value decomposition and the Moore-Penrose inverse to minimize the linear least-squares objective:

$$\left\| \begin{bmatrix} \phi^\top S & A \end{bmatrix} \begin{bmatrix} \theta_z \\ \theta_a \end{bmatrix} - \widetilde{\phi}^\top S' \right\|_2, \tag{149}$$

where $S$ and $S'$ are matrices with each row corresponding to the sampled states (histories) and next states (histories), respectively, and, similarly, $A$ is a row-wise matrix of the sampled actions. The $\widetilde{\phi}$ is set as $\phi$ in online target, or $\bar{\phi}$ in detached target and EMA target where the Polyak step size $\tau = 0.005$. To avoid numerical issues, singular values close to zero are discarded according to the default behavior of JAX's (?) `jax.numpy.linalg.lstsq` method when using `float32` encoding.

**Mountain car (Moore, 1990).** We follow the dynamics and parameters used in (Sutton & Barto, 2018, Example 10.1). We encode states using a $10 \times 10$ uniform grid of radial basis function (RBF), e.g., $f_i(s) = \exp(-(s - c_i)^\top \Sigma^{-1}(s - c_i))$ for an RBF centered on $c_i$, and with a width corresponding to $0.15$ of the span of the state space. Specifically, $\Sigma$ is diagonal and normalizes each dimension such that the width of the RBF covers $0.15$ in each dimension. As a result, the total number of features $k = 100$. Actions are encoded using one-hot encoding and $|\mathcal{A}| = 3$. The policy used to generate data is an energy pumping policy which always picks actions that apply a force in the direction of the velocity and applies a negative force when the speed is zero.

**Load-unload (Meuleau et al., 2013).** Load-unload is a POMDP with 7 states arranged in a chain. There are 2 actions which allow the agent to deterministically move left or right along the chain, while attempting to move past the left-most or right-most state results in no movement. There are three possible observations which deterministically correspond to being in the left-most state, the right-most state or in any one of the 5 intermediate states. Observations and actions are encoded using one-hot encodings. The agent's state correspond to the history of observation and actions over a fixed window of size 20 with zero padding for a total of $k = 98$ features ($k = 20 \times 3 + 19 \times 2$). Finally, the policy used to generate trajectories is a stateful policy that repeats the last action with probability 0.8 and always starting with the move-left action.

### E.2 MDP EXPERIMENTS IN SEC. 5.1 AND SEC. 5.2

**Standard MuJoCo in Sec. 5.1.** This is a popular continuous control benchmark from OpenAI Gym (Brockman et al., 2016). We evaluate on Hopper-v2 (11-dim), Walker2d-v2 (17-dim), HalfCheetah-v2 (17-dim), Ant-v2 (111-dim), and Humanoid-v2 (376-dim), where the numbers in the brackets are observation dimensions.

**Distracting MuJoCo in Sec. 5.2.** We follow Nikishin et al. (2022) to augment the state space with a distracting dimension in Hopper-v2, Walker2d-v2, HalfCheetah-v2, and Ant-v2. The number of distractors varies from $2^4 = 16$ to $2^8 = 256$. Therefore, the largest observation dimension is $256 + 111 = 367$ in distracting Ant-v2. The distractors follow i.i.d. standard Gaussian $\mathcal{N}(0, I)$.

Our algorithm setup in Sec. 5.1 and Sec. 5.2 largely follows the code of ALM(3) (Ghugare et al., 2022)[16]. The original ALM paper also introduces an ablation of the method, "ALM-no-model", which uses model-free RL (rather than SVG) to update the actor parameters. This ablation is structurally similar to our method, which similarly avoids using a model. However, ALM-no-model still employs a reward model and a latent model for learning representations, although not for updating policy.

Below, we compare ALM and our minimalist $\phi_L$ implementation. We show ablation results comparing our method and ALM variants in Sec. G.

**Differences between our minimalist $\phi_L$ (with reverse KL and EMA targets) and ALM.**

- **Reward model**: we remove reward models from both ALM(3) and ALM-no-model. It should be noted that although ALMs learn reward models, they do not update their encoders through reward prediction loss.
- **Encoder objective**: our state encoder ($\phi$) is updated by Eq. 5. Given a probabilistic encoder $\mathbb{P}_\phi(z \mid s)$ and a probabilistic latent model $\mathbb{P}_\theta(z' \mid z, a)$ for MDPs, and the latent state $z_\phi \sim \mathbb{P}_\phi(z \mid s)$, we formulate our encoder objective for a data tuple $(s, a, r, s')$ as follows:

$$\min_\phi \quad (Q_\omega(z_\phi, a) - Q^{\text{tar}}(s, a, s', r))^2 - Q_\omega(z_{\overline{\phi}}, \pi_\nu(z_\phi)) + D_{\text{KL}}(\mathbb{P}_\theta(z' \mid z_\phi, a) \,||\, \mathbb{P}_{\overline{\phi}}(z' \mid s')). \tag{150}$$

In contrast, **ALM-no-model** employs a more complicated objective to train the state encoder $\phi$. It performs a 1-step rollout with the reward model $R_\mu(z, a)$ without a discount factor, and modify the stop-gradients on latent states within the $Q$-value. Given a data tuple $(s, a, s')$, the objective is

$$\min_\phi \quad -R_\mu(z_\phi, a) - \mathbb{E}_{z'_{\phi,\theta} \sim \mathbb{P}_\theta(\cdot \mid z_\phi, a)} \left[ Q_\omega(z'_{\phi,\theta}, \pi_\nu(z'_{\overline{\phi},\theta})) \right] + D_{\text{KL}}(\mathbb{P}_\theta(z' \mid z_\phi, a) \,||\, \mathbb{P}_{\overline{\phi}}(z' \mid s')), \tag{151}$$

where they eliminate the mean-squared TD loss and maximize the $Q$-value through its latent states rather than actions, as done in our objective. **ALM(3)** extends ALM-no-model by implementing a 3-step rollout in the encoder objective.

To isolate the design of stop-gradients in $Q$-value and mean-squared TD error, we introduce **ALM(0)** that lies between ALM-no-model and ours with the 0-step objective:

$$\min_\phi \quad -Q_\omega(z_\phi, \pi_\nu(z_{\overline{\phi}})) + D_{\text{KL}}(\mathbb{P}_\theta(z' \mid z_\phi, a) \,||\, \mathbb{P}_{\overline{\phi}}(z' \mid s')). \tag{152}$$

---

[16]https://github.com/RajGhugare19/alm

Table 3: **Hyperparameters used in Markovian agents in standard and distracting MuJoCo.**

| Hyperparameter | Value |
|---|---|
| Discount factor ($\gamma$) | 0.99 |
| Warmup steps | 5000 |
| Target network update rate ($\tau$) | 0.005 |
| Replay buffer size | $10^6$ for Humanoid-v2 and $10^5$ otherwise |
| Batch size | 512 |
| Learning rate | 0.0001 |
| Max gradient norm | 100 |
| Latent state dimension | 50 |
| Exploration stddev. clip | 0.3 |
| Exploration stddev. schedule | $\text{linear}(1.0, 0.1, 100000)$ |
| Auxiliary loss coefficient ($\lambda$) | 1.0 for ZP-FKL, ZP-RKL and OP, and 10.0 for ZP-$\ell_2$ |

- **Actor objective**: our algorithm share the same actor objective with ALM-no-model and ALM(0), compared to ALM(3) which uses SVG with a 3-step rollout (Heess et al., 2015) and additional intrinsic rewards (*i.e.* the negative reverse KL divergence term; see Eq. 8 in ALM paper for details).

**Implementation details for our minimalist algorithm learning $\phi_L$, and learning $\phi_O$ and $\phi_{Q^*}$.** We follow the exact implementation of the network architectures in ALM(3). The encoder, actor, and critic are parameterized as 2-layer neural networks with 512 hidden units. The latent transition model (only used in learning $\phi_L$) and observation predictor (only used in learning $\phi_O$) are parameterized as 2-layer networks with 1024 hidden units. The probabilistic encoder, latent model and decoder output a Gaussian distribution with a diagonal covariance matrix. We apply layer normalization (Ba et al., 2016) after the first layer of the critic network. We use ELU activation (Clevert et al., 2015) and Adam optimizers (Kingma & Ba, 2015) for all networks.

We enumerate the values of our hyperparameters in Table 3. If a hyperparameter is shared with ALM(3), we maintain the same value as that used in ALM(3) (Ghugare et al., 2022, Table 3).

### E.3 POMDP EXPERIMENTS IN SEC. 5.3

**MiniGrid in Sec. 5.3.** This is a widely-used discrete gridworld benchmark from Farama foundation (Chevalier-Boisvert et al., 2018; 2023). In this benchmark, an agent has a first-person view to navigate a 2D gridworld with obstacles (*e.g.*, walls and lava). Some tasks require the agent to pick up keys and open doors to navigate to the goal location. The agent's observations are symbolic (not pixel-based) with a size of $7 \times 7 \times 3$ where $7 \times 7$ is the spatial field of view, and the 3 channels encode different semantics. The action space is discrete with 7 options: turn left, turn right, move forward, pick up, drop, toggle, and done. Tasks are goal-oriented; the episode terminates immediately when the agent reaches the goal, or times out after a maximum of $T$ steps. Rewards are designed to encourage fast task completion. A successful episode yields a reward of $1 - 0.9 * H/T \in [0.1, 1.0]$ at the terminal step, where $H$ denotes the total steps. Failed episodes result in a reward of $0.0$.

We select 20 tasks in MiniGrid, following the recent work RQL-AIS (Seyedsalehi et al., 2023). All of these tasks require memory in an agent. The tasks are grouped as follows:

- **SimpleCrossing** (4 tasks): SimpleCrossingS9N1, SimpleCrossingS9N2, SimpleCrossingS9N3, SimpleCrossing11N5

- **LavaCrossing** (4 tasks): LavaCrossingS9N1, LavaCrossingS9N2, LavaCrossingS9N3, LavaCrossing11N5

- **Unlock** (2 tasks): Unlock, UnlockPickup

- **DoorKey** (3 tasks): DoorKey-5x5, DoorKey-6x6, DoorKey-8x8

- **KeyCorridor** (3 tasks): KeyCorridorS3R1, KeyCorridorS3R2, KeyCorridorS3R3

- **ObstructedMaze** (2 tasks): ObstructedMaze-1Dl, ObstructedMaze-1Dlh

- **MultiRoom** (2 tasks): MultiRoom-N2-S4, MultiRoom-N4-S5

In each group, we arrange the tasks by increasing level of difficulty. Please refer to the MiniGrid website[17] for detailed descriptions. Note that while RQL-AIS also evaluates the RedBlueDoors tasks, we have omitted them from our selection as they are MDPs.

**Implementation details on algorithms.** We adopt the RQL-AIS codebase[18] for our implementation of a non-distributed version of R2D2 (Kapturowski et al., 2018). We retain their exact implementation and hyperparameters for R2D2, which includes a recurrent replay buffer with uniform sampling, a $50$-step burn-in period, a $10$-step rollout, and a stepsize of $5$ for multi-step double Q-learning. The only difference is that we replace the periodic hard update of target networks with a soft update, to align with our EMA setting.

We implement our end-to-end approaches based on R2D2. Minimalist $\phi_L$ introduces a single auxiliary task of ZP; while $\phi_O$ adds a single auxiliary task of OP. Both use deterministic $\ell_2$ loss. We normalize the loss coefficient $\lambda$ Eq. 5 by the output dimension (*i.e.*, 128 for ZP loss and 147 for OP loss) to balance with Q-learning scalar loss. We tune the normalized $\lambda$ between $(0.01, 0.03, 0.3, 1.0, 3.0, 10.0, 100.0)$ in SimpleCrossing and LavaCrossing tasks. We find that $0.01$ works best for OP and $1.0$ best for ZP.

Furthermore, we also implement the phased approaches based on R2D2. Both $\phi_L$ (RP + ZP) and $\phi_O$ (RP + OP) freeze the encoder parameters during Q-learning. We introduce the coefficient $\alpha$ multiplied to ZP or OP loss to integrate with RP loss. All three losses use deterministic $\ell_2$ objectives. We normalize $\alpha$ to balance reward scalar loss and tune it between $(0.01, 0.1, 0.3, 1.0, 3.0, 10.0)$ in SimpleCrossing and LavaCrossing tasks. We find $1.0$ works best for both RP + ZP and RP + OP.

We enumerate the values of our hyperparameters in Table 4. If a hyperparameter is shared with R2D2 implemented by RQL-AIS, we maintain the same value as that used in RQL-AIS paper (Seyedsalehi et al., 2023, Table 3). Our network architecture exactly follows RQL-AIS (see their Appendix F).

Lastly, it is important to highlight the distinction between our implementation of RP + OP and the original RQL-AIS approach (Seyedsalehi et al., 2023). Both approaches aim to learn $\phi_O$ in a phased manner with the same architecture. The main differences are:

- RQL-AIS employs a pre-trained autoencoder to compress the $147$-dimensional observations into $64$-dimensional latent representations. Then RQL-AIS trains their agent using latent representations while keeping the autoencoder parameters frozen. In contrast, our RP + OP implementation removes the autoencoder and instead directly predicts raw observations.

- RQL-AIS uses MMD loss for observation prediction, which we show is equivalent to learning EZP condition (see our discussion in Sec. C on AIS). Thus, in RP + OP implementation, we replace the MMD loss with an $\ell_2$ loss.

- The loss coefficient $\alpha$ is set to $0.5$ in RQL-AIS while $0.1/147$ in our implementation.

- We use soft update on target Q network to align with our other implementations.

Despite these implementation differences, we find that our RP + OP implementation performs *similarly* to RQL-AIS across the 20 tasks.

### E.4 EVALUATION METRICS

We evaluate the **episode return** by executing the deterministic version of the actor to compute the undiscounted sum of rewards.

We estimate the **rank** of a batch of latent states by calling `torch.linalg.matrix_rank(atol, rtol)` function in PyTorch (Paszke et al., 2019). This function calculates the number of singular values that are greater than $\max(\texttt{atol}, \sigma_1 * \texttt{rtol})$ where $\sigma_1$ is the largest singular value. In MDP experiments, the batch has a size of $(512, 50)$ with `atol=1e-2, rtol=1e-2`. In POMDP experiments, the batch has a size of $(256 * 10, 128)$

---

[17]https://minigrid.farama.org/environments/minigrid/
[18]https://github.com/esalehi1996/POMDP_RL

Table 4: **Hyperparameters used in recurrent agents in MiniGrid.**

| Hyperparameter | Value |
|---|---|
| Discount factor ($\gamma$) | 0.99 |
| Number of environment steps | $4 * 10^6$ |
| Target network update rate ($\tau$) | 0.005 |
| Replay buffer size | full |
| Batch size | 256 |
| Learning rate | 0.001 |
| Latent state dimension | 128 |
| Epsilon greedy schedule | exponential$(1.0, 0.05, 400000)$ |
| R2D2 sequence length | 10 |
| R2D2 burn-in sequence length | 50 |
| $n$-step TD | 5 |
| Training frequency | every 10 environment steps |
| Auxiliary loss coefficient ($\lambda$) | 1.0/128 for ZP and 0.01/147 for OP |
| Loss coefficient for phased training ($\alpha$) | 1.0/128 for RP + ZP and 1.0/147 for RP + OP |

with `atol=1e-3, rtol=1e-3`, where we reshape the 3D tensor of $(256, 10, 128)$ size into 2D matrix.

Each algorithm variant of the experiments is conducted across 12 individual runs in MDPs and 9 individual runs in POMDPs.

We employ the Rliable library (Agarwal et al., 2021) to compute the IQM and its CI for the aggregated curves (Fig. 6). Essentially, IQM is the $25\%$ trimmed mean over the data on 20 tasks with 9 seeds, *i.e.*, 180 runs.

### E.5  COMPUTATIONAL RESOURCES

It requires around 1.5 days for us to train our algorithm in a (distracting) MuJoCo task for 1.5M environment steps with 3 runs executed in parallel. The 3 runs share a single A100 GPU and utilize 3 CPU cores.

On the same machine, training cost (in secs) per update for Ant-v2 is as follows: model-free agents take around 0.032s, self-predictive and observation-predictive agents with $\ell_2$ objective take around 0.036s (13% more), self-predictive and observation-predictive agents with KL objective take around 0.038s (19% more), ALM(3) agent takes around 0.058s (81% more). The brackets show the percentage increase compared to model-free agents.

It requires around 0.5 days for us to train our algorithm in a MiniGrid task for 4M environment steps with 3 runs executed in parallel. The 3 runs share a single V100 GPU and utilize 3 CPU cores.

## F  ARCHITECTURE AND CODE

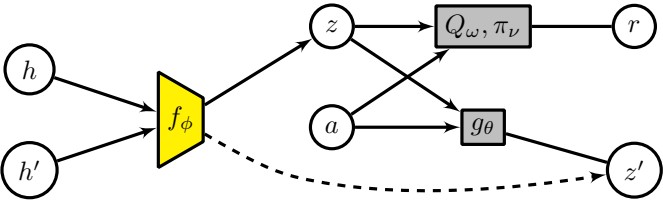

Figure 9: **Architecture of our minimalist $\phi_L$ algorithm.** The dashed edge indicates the stop-gradient operator; the undirected edges indicate learning from grounded signals of rewards or next latent states.

Besides Algo. 1, we provide a pseudocode written in PyTorch syntax (Paszke et al., 2019) in Algo. 2. Fig. 9 shows our architecture. We also have open-sourced our code at https://github.com/twni2016/self-predictive-rl.

---

**Algorithm 2** Our loss function for learning self-predictive representation in PyTorch syntax

```python
def loss(hist, act, next_obs, rew):
    # hist:(B,T,O+A), act:(B,A), next_obs:(B,O), rew:(B,1)
    from torch import cat; from copy import deepcopy

    # Encode histories into latent states
    h_enc = Encoder(hist) #(B,Z)
    next_hist = cat([hist, cat([act, next_obs], dim=-1)], dim=1) #(B,T+1,O+A)
    next_h_enc_tar = Encoder_Target(next_hist) #(B,Z)

    # Compute RL loss
    td_tar = rew + gamma * Critic_Target(next_h_enc_tar, Actor(next_h_enc_tar))
    critic_loss = ((Critic(h_enc, act)- td_tar.detach())**2).mean()
    actor_loss = -deepcopy(Critic)(h_enc.detach(), Actor(h_enc)).mean()

    # Compute ZP loss
    zp_loss = ((Latent_Model(h_enc, act) - next_h_enc_tar)**2).sum(-1).mean()

    return critic_loss + actor_loss + zp_coef * zp_loss
```

---

## G   ADDITIONAL EMPIRICAL RESULTS

**Ablation studies comparing our minimalist $\phi_L$ to ALM variants.** Fig. 10 shows the comparison between our method and several ALM variants (introduced in Sec. E.2). Both ALM(3) and ALM-no-model have similar encoder objectives (1-step versus 3-step) but differ in actor objective (model-free TD3 versus model-based SVG); their performance is similar except for the Humanoid-v2 tasks, suggesting that the model-based actor update is most useful on higher-dimensional tasks. Comparing ALM-no-model and ALM(0), which only differ in encoder objective (1-step versus 0-step), we see that ALM(0) performs notably worse. This suggests that the use of stop-gradients in $Q$-value and the omission of mean-squared TD-error in Eq. 152 and Eq. 151 might be problematic, although this issue can be considerably mitigated by the 1-step variant. Finally, our minimalist $\phi_L$ Eq. 150 performs comparably to ALM-no-model on most tasks and significantly outperforms ALM(0). These results suggest that our method can achieve the benefits of a 1-step rollout without having to unroll a model; however, a method that uses a 3-step rollout can sometimes achieve better results.

**Additional results on ZP targets in standard MuJoCo.** Fig. 11 shows the performance and the estimated representational rank for the ZP KL divergences (FKL and RKL). Similar to findings in $\ell_2$ objective (Fig. 4), we notice significantly lower returns when removing the stop-gradient version (Detached and EMA). Surprisingly, this decreased performance does not seem to be caused by dimensional collapse; on most tasks, the online version of the KL objective does not suffer from dimensional collapse observed for the $\ell_2$ objective. These findings suggest that our estimated representational rank may not be correlated with expected returns.

**Ablation studies on ZP loss in standard MuJoCo.** Fig. 12 shows the ZP losses ($\ell_2$ loss, FKL loss, RKL loss) for each ZP objective within our minimalist $\phi_L$ algorithm. We include results for the online, detached, and EMA targets. As expected, online ZP targets directly minimize ZP losses, thus reaching much lower ZP loss values. However, a lower ZP loss value does not imply higher returns, since the agent needs to balance the RL loss and ZP loss. In future work, we aim to explore strategies to effectively decrease ZP loss without compromising the performance of the stop-gradient variant.

**Failed experiments in standard MuJoCo.** We did not explore the architecture design and did little hyperparameter tuning on our algorithm. Nevertheless, we observed two failure cases. To match the assumption of Thm. 3 that the gradient w.r.t. the latent transition parameters $\theta$ reaches zero, we experimented with higher learning rates (0.1, 0.01, 0.001) for updating the parameters $\theta$ in MuJoCo tasks. Yet, we did not observe any performance increase compared to the default learning rate (0.0001). Secondly, inspired by the findings in Fig. 12, we tried a constrained optimization on auxiliary task to adaptively update the loss coefficient using gradient descent-ascent for the stop-gradient version. However, this resulted in a significant performance decline without an explicit decrease in ZP loss values.

**Full per-task curves in distracting MuJoCo.** Fig. 13 shows all learning curves in distracting MuJoCo.

**Full per-task curves in MiniGrid.** Fig. 14 shows all learning curves in MiniGrid tasks. Minimalist $\phi_L$ (ZP) is better than model-free RL (R2D2) in 8 of 20 tasks, and similar in the others except for a single task. On the other hand, $\phi_O$ (OP) is better than model-free RL (R2D2) in 10 tasks, with the other tasks being identical. Since MiniGrid tasks are deterministic without distraction and the observation is not high-dimensional, $\phi_O$ (OP) outperforming $\phi_L$ (ZP) in 7 tasks is expected. The end-to-end $\phi_L$ (ZP) surpasses its phased counterpart (RP + ZP) in 14 tasks, with the rest tasks being the same. The end-to-end $\phi_O$ (OP) is better than its phased counterpart (RP + OP) in 7 tasks, but falls short in 4 tasks. These findings underline the efficacy of the end-to-end approach to learning $\phi_L$ over the phased approach.

Fig. 15 shows all matrix rank curves in MiniGrid tasks. Across all 20 tasks, online ZP targets consistently have the *lowest* matrix rank, aligned with our prediction from Thm. 3. However, while Thm. 3 shows that both detached and EMA targets avoid collapse in *linear* setting, we observe that detached targets severely collapse in 3 tasks, a phenomenon absent with EMA targets. This prompts further theoretical investigation in a *nonlinear* context. As expected, OP consistently achieves the highest rank compared to ZP and R2D2, since OP learns the finest abstraction.

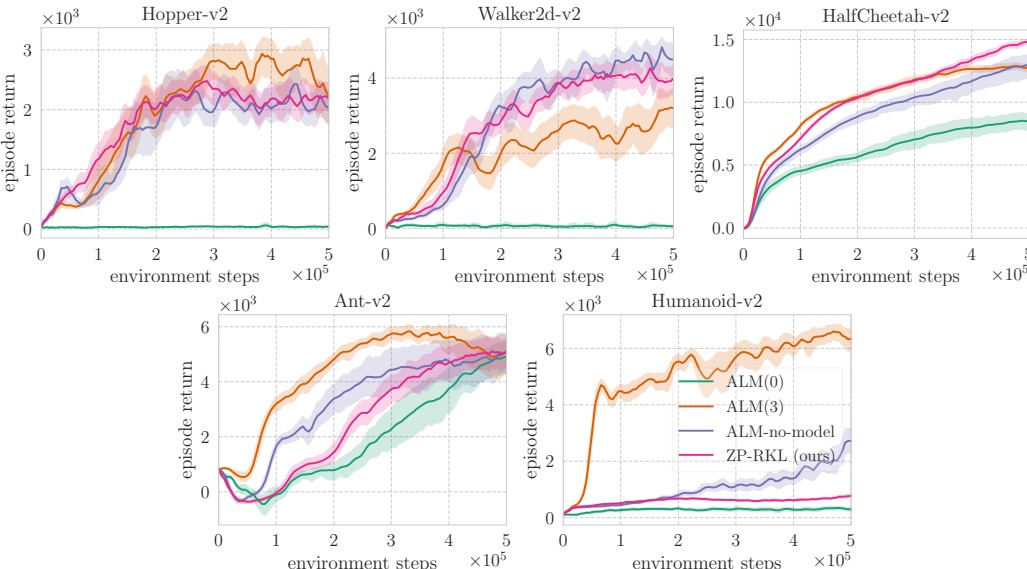

Figure 10: Ablation studies on ALM variants (ALM(3), ALM-no-model, ALM(0)) and our minimalist $\phi_L$ (ZP-RKL with EMA targets).

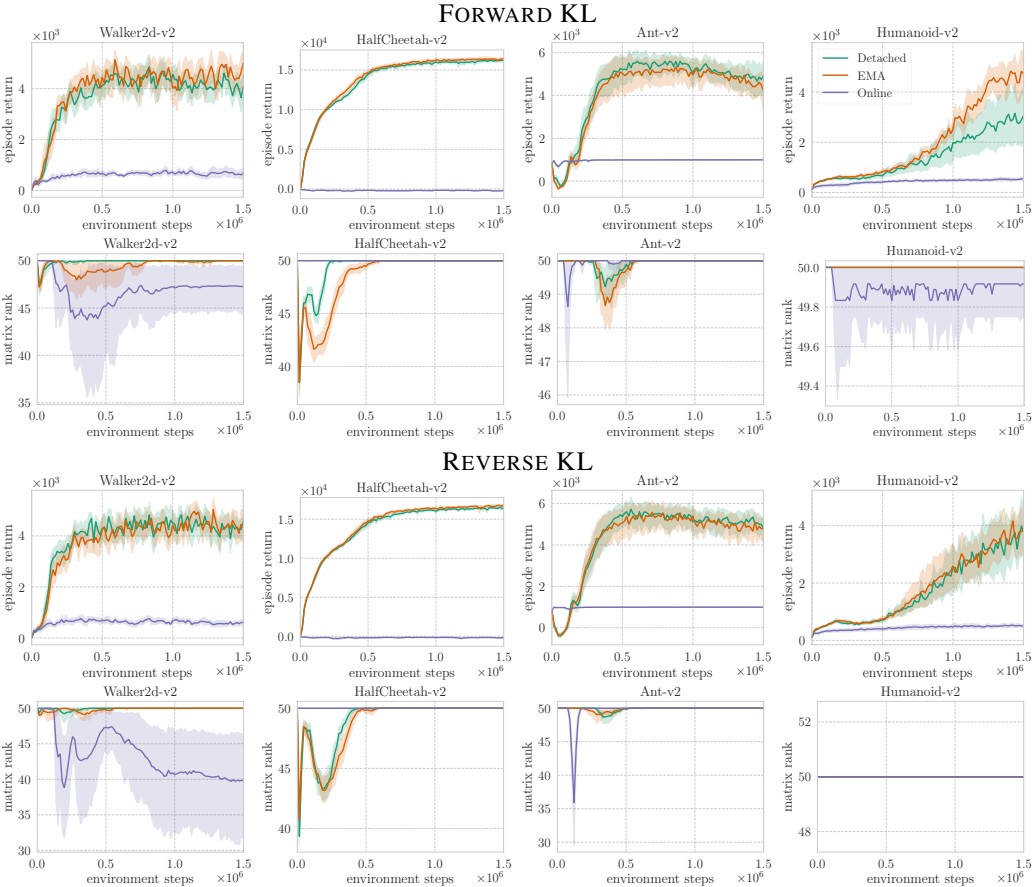

Figure 11: **Representation collapse is less severe with online targets for KL objectives.** On four benchmark tasks, we observe that using the online ZP target results in lower returns. Rows 1 and 2 show results for the forward KL; rows 3 and 4 show result for the reverse KL.

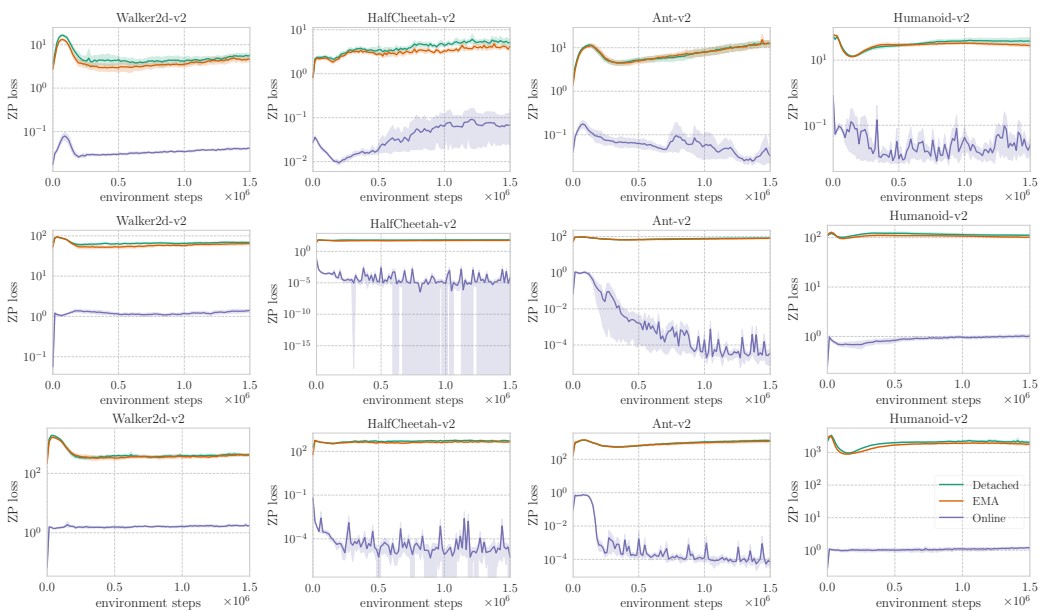

Figure 12: **Online ZP targets reach much smaller values on ZP objectives than stop-gradient ZP targets in standard MuJoCo.** Top row: ZP with $\ell_2$ objective; Middle row: ZP with FKL objective; Bottom row: ZP with RKL objective.

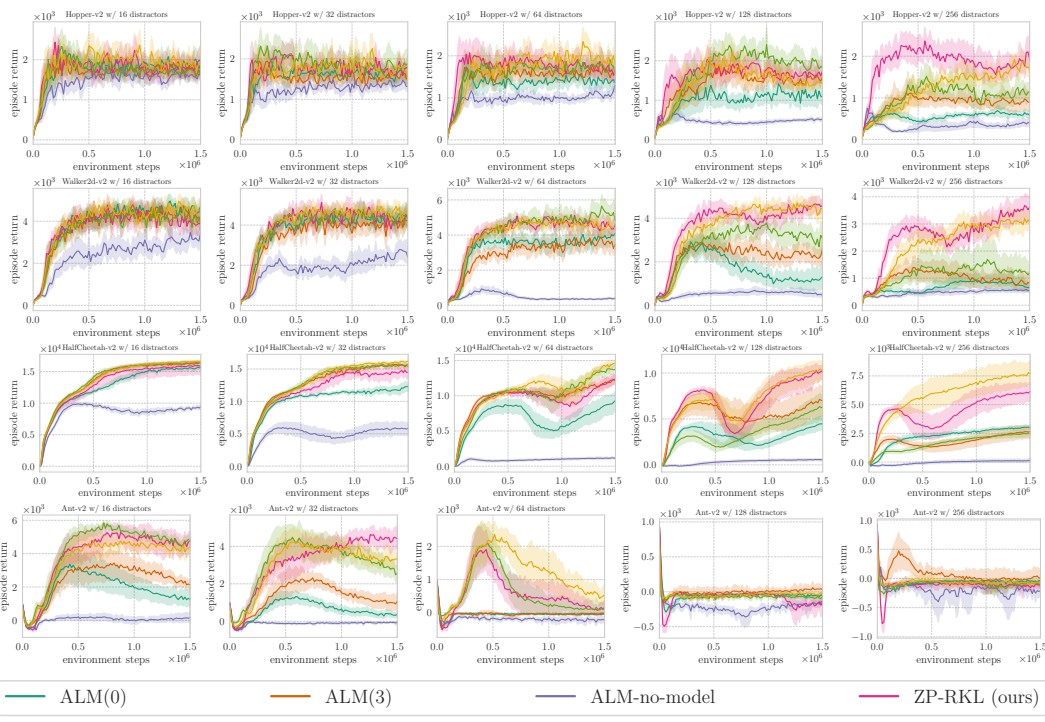

Figure 13: Full learning curves on distracting MuJoCo benchmark.

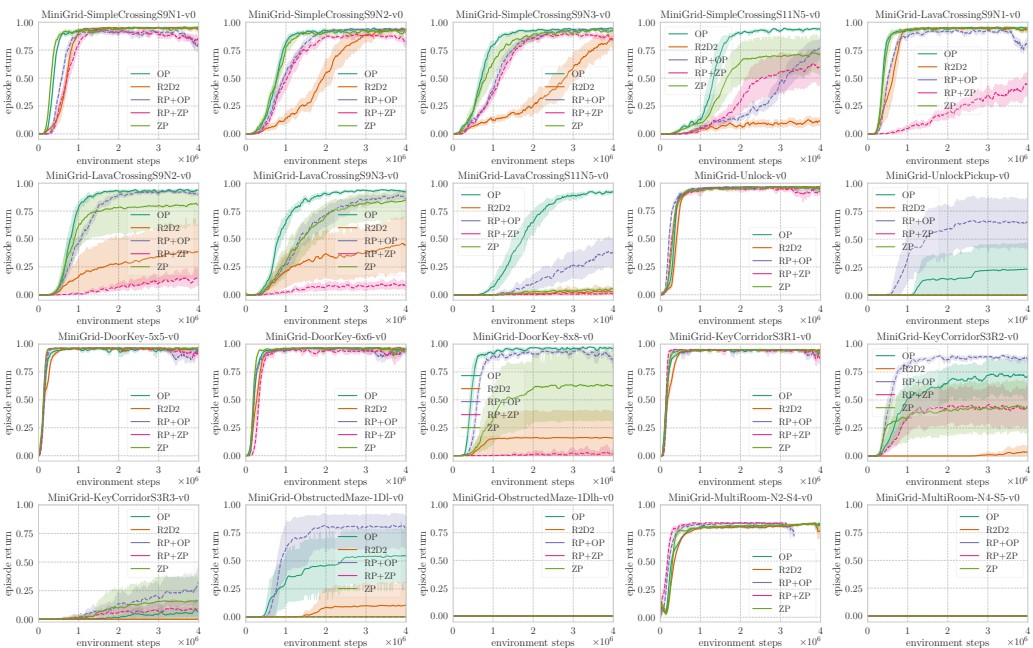

Figure 14: The **episode return** between $\phi_{Q^*}$ (R2D2), $\phi_L$ (ZP, RP + ZP), $\phi_O$ (OP, RP + OP) in 20 MiniGrid tasks over 4M steps, averaged across $\geq$ 9 seeds. The end-to-end approaches (R2D2, ZP, OP) are shown by **solid** curves, while the phased ones (RP + ZP, RP + OP) are shown by **dashed** curves. The ZP targets use EMA. Tasks sharing the same prefixes (e.g., SimpleCrossing, KeyCorridor) are arranged in order of increasing difficulty.

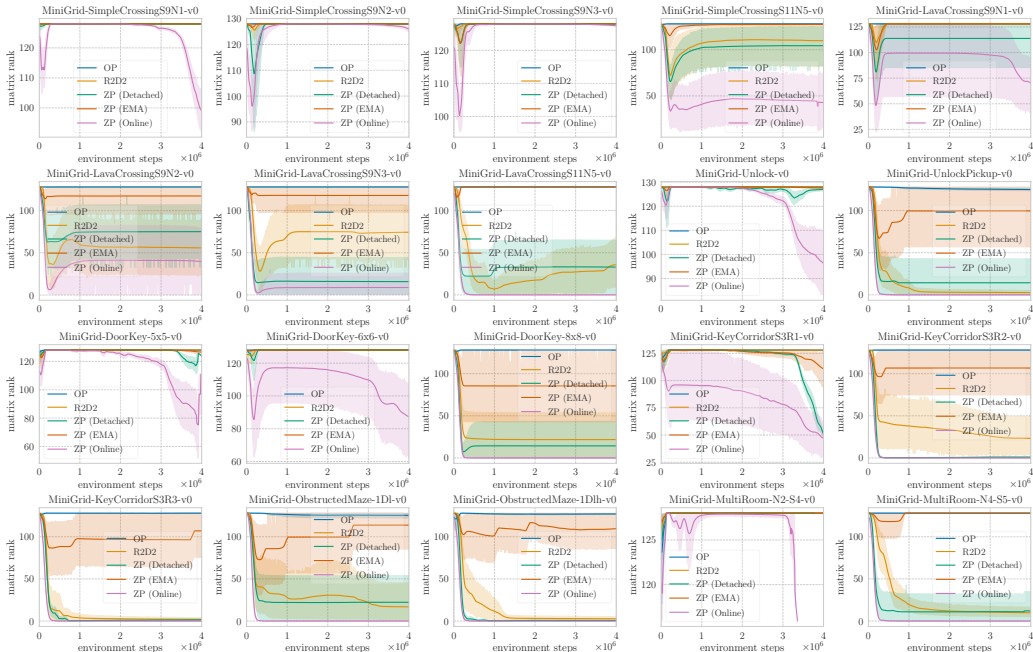

Figure 15: The estimated **matrix rank** between **ZP targets** (online, detached, EMA), R2D2, and OP in 20 MiniGrid tasks over 4M steps, averaged across $\geq$ 9 seeds. The maximal achievable rank is 128.