# OpenReview forum: "Bridging State and History Representations: Understanding Self-Predictive RL"
_ICLR.cc/2024/Conference — ICLR 2024 poster_

### Official Review · Reviewer_sXU8 · 2023-10-30

**Soundness:** 3 good
**Presentation:** 3 good
**Contribution:** 4 excellent
**Rating:** 8
**Confidence:** 3

**Summary:**

The work overviews existing abstractions for state/history representations and their "conditions", and, with an implication graph, shows how are they connected (supported by proofs). It overviews various RL algorithms and classifies them by which abstraction and conditions they use. Finally, based on their theoretical findings, the authors suggests a minimalist algorithm (and its variants), which they empirically evaluate and compare, from various angles.

**Strengths:**

The paper is concise and clearly written. It is noteworthy that the studied topic is very large and requires lot of details, and that the Appendix contains some interesting details and even novel contributions (e.g., details regarding the implication graph), which implies that a journal format would suit this work better. Nevertheless, the authors manged to fit the most interesting information in the page limit, and hence I will focus only on the main text.

Given by the number of works focusing on representation learning (Table 1), it is clear that it is an important topic. Hence the theoretical connections given in the paper between the individual parts of the abstractions is very significant to avoid double and superfluous work. Also, the contribution about stop-gradients, connection to representation collapse and the experiments showing the rank of the weight matrices is very interesting. The theoretical part of the paper is very well done and would alone justify my rating of the paper.

In the experimental section, the authors verify their theoretical contributions. The experiments about stop-gradients are good. The experiments showing different variants of the minimalistic algorithm (Figure 3) are somewhat inconclusive, or require further discussion (e.g., claims about better/similar performance compared to ALM(3), or no justification for superior performance of ALM on Humanoid). I appreciate the negative result in 5.2 for ZP objective hypothesis.

**Weaknesses:**

As mentioned, the paper sometimes outsources interesting details into the Appendix and the experiment in 5.1 is inconclusive and requires unbiased and fair discussion (also see questions).

**Questions:**

- In Sec. 5.2, you write: "Surprisingly, model-free RL (ϕ_{Q∗}) performs worse than ϕ_O." Isn't that expected?
- Given the scope of the article, why did you preferer conference vs. journal? (not answering this question can be understandable)
- Please elaborate on "This suggests that the primary advantage ALM(3) brings to model-free RL, lies in state representation rather than policy optimization" from Sec. 5.1. Isn't your model based on ALM? How does the stripped-down version of ALM (that you use as a basis for your augmentations) compare to ϕ_Q∗ (TD3)? Is it the same?

Suggestions:
- define FKL / RKL abbr. - good place good be on p. 5 after "includes forward and reverse KL".
- p. 5: detached from the computation graph [OR] using a copy ...
- in Fig. 6 caption, mention that RP+OP / RP+ZP are phased; it is not obvious what the difference is and why they should perform worse

Nitpicking:
- legend in Figure 5 is slightly different from the graphs

---

> ### Author Response · Authors · 2023-11-16
> **Authors' response**
>
> Thank you for your positive feedback!
>
> > The experiments showing different variants of the minimalistic algorithm (Figure 3) are somewhat inconclusive or require further discussion (e.g., claims about better/similar performance compared to ALM(3)).
> > Please elaborate on "This suggests that the primary advantage ALM(3) brings to model-free RL, lies in state representation rather than policy optimization" from Sec. 5.1.
>
> Please see our general response on these two points.
>
> > No justification for superior performance of ALM on Humanoid.
>
> We have recognized this gap and have included a justification in our revision for ALM(3)'s superior performance in the Humanoid task. We suggest that this is likely attributed to the effectiveness of SVG policy optimization in leveraging first-order gradient information from latent dynamics, a feature that seems particularly advantageous in high-dimensional tasks like Humanoid.
>
>
> > Isn't your model based on ALM? How does the stripped-down version of ALM (that you use as a basis for your augmentations) compare to ϕ_Q∗ (TD3)? Is it the same?
>
> Our minimalist algorithm shares the goal of learning self-predictive representations with ALM, but diverges in the approach to policy optimization. The key differences in policy optimization between our algorithm and various ALM versions are detailed in Appendix D.2 and can be summarized as follows:
>
> - ALM(3): uses SVG with 3-step horizon planning and additional intrinsic rewards
> - ALM-no-model: uses SVG with 1-step horizon planning
> - ALM(0): uses model-free TD3 in the latent space
> - Our minimalist algorithm: uses model-free TD3 in the latent space, akin to ALM(0), but with distinct encoder objectives regarding detaching latent states (see Appendix D.2 for details)
>
> Regarding your question on the comparison to $\phi_{Q^*}$ (TD3), the stripped-down version of ALM (assumed to be ALM-no-model or ALM(0)) differs from [TD3 in the original state space ($\phi_{Q^*}$)] as it operates in the latent state space and aims to learn $\phi_L$.
>
>
> > In Sec. 5.2, you write: "Surprisingly, model-free RL ($\phi_{Q∗}$) performs worse than $\phi_O$." Isn't that expected?
>
> This finding was indeed unexpected. The rationale is that model-free RL, not requiring observation prediction, would presumably be less susceptible to distractions. However, our results indicate otherwise. We have also included the explanation to the new version.
>
> > Given the scope of the article, why did you preferer conference vs. journal? (not answering this question can be understandable)
>
> We chose to submit to ICLR due to its high relevance to our research topic (representation learning) and its high impact factor. Additionally, we plan to submit to a journal for more in-depth reviews.
>
>
> > define FKL / RKL abbr. - good place good be on p. 5 after "includes forward and reverse KL".
>
> Thanks and we've added it.
>
> > p. 5: detached from the computation graph [OR] using a copy ...
>
> Here we emphasized that the copy of $\phi$ is also detached from the computation graph (i.e., it is a `deepcopy` in python), so we think "and" is more appropriate.
>
> > in Fig. 6 caption, mention that RP+OP / RP+ZP are phased; it is not obvious what the difference is and why they should perform worse
>
> The caption of Figure 6 has been updated to indicate that RP+OP / RP+ZP use a phased training approach, as detailed in Section 3.1.
>
> Our initial hypothesis did not anticipate a significant performance difference between the phased and end-to-end approaches, given that both utilize the same R2D2 backbone and target similar representations. However, the end-to-end approach was found to be at least comparable, if not superior, to the phased one in our experiments. We speculate this could be due to the phased approach dealing with accumulated errors from three separate losses (reward prediction, OP or ZP, Q-learning loss) as opposed to the end-to-end approach's two losses (OP or ZP, Q-learning loss), potentially complicating the optimization process.
>
>
> > legend in Figure 5 is slightly different from the graphs
>
> We have carefully reviewed the figure and its legend but did not identify any discrepancies. Could you please specify which part of the legend you find to be different from the graph?
>
> Hope our responses have resolved your concerns!

---

> > ### Comment · Reviewer_sXU8 · 2023-11-18
> >
> > I thank the authors for their response and appreciate the answers.
> >
> > Regarding the legend in Fig. 5 - the markers have a white border in the graphs, while in the legend they do not. It is of least priority, though.

---

> ### Author Response · Authors · 2023-11-18
>
> Thanks for your positive feedback and pointing the issue out! Upon reviewing the code, we found that the absence of borders around markers in legends seems a default behavior of the seaborn library. It is a bit hard to change this behavior. Therefore, to maintain consistency, we opted to remove the edges from the markers in the plots, which also makes the plots clearer. This change has been applied to the updated figure in the latest PDF. Please have a look!

---

### Official Review · Reviewer_kS5Z · 2023-10-31

**Soundness:** 3 good
**Presentation:** 2 fair
**Contribution:** 3 good
**Rating:** 8
**Confidence:** 4

**Summary:**

The paper proposes a unification of various state representation learning algorithms under the umbrella of self-predictive representations. They draw relations between different algorithms under their terminology and even suggest why stop-gradients are useful when learning self-predictive representations.

**Strengths:**

1. I agree very much with the paradox of choice that is stated in the paper. I think a unification is much needed, and this paper is valuable from that point of view.
2. The paper is generally well-written.
3. Table 1 and Figure 1 are useful to understand the framework.

**Weaknesses:**

See questions

**Questions:**

1. While the main message of the paper seems clear, I am struggling to understand the core message of the empirical section. Few questions:
  - Should we be using minimalist over everything else? If so, ALM outperforming minimalist in Figure 3 on multiple occasions would not support that.
  - Is minimalist better than $\phi_{Q^{*}}$? I think this is somewhat obvious, which is the reason we have all these works on representation learning (this is not a criticism, just a remark).
  - Is it broadly to categorize the entire zoo of representation learning algorithms into their essential components i.e. $\phi_{Q^*}$, $\phi_L$, $\phi_O$ and just study how they perform empirically?
  - Misc point:
     - I understand the point of the different loss function and representation collapse experiments, and those are useful.
     - It would be very useful to include $\textit{why}$ it is important to test the questions that are posed in Section 5, and what the final algorithm recommendation should be.
     - The paper discusses this paradox of choice, but I still find it unclear how to make a choice at the end of the paper.
2. The sample efficiency claims in Section 5.1 are unclear to me. It appears that ALM(3) outperforms $\phi_L$ in all cases. It is true that $\phi_L$ does better than $\phi_{Q^*}$, but the former does not seem to be. I would suspect that ALM(3) will do better because it is explicitly learning a reward model, whereas $\phi_L$ is suggesting that the reward model/representations are implicitly learned based on their implication graph.
3. Is there intuition for why $\phi_O$ may struggle with distractors? The question is posed in the empirical section, but it’s a bit unclear what motivates this question.

I should note that I do like the paper, but the above things are confusing. I would be willing to re-evaluate the score based on the response to the above.

---

> ### Author Response · Authors · 2023-11-16
> **Authors' response**
>
> Thank you for your positive feedback! In response to the main concerns about clarity, we've extensively revised our paper. This includes adding a new section on recommendations (Section 6) and a detailed discussion on motivations (Appendix G). We've also carefully addressed other writing concerns through targeted revisions and detailed responses below.
>
> > It would be very useful to include *why* it is important to test the questions that are posed in Section 5.
> > Is there intuition for why $\phi_O$ may struggle with distractors? The question is posed in the empirical section, but it’s a bit unclear what motivates this question.
>
> Thank you for highlighting these points! To address your queries, we have detailed the motivation behind each hypothesis in **Appendix G ("Motivating Our Hypotheses")**. This section aims to clarify the rationale behind our research questions, including the specific inquiry about the performance of $\phi_O$ in the presence of distractors. We invite you to review this dedicated section and would appreciate your feedback on its clarity.
>
> > It would be very useful to include what the final algorithm recommendation should be.
>
> This a very good point! Please see our general response on our **recommendations**.
>
> > Should we be using minimalist over everything else? If so, ALM outperforming minimalist in Figure 3 on multiple occasions would not support that.
>
> Our intention is not to position the minimalist algorithm as superior to others, such as ALM, in all aspects. The primary goal of the minimalist algorithm is to provide a straightforward means to validate the hypotheses derived from our theoretical framework. Its simplicity aids in easy implementation and understanding. Additionally, it serves as a baseline for future research, enabling separate examination of the impacts of representation learning and policy optimization.
>
> As an example, in Section 5.1, we discuss how ALM's performance advantage over TD3 at 500k steps in MuJoCo tasks is largely attributable to its approach to representation learning, illustrating the utility of the minimalist algorithm in isolating these factors for analysis.
>
> > Is minimalist better than $\phi_{Q^*}$? I think this is somewhat obvious, which is the reason we have all these works on representation learning (this is not a criticism, just a remark).
>
> Our experiments demonstrate that the minimalist algorithm indeed outperforms $\phi_{Q^*}$ in the tasks we evaluated, which include standard MuJoCo, distracting MuJoCo, and MiniGrid. However, it's important to note that these findings may not generalize to every RL task. The performance of deep RL algorithms can vary significantly depending on the specific task structure.
>
> > Is it broadly to categorize the entire zoo of representation learning algorithms into their essential components i.e. $\phi_{Q^*},\phi_L,\phi_O$, and just study how they perform empirically?
>
> Yes, we think it is indeed a good future direction for empirical studies.  Alongside this, we also see significant potential in developing new theoretical frameworks. These could include understanding the interplay between representation learning and policy optimization, as well as establishing formal guarantees about the relative sample efficiency of learning different representations, such as $\phi_L$ over $\phi_O$ in distracting tasks, and $\phi_O$ over $\phi_{Q^*}$ in tasks with sparse rewards.
>
> > The sample efficiency claims in Section 5.1 are unclear to me. It appears that ALM(3) outperforms $\phi_L$ in all cases.
>
> Our data shows varied results across different tasks. In Walker2d and HalfCheetah, $\phi_L$ outperforms ALM(3), while in the Ant task, they demonstrate comparable sample efficiency *at 500k steps*. However, in the Humanoid task, ALM(3) significantly outperforms $\phi_L$. For further clarification, please refer to our general response on this topic.
>
>
> > I would suspect that ALM(3) will do better because it is explicitly learning a reward model, whereas $\phi_L$ is suggesting that the reward model/representations are implicitly learned based on their implication graph.
>
> We want to clarify that although ALM(3) explicitly learns a reward model, its encoder objective does *not* include reward prediction, as evidenced by [their code](https://github.com/RajGhugare19/alm/blob/main/agents/alm.py#L209) uses `z_dist.sample()` that detaches the latent states when learning the reward model. Therefore, it's unlikely that explicit reward prediction is a significant factor in learning an effective encoder. Rather, ALM(3)'s enhanced performance can more plausibly be attributed to its use of SVG planning for policy optimization. This clarification has been added to the new version of our paper.
>
>
> Hope our responses have resolved your concerns!

---

> > ### Comment · Reviewer_kS5Z · 2023-11-17
> >
> > Thank you to the authors for their response. I am satisfied with the response. I do like the paper and will raise the score. I think the documentation/unification of all these algorithms and distilling them down to their core components is valuable.

---

### Official Review · Reviewer_5DUm · 2023-11-01

**Soundness:** 3 good
**Presentation:** 3 good
**Contribution:** 3 good
**Rating:** 8
**Confidence:** 3

**Summary:**

The paper proposes a conceptual framework for unifying many existing techniques for decision-making-focused representation learning based on the concept of "self-prediction". Within this framework, the paper then proposes a simple and seemingly novel "minimalist" approach to construct representations simply by training a model-free agent and simultaneously learning an abstract forward model while preventing gradients from flowing backward through the encoder from the targets. The framework allows a large number of representation learning methods to be evaluated side-by-side while controlling for other factors, which enables the paper to present and evaluate several hypotheses about the various learning objectives. The experiments suggest that the proposed "minimalist" approach works well.

**Strengths:**

Overall, this is a nice contribution!

The shared framework is helpful for comparing a seemingly endless number of decision-making-focused representation learning methods. It distills these methods down to their core representation learning ideas, which allows for comparing those ideas without needing to worry about the remaining complexity (such as the particular RL algorithm employed). I suspect the community will find this quite valuable.

The paper also offers interesting insights, such as the fact that online targets do not guarantee the same fixed points as the objectives with stop-gradients.

The proposed "minimalist" approach is extremely simple and clean. It's shocking that this hasn't been tried before, yet I cannot think of an example where it has. Time will tell if this approach generalizes to other problems, but if it does, it will greatly simplify many projects that rely on representation learning.

The experiments seem well designed, the baselines seem well chosen, and the results look promising. I particularly liked the experiments measuring the change in matrix rank over time---nice job. That is clear evidence that the method avoids representation collapse.

**Weaknesses:**

My concerns are minor.

- I'm not familiar with using the phrase "distracted MDPs" to mean MDPs with distracting elements.
- Maybe I'm wrong, but I feel like we already know that ZP + $\phi_{Q*}$ implies RP.
- Top of p5: The use of $\mathbb{P}$ and $\mathbb{Q}$ is confusing, given that $P$ and $Q$ are often overloaded in RL.
- Towards end of sec 4.2, last para: "...cosine similarity between columns of the learned $\phi$. As expected by Thm. 3, [...] stay several orders of magnitude smaller when using stop-gradient." How does Thm 3 predict low absolute cosine similarity? Because we start with full rank?
- Fig 3. Kind of hard to distinguish lines.
- Sec 5, first para: would be helpful to summarize the findings when introducing the hypotheses.
- p7, penultimate para: I don't know about similar sample efficiency in ant. It seems a lot worse. I also don't know if I agree with the conclusion that "the primary advantage ALM(3) brings to model-free RL, lies in the state representation rather than policy optimization, except for Humanoid." I feel like it's too soon to conclude something as sweeping as that.
- Fig 5. A little difficult to read. See if maybe log return would help?
- Top of p9, minigrid experiment. This feels like a minor bait-and-switch. Normally minigrid uses pixels, no? And why does detached rank drop on minigrid when theory suggests it should remain high?
- "Validation of end-to-end hypothesis" (and throughout). It's a bit hard to keep track of which is which. Text uses $\phi_o$, $\phi_L$, but fig uses OP, ZP, RP+OP, RP+ZP.

**Questions:**

Do the authors have a clear recommendation on when to use OP vs ZP vs RP+[*]? It would be nice to have some clear takeaways after all this analysis.

---

> ### Author Response · Authors · 2023-11-16
> **Authors' response**
>
> Thank you for your positive feedback!
>
> > I'm not familiar with using the phrase "distracted MDPs" to mean MDPs with distracting elements.
>
> We appreciate this observation and have made adjustments to clarify our terminology. We've changed "distracted MDPs" to "distracting MDPs" in our paper. Additionally, we've included a footnote stating, "Distracting MDPs refer to MDPs with distracting observations irrelevant to optimal control in this work," to prevent any potential confusion.
>
> > Maybe I'm wrong, but I feel like we already know that ZP + $\phi_{Q^*}$ implies RP.
>
> You raise a valid point. While the connection between ZP + $\phi_{Q^*}$ and RP might seem intuitive, our work contributes to formalizing this relationship. The inverse Bellman equation in our proof, also used in imitation learning literature [Garg et al., IQ-Learn: Inverse soft-Q Learning for Imitation], provides a novel perspective specific to state or history representations for RL. To the best of our knowledge, this precise articulation is new in both MDPs and POMDPs.
>
> > Top of p5: The use of $\mathbb P$ and $\mathbb Q$ is confusing, given that $P$ and $Q$ are often overloaded in RL.
>
> We acknowledge this potential confusion. To address it, we have now replaced $\mathbb Q$ with $\mathbb P$ in our paper, eliminating the notational conflict and avoiding confusion with Q-values in RL.
>
> > How does Thm 3 predict low absolute cosine similarity? Because we start with full rank?
>
> In this experiment, the latent state dimension is set to $2$ and we orthogonally initialize the $\phi$ matrix. This results in $\phi^\top \phi = [1, 0; 0, 1]$ initially. According to Thm 3,  $\phi^\top \phi$ maintains its initial value in continuous-time. Thus, the cosine similarity (and the inner product) between the two columns of $\phi$ will remain close to $0$ in discrete-time.
>
> Therefore, Thm 3 predicts low absolute cosine similarity because we initialize $\phi$ as an orthogonal matrix, which is a stronger condition of having full rank.
>
> > Fig 3. Kind of hard to distinguish lines.
>
> Acknowledging the difficulty in distinguishing lines in Figure 3, we also want to point out that the overlap of curves mainly illustrates similar performance trends. Here we offer a qualitative comparison at 500k steps using symbols: $<$ for worse, and $\ll$ for much worse. Hope this will help you read the figure.
>
> - Walker2d: ALM(3) and TD3 (they are quite similar) $\ll$ OP and ZP methods (they are quite similar).
> - HalfCheetah: TD3 $<$ ALM(3) $<$ OP methods $<$ ZP methods.
> - Ant: TD3 $\ll$ OP-FKL $<$ the other methods. Note that ALM(3) initially leads but later declines.
> - Humanoid: TD3 and ZP methods $<$ OP methods $\ll$ ALM(3).
>
> > Sec 5, first para: would be helpful to summarize the findings when introducing the hypotheses.
> > Do the authors have a clear recommendation on when to use OP vs ZP vs RP+[*]? It would be nice to have some clear takeaways after all this analysis.
>
> Thanks for your suggestion! We've added our findings to the paper and please the general response on **recommendations**.
>
> > p7, penultimate para: I don't know about similar sample efficiency in ant. It seems a lot worse. I also don't know if I agree with the conclusion that "the primary advantage ALM(3) brings to model-free RL, lies in the state representation rather than policy optimization, except for Humanoid." I feel like it's too soon to conclude something as sweeping as that.
>
> Please see our clarification on this in our general response.
>
> > Fig 5. A little difficult to read. See if maybe log return would help?
>
> We tried to plot the log returns but found it did not help clarity much. This is due to the scale of episode returns, which are mainly around $10^3$, with the exception of the Ant. In the Ant with $\ge 2^7$ distractors, all methods are close to $0$, indicating they are equally bad. For the other environments with large number of distractors, the results clearly show that TD3 is the worst performer. Following this, OP-$\ell_2$, OP-FKL, and ZP-FKL exhibit similar performance levels. ZP-$\ell_2$ and ZP-RKL stand out as the best performers.
>
> > And why does detached rank drop on minigrid when theory suggests it should remain high?
>
> We acknowledge that the observed drop in detached rank in the MiniGrid experiment is intriguing, especially since our theory does not predict a significant degradation in such cases. We plan to explore this further in our future research.
>
>
> >"Validation of end-to-end hypothesis" (and throughout). It's a bit hard to keep track of which is which. Text uses $\phi_O, \phi_L$  but fig uses OP, ZP, RP+OP, RP+ZP.
>
> To address this, we have revised the text in the new version of our paper to enhance readability and ensure consistency between the text and figures.

---

> > ### Author Response · Authors · 2023-11-17
> > **Authors' response (cont.)**
> >
> > > Top of p9, minigrid experiment. This feels like a minor bait-and-switch. Normally minigrid uses pixels, no?
> >
> > In our MiniGrid experiment, we used vector-based (symbolic) observations, which is a standard option in MiniGrid, rather than the pixel-based representation. This approach aligns with common practices in recent research. For instance:
> >
> > - Parker-Holder et al., Evolving Curricula with Regret-Based Environment Design, NeurIPS 2022. (They used "147-dimensional observation".)
> > - Campero et al., Learning with AMIGo: Adversarially Motivated Intrinsic Goals, ICLR 2021. (They mentioned that the observations are "symbolic".)
> > - Zhang et al., NovelD: A Simple yet Effective Exploration Criterion, NeurIPS 2021.
> > - Jiang et al., Replay-Guided Adversarial Environment Design, NeurIPS 2021.
> > - Seo et al., State Entropy Maximization with Random Encoders for Efficient Exploration, ICML 2021. ([Their code](https://github.com/younggyoseo/RE3/blob/master/a2c_re3/rl-starter-files/rl-starter-files/utils/env.py) indicates they use symbolic observations.)
> > - Dennis et al., Emergent Complexity and Zero-Shot Transfer via Unsupervised Environment Design, NeurIPS 2020.
> > - Goyal et al., InfoBot: Transfer and Exploration via the Information Bottleneck, ICLR 2019. (They mentioned that the observations are "not RGB images".)
> >
> > Hope our responses have resolved your concerns!

---

> > > ### Comment · Reviewer_5DUm · 2023-11-22
> > > **Thanks for your response**
> > >
> > > Thank you for the response. I think this work will be valuable for helping make sense of the connections between all these different methods.

---

### Official Review · Reviewer_fv1F · 2023-11-01

**Soundness:** 3 good
**Presentation:** 1 poor
**Contribution:** 2 fair
**Rating:** 3
**Confidence:** 2

**Summary:**

The paper attempts at providing a unified view at self-predicting reinforcement learning. Different self-predicting representation targets are described, and the prior work is classified according to the target learned. Based on the unified view, a minimalist algorithm learning a self-predicting representation is learned. The algorithm is evaluated on a set of benchmarks.

**Strengths:**

The paper encompasses a broad range of recent and ongoing work on self-predicting RL, providing a unifying view. Theoretical results are presented, with proofs in the appendix. An algorithm proposed in the paper is evaluated on benchmarks.

**Weaknesses:**

The results in the paper are either trivial or indecisive. The paper is built around an insight that different works do similar things trying to optimize self-prediction of certain features, but this is, in my opinion, trivial. A classification of things that can be optimized for self-prediction is worth a survey, but this paper is not a survey. The paper uses a lot of abbreviations, the proofs a sketchy and uncommented, and veryfiying or even following the proofs takes tremendous effort.

The empirical evaluation is unconvincing. According to the plots presented in the paper, the proposed unified algorithm does not outperform (and does not always perform comparably) to algorithms from the literature. Looking at the algorithm pseudocode and implementation, this is not surprising, given that the 'minimalist algorithm' is more of a boilerplate, which, when filled with details, reduces to one of the earlier published algorithms. However, any practical implementation requires attending to details, which the unifying minimalist algorithms fails to achieve.

The paper would be extremely hard to follow, in my opinion, for an outsider, or for someone less familiar with the slang of a particular research group. For example, the paper discusses (and presents theoretical results) wrt to "stop-gradient" technique without formally defining the technique (which is described in passing and requires referring to the cited sources to understand the paper).

I believe that his research may have a potential, but for a publication, I would suggest deciding on a small subset of ideas among those sketched in this submission, presenting them thoroughly and rigorously, with proofs that are possible to follow, and accompanied by an implementation that brings competitive results, in some form.

**Questions:**

In the introduction, you are writing "However, this abundance of
methods may have inadvertently presented practitioners with a “paradox of choice”, hindering their
ability to identify the best approach for their specific RL problem."

How does your paper help practitioners identify and use the best approach for their specific RL problem? Can you give an example of application? For example, suggest (and describe) a simple RL problem, and show how your result help choose the best solution.

---

> ### Author Response · Authors · 2023-11-16
> **Authors' response**
>
> Thank you for your feedback! It seems the main concern is the limited contribution. We agree that many prior methods are based on a similar intuition of self-prediction. However, these methods come from different perspectives and have various theoretical properties with nuanced-yet-crucial implementation differences (see Table 1 and Appendix C). Our objective is to contextualize these varied approaches within a unified framework, aiming to distill fundamental principles of representation learning in RL. This endeavor seeks not just to deepen theoretical understanding and prevent overlapping research efforts, but also to offer practical guidance to practitioners (Section 6), as elaborated in our general response.
>
>
> > How does your paper help practitioners identify and use the best approach for their specific RL problem?
>
> We have included recommendations for practitioners in Section 6 in the revision. For more detailed information, please refer to our general response on **recommendations**.
>
> > The results in the paper are either trivial or indecisive. The paper is built around an insight that different works do similar things trying to optimize self-prediction of certain features, but this is, in my opinion, trivial.
>
> We respectfully disagree with the view that our work is trivial or unconvincing.  Contrary to this opinion, the other three reviewers (5DUm, kS5Z, sXU8) have recognized the significance and value of our unified framework, which is much needed to the RL community. 5DUm noted its utility in enabling comparisons of various methods based on their core representation learning principles without being confounded by other components. sXU8 commented on its potential to prevent redundant research efforts.
>
>
> > The paper uses a lot of abbreviations.
>
> To enhance readability, we have implemented clickable abbreviations that link directly to their definitions. Additionally, we occasionally spell out the full names of these abbreviations to aid recall.
>
> > The proofs a sketchy and uncommented, and verifying or even following the proofs takes tremendous effort.
>
> Are there any specific proofs that you found challenging to follow? We would appreciate your pointing them out and offer further clarifications as needed.
>
> > The empirical evaluation is unconvincing. According to the plots presented in the paper, the proposed unified algorithm does not outperform (and does not always perform comparably) to algorithms from the literature.
>
> The aim of the experiments and the paper as a whole is not to propose a SOTA or competitive method, but rather understand various self-predictive methods within the broader context of representation learning. The empirical evlauation is intended solely to validate our hypotheses suggested by our theory. In fact, our empirical results validate most of our hypotheses.
>
> Notably, the findings related to stop-gradients were particularly convincing, as acknowledged by the other reviewers (5DUm, kS5Z, sXU8). For detailed clarification on the comparisons with ALM(3), please refer to our general response.
>
> > Looking at the algorithm pseudocode and implementation, this is not surprising, given that the 'minimalist algorithm' is more of a boilerplate, which, when filled with details, reduces to one of the earlier published algorithms. However, any practical implementation requires attending to details, which the unifying minimalist algorithms fails to achieve.
>
> To the best of our knowledge, our algorithm, employing end-to-end training with a self-predictive auxiliary task, is novel within the POMDP literature. Is the reviewer aware of prior work that can be reduced to our algorithm in POMDPs?
>
> In addition, we do care about practical implementation in our algorithm. For example, we tuned the coefficient of the auxiliary task for each benchmark. We also base our algorithm on existing, well-tuned implementations, rather than writting a new one from scratch.

---

> > ### Author Response · Authors · 2023-11-16
> > **Authors' response (cont.)**
> >
> > > The paper would be extremely hard to follow, in my opinion, for an outsider, or for someone less familiar with the slang of a particular research group. For example, the paper discusses (and presents theoretical results) wrt to "stop-gradient" technique without formally defining the technique (which is described in passing and requires referring to the cited sources to understand the paper).
> >
> > We regret to hear that the paper is extremely hard to follow, particularly as the other reviewers (kS5Z, sXU8, 5DUm) have commended its clarity. Could the reviewer give more examples on the clarity issue?
> >
> > Regarding the "stop-gradient" technique, we have made an effort to define it clearly immediately following Equation 3, without requiring external references. We would appreciate it if the reviewer could pinpoint which aspect—online, detached, or EMA—is causing confusion.
> >
> > Moreover, the suggestion that our work employs slang specific to a particular research group seems unfounded. The concepts we use are drawn from a diverse array of sources across the field, reflecting a broad spectrum of research. For instance, the three abstractions are derived from Li et al., 2006, and Subramanian et al., 2022. The term "self-predictive representations" originates from Schwarzer et al., 2020. Our stop-gradient analysis builds upon Tang et al., 2022, and the ALM(3) is from Ghugare et al., 2022. Notably, none of these papers share co-authors or institutional affiliations.
> >
> > > for a publication, I would suggest deciding on a small subset of ideas among those sketched in this submission, presenting them thoroughly and rigorously, with proofs that are possible to follow, and accompanied by an implementation that brings competitive results, in some form.
> >
> > Thank you for the suggestions and we've improved our writing in the revision.
> >
> > > Can you give an example of application? For example, suggest (and describe) a simple RL problem, and show how your result help choose the best solution.
> >
> > In Section 5.2 and 5.3, we have already included two concrete RL examples to  guide practitioners in selecting the most suitable approach for their specific RL challenges. Our empirical findings suggest that for distracting MDPs, it'd be better to predict latent states (e.g., using our minialist algorithm) instead of predicting observations. For sparse-reward POMDPs, it'd be better to predict observations (e.g., using end-to-end learning) instead of predicting latent states or using model-free RL.

---

> > ### Comment · Reviewer_fv1F · 2023-11-18
> >
> > I am going to re-read the paper carefully and update my review based on this. It possible that I was too quick judging the paper, based on other reviewers' opinions. It is also possible that the paper's style is beyond my mental ability of comprehension.

---

> > > ### Author Response · Authors · 2023-11-18
> > >
> > > Thank you for your quick response! If you have any question during re-evaluation, please let us know!

---

> > > ### Comment · Reviewer_fv1F · 2023-11-21
> > > **More detailed comments`**
> > >
> > > I've taken time to re-read the paper. Here are my notes (I am not picky, if you want me to be picky there will be ten times more such notes) from 4.5 first pages (and one proof from the appendix).
> > >
> > > Page 2: "encoder of the value function" -> "encoding learned with/for value function"
> > >
> > > Page 2: next latent state prediction --- the distribution of the encodings rather than the state is predicted.
> > >
> > > Page 2: "encoder is capable of predicting" the encoder is defined only encodes the state.
> > >
> > > Page 3: "expected next latent state" --- the latent state is not defined on a codomain for which the mean can be defined (e.g. R^n or Z^n).
> > >
> > > Page 4: "The bootstrapping effect ..." phi does not in "both sides of the next latent state prediction" explicitly, one must guess that z' = phi(h') .
> > >
> > > Page 4-5" "we parameterize an encoder .. deterministic case ... stochastic case. This is not a stochastic encoder, but a probabilistic one. A probabilistic encoder returns a distribution \Delta(Z), a stochastic encoder returns a sample from a distribution. This is significantly different. There is almost no way to train a 'deterministic encoder' using differentiable models (deep networks).
> > >
> > > Page 5: after expr (1), whjere D(., .) \in R compares two distribution. When (1) reaches minimum --- it does not have to, functions on R do not have to have a minimum.If it is on R+, then it still would be good to know what features D should possess.
> > >
> > > Page 5: Equations (2) and (3) expectations are on variable o' which does not appear in the expression under expectation.
> > >
> > > Page 5: D (apparently divergence) uses different notation in (1) --- D(., .) and in (3) D(.||.) . This is important because D(., .) commonly implies a metric, or at least a symmetric quantity, while D(.||.) commonly implies assymetric function, such as divergence.
> > >
> > > Proof of proposition 1 (appendix): second terms in eqs (90) and (91) are mutually cancelled, despite being different.

---

> > > > ### Comment · Reviewer_fv1F · 2023-11-21
> > > > **Updated impression**
> > > >
> > > > My updated impression is that the paper is very sloppily written, hard to follow, and that the proofs can't be trusted. The first proof I checked had an obvious error.
> > > >
> > > > The paper may have some great ideas, but it must significantly rewritten and improved before it can be considered for publication.

---

> ### Author Response · Authors · 2023-11-21
>
> Thank you for your valuable feedback! We have addressed the main concerns around some confusing terminologies and the seemingly incorrect proof in the appendix. Below are our responses to each point, with the changes in the updated PDF highlighted in magenta.
>
> > Page 2: "encoder of the value function" -> "encoding learned with/for value function"
>
> We've fixed this term to be "the encoder *learned for* the value function" in the new revision to avoid confusion.
>
> > Page 2: next latent state prediction --- the distribution of the encodings rather than the state is predicted.
>
> We've renamed this term to be "next latent state *distribution* prediction (ZP)" to avoid confusion.
>
>
> > Page 2: "encoder is capable of predicting" the encoder is defined only encodes the state.
>
> We've changed this to "the encoder *can be used to* predict" for clearer understanding.
>
> > Page 3: "expected next latent state" --- the latent state is not defined on a codomain for which the mean can be defined (e.g. R^n or Z^n).
>
> We've added this assumption into Appendix A.2 (see "Remark on the latent state distribution") to address your concern.
>
> > Page 4: "The bootstrapping effect ..." phi does not in "both sides of the next latent state prediction" explicitly, one must guess that z' = phi(h') .
>
> We've added this explanation "since $z'$ also relies on $\phi(h')$" to make it clearer.
>
> > Page 4-5" This is not a stochastic encoder, but a probabilistic one. A probabilistic encoder returns a distribution \Delta(Z), a stochastic encoder returns a sample from a distribution. This is significantly different.
>
> We've renamed all instances of "stochastic encoder" to "*probabilistic encoder*" to avoid confusion between these two concepts.
>
>
> > There is almost no way to train a 'deterministic encoder' using differentiable models (deep networks).
>
> We believe we can, and actually have, trained a deterministic encoder with neural networks in our implementation. The deterministic encoder $f_\phi$ takes input a history $h$ and outputs the latent state $z$. We assume $f_\phi$ is differentiable w.r.t. $\phi$ for any history $h$, which is the standard assumption in deep RL.
>
> > Page 5: after expr (1), where D(., .) \in R compares two distribution. When (1) reaches minimum --- it does not have to, functions on R do not have to have a minimum.If it is on R+, then it still would be good to know what features D should possess.
>
> We've changed $\mathbb R$ to $\mathbb R_{\ge 0}$ (the set of positive reals and zero) to ensure the existence of a minimum.
>
> > Page 5: Equations (2) and (3) expectations are on variable o' which does not appear in the expression under expectation.
>
> We want to clarify that $o'$ indeed appears in the expression through $h' = (h, a, o')$. We define $h'$ in the "MDPs and POMDPs" in the background section.
>
> > Page 5: D (apparently divergence) uses different notation in (1) -- $D(., .)$ and in (3) $D(.||.)$.
>
> We've unified the notation to $\mathbb D(\cdot\mid \mid \cdot)$ to avoid confusion. Since we allow both symmetric and asymmetric functions, we use this notation.
>
> > Proof of proposition 1 (appendix): second terms in eqs (90) and (91) are mutually cancelled, despite being different.
>
>
> We believe our proof is correct. The two cancelled terms in (90) and (91) are equal due to the following elementary fact:  Inner product is a bilinear form, i.e., $\langle a + b, c \rangle = \langle a, c \rangle + \langle b, c \rangle$. Therefore, it follows that for a constant vector $a$ and random vector $X$ (with PDF $P_X$),
> $$
> \mathbb E [ \langle a, X \rangle ] = \int_{x} P_X(x) \langle a, x \rangle dx = \int_{x}  \langle a, P_X(x) x \rangle dx = \langle a, \int_x P_X(x)x dx \rangle = \langle a, \mathbb E[X] \rangle.
> $$
> Note that in our case, $g_\theta(f_\phi(h),a)$ does not depend on $o'$, thus it can be treated as a constant for the derivation.
>
> We hope our responses and the revisions in the manuscript have resolved the concerns raised. Looking forward to your feedback!

---

> > ### Comment · Reviewer_fv1F · 2023-11-22
> > **not resolved**
> >
> > There are many more such shortcomings in the paper. I am not here to check and fix it, just to point at the current situation.
> >
> > Regarding the proof of proposition 1: the proposition 1 is a known fact: minimizing squared error minimizes error of the mean of the distribution. However, the proof is written is wrong in the sense that it does not prove anything, unless you add clarifications.
> >
> > Regarding deterministic encoder. Any encoder is probabilistic. If it "outputs latent state" and you use some loss with regard to that state, you assume a distribution of the latent state for which the loss is the log likelihood. For example, mean squared loss is the assumption of normally distributed latent state. My comment was about incoherence of the paper, there is no deterministic vs. stochastic encoder dilemma, the encoder is always probabilistic, with the difference in the assumed output distribution. You are discussing the difference between assuming parametric, e.g. normal (or Laplace for MAE, for example) distribution and a non-parametric (e.g. categorical on a sample set or density estimated by a network) distribution on the output on the encoder.
> >
> > I can further elaborate on problems in the paper, but I will refrain from that.  The paper is very sloppy. It should be fully rewritten and properly prepared before it can be published, in my opinion. Not in this form.

---

> > > ### Comment · Reviewer_5DUm · 2023-11-22
> > > **Re: deterministic encoders**
> > >
> > > I have to side with the authors regarding deterministic encoders. If the function mapping from input to latent state is a deterministic function, then it's a deterministic encoder, regardless of what loss you want to use. There is a nice interpretation that MSE corresponds to a Gaussian assumption, but the encoder itself can be deterministic nevertheless. It may produce an output that is a random variable, and that random variable may be approximately normally distributed, but if the function is deterministic, I'm inclined to call it a deterministic encoder.

---

> > > ### Author Response · Authors · 2023-11-22
> > >
> > > Thank you, Reviewer fv1F, for your prompt and insightful feedback! It seems that most of your *detailed concerns* have been resolved. Below are our responses, together with the updated PDF, to your remaining concerns.
> > >
> > > We also appreciate Reviewer 5DUm's engagement with this discussion and their insightful comment!
> > >
> > > > The proposition 1 is a known fact: minimizing squared error minimizes error of the mean of the distribution.
> > >
> > > We agree that the proof of Proposition 1 **is related to** this known fact "minimizing squared error minimizes error of the mean of the distribution". However, we want to clarify that Proposition 1 itself is **novel** in the context of showing the connection between the ideal and practical objectives in RL. It has an implication that in deterministic tasks, it suffices to use the practical $\ell_2$ objective, which we believe is also a novel and useful insight.
> > >
> > > > However, the proof is written is wrong in the sense that it does not prove anything, unless you add clarifications.
> > >
> > > We kindly hope that the following explanation will help you understand why the proof indeed proves the proposition 1.
> > >
> > > The proof first shows that $\mathcal L_{ZP,\ell}(\phi,\theta; h,a)  \le J_{\ell}(\phi,\theta,\phi;h,a)$, which states that the ideal objective is upper bounded by the practical objective. Then the proof points out that the *ideal* objective can only lead to EZP (expected ZP) condition. Thus, we draw to conclusion (Proposition 1): The practical objective $J_{\ell}(\phi,\theta,\phi;h,a)$ is an upper bound of the ideal objective $\mathcal L_{ZP,\ell}(\phi,\theta;h,a)$ that targets EZP condition.
> > >
> > > To enhance clarity, we've inserted a note in magenta in the updated PDF, specifically explaining the cancellation of inner product terms due to bilinearity, which underpins part of the proof's logic.
> > >
> > > > Regarding deterministic encoder.
> > >
> > > First, we appreciate and agree with your point on "mean squared loss is the assumption of normally distributed latent state", also recognized by Reviewer 5DUm. We also agree with you that "Any encoder is probabilistic" in the sense that deterministic encoder is a special case of probabilitic encoders, which we've acknowledged as a Footnote 5 in the latest PDF.
> > >
> > > Building upon Reviewer 5DUm's response, we would like to further justify our consideration of deterministic encoders. This approach is grounded in a large body of prior works such as [Gelada et al., 2019; Schwarzer et al., 2020; Tomar et al., 2021; Hansen et al., 2022; Ye et al., 2021] that employ deterministic encoders with Eq. 2. Our adherence to this established notation facilitates a understanding of these studies within our proposed unified framework. Additionally, as written in a footnote in our paper, it's noteworthy that the optimal encoder can, in fact, be deterministic. This makes the discussion of deterministic encoders relevant.  The rationale for this follows that the exact belief update process in POMDPs is deterministic, i.e., $b(h') = F(b(h), a, o')$ where $b$ is the encoder for belief abstraction and $F$ is a deterministic function. We've included this reasoning into the Footnote 5 for clarity.
> > >
> > > We hope these comments will resolve your remaining concerns!

---

### Author Response · Authors · 2023-11-16
**Outline of Our Revision**

In response to the insightful suggestions from the reviewers, we have made several updates to our paper. These revisions are highlighted in magenta in the updated PDF for easy identification. Here is the outline:
- **Experimental details**, including distracting MDPs, sample efficiency at 500k steps, and the other points raised by the reviewers.
- **Recommendations for practitioners**. We've included a new section of recommendations for practitioners in Sec. 6. Please see our general response for details. Due to the space limit, we move the discussion section to Appendix H.
- **Motivation**. We've written a section in Appendix G to motivate our hypotheses.

---

### Author Response · Authors · 2023-11-16
**General Response**

## Recommendations for choosing algorithms (fv1F, kS5Z, 5DUm)

The reviewers ask us about our final recommendations for RL practitioners. This is a good point. Although we have given the validation result for each hypothesis in the original paper, it is better to also provide a final summary. Thus, in the revision we have written our recommendations in Sec. 6. Below we copy them for your reference. We would appreciate any feedback on whether these guidelines are helpful and meet the expectations raised in your reviews.

**Recommendations.** Drawing from our theoretical insights and empirical results, we suggest the following guidance for RL practitioners:

1. **Analyze your task first.** For example, in noisy or distracting tasks, consider using self-predictive representations. In sparse-reward tasks, consider using observation-predictive representations. In deterministic tasks, choose $\ell_2$ objectives for representation learning.
2. **Use our minimalist algorithm as your baseline.** Our algorithm allows for an independent evaluation of representation learning and policy optimization effects.  Start with end-to-end learning and model-free RL for policy optimization.
3. **Implementation tips.** For our minimalist algorithm, we recommend adopting the $\ell_2$ objective with EMA ZP targets first. When tackling POMDPs, start with recurrent networks as the encoder.



## Clarifying our interpretation on comparing ALM(3) (all reviewers)


> Can you elaborate on "This suggests that the primary advantage ALM(3) brings to model-free RL, lies in state representation rather than policy optimization"?

In our study, we observe that in 3 out of the 4 tasks (Walker2d, HalfCheetah, Ant), our minimalist algorithm reaches similar efficiency to ALM(3) *at 500k steps* and outperformed $\phi_{Q^*}$. Thus, in our comparison between ALM(3) and our algorithm, both targeting similar representation learning objectives, we find that the choice of policy optimization method (model-free TD3 vs. model-based SVG) has a minimal impact on task performance in these 3 tasks.

To emphasize the context of our findings, we have revised our paper to read: "the primary advantage ALM(3) brings to model-free RL **in these MuJoCo tasks**, lies in state representation rather than policy optimization."


> Claims about better/similar performance compared to ALM(3) are somewhat inconclusive, as it appears that ALM(3) outperforms $\phi_L$ in all cases.

Acknowledging ALM(3)'s initial superiority in the Ant task, we emphasize that our algorithm is on par with it in sample efficiency at 500k steps.  We've specified this in our revised paper as: "... in the Ant ... our minimalist $\phi_L$ can attain similar or even better sample efficiency compared to ALM(3) **at 500k steps**."

---

### Meta-Review · Area_Chair_vV3r · 2023-12-07

**Metareview:**

The author provide an overview of existing representation learning objectives for reinforcement learning and experimentally investigate a particularly simple (and elegant) instance of this design space. As suggested by reviewers, the paper now also makes practical recommendations that will be valuable to practitioners. Furthermore, the authors show interesting empirical comparisons that indicate that EMA targets may not be necessary for self-predictive representations and stop-gradient may be sufficient. While the novelty of the unified perspective is somewhat limited and with a focus on representation learning, one would expect evaluations on environments with image inputs instead of just proprioceptive or symbolic inputs, the paper provides a comprehensive overview and simple baseline with value to the research community.

**Justification For Why Not Higher Score:**

Limited novelty and empirical performance, no experiments with image inputs

**Justification For Why Not Lower Score:**

Comprehensive overview, simplicity

---

### Decision · Program_Chairs · 2024-01-16

Accept (poster)